# The lipid transfer protein STARD7 controls intestinal tumor development in a context-dependent manner

Kateryna Shostak[1], Yu Chen[1], Chloé Maurizy [1], Gilles Rademaker[2], Xinyi Xu[1], Arnaud Blomme [3], Pierre Close[3,4], Olivier Renson [5], Matthias Van Hul[6], Patrice D Cani [4,6,7], Sebastian Klein[8,15], Alexandra Florin[8], Reinhard Büttner [8], Didier Cataldo [9], Philippe Delvenne[10], Ivan Nemazanyy [11], Caroline Wathieu[1], Alexandre Hego[12], Sandra Ormenese[12], Olivier Peulen [2], Marc Thiry [13], Roopesh Krishnankutty [14], Jair Marques Jr[14], Alex von Kriegsheim[14] & Alain Chariot [1,4✉]

## Abstract

The role of phosphatidylcholine transporters such as Stard7 in intestinal cancer development is unknown. To explore this issue, we generated a mouse model lacking Stard7 in intestinal epithelial cells (IECs). Loss of Stard7 impaired mitochondrial Complex I activity, led to a severe metabolic and lipid reprogramming, enhanced mitochondrial ROS production and potentiated an mTORC1/ATF4 signature. As a result, levels of enzymes involved in serine biosynthesis were enhanced in Stard7-deficient IECs. We next assessed the consequences of Stard7 deficiency in both Wnt-dependent tumor initiation and in inflammation-driven tumor development. Strikingly, despite generating similar molecular signatures, Stard7 deficiency inhibited tumor development in Azoxymethane (AOM)/Dextran Sulfate Sodium (DSS)-treated mice but promoted Wnt-driven cancer initiation in the intestine. Apc[+/Min] mice lacking Stard7 in IECs developed more tumors in the distal colon as well as a specific microbiota signature. Collectively, our results suggest that the genetic status critically controls the effects of Stard7 deficiency on intestinal tumor development.

**Keywords** Wnt-driven Tumor Initiation; Inflammation-driven Intestinal Cancers; STARD7; mTORC1
**Subject Categories** Cancer; Digestive System

## Introduction

Mitochondria are critically involved in cellular metabolism, stress responses and in the regulation between cell survival and cell death in multiple tissues. As such, they contribute to cell fate decisions, including in the intestinal epithelium (Rath et al, 2018). The single layer of intestinal epithelial cells (IECs), which renews every 3–5 days, includes intestinal stem cell (ISC) located at the bottom of each intestinal crypt as well as progenitors which undergo cell proliferation in the transit amplifying zone following Wnt signaling activation (Beumer and Clevers, 2021). While progressing to the top of each crypt, IEC progenitors undergo cell differentiation to generate the secretory or absorptive lineage (Beumer and Clevers, 2021). Because ISCs, progenitors and differentiated IECs show distinct status of Wnt signaling, a key pathway involved in the expression of multiple metabolic genes, they consequently harbor differences in metabolic pathways such as oxidative phosphorylation and glycolysis (Sethi and Vidal-Puig, 2010; Rodriguez-Colman et al, 2017).

Mitochondrial-derived mediators serve as signaling molecules and include reactive oxygen species (ROS), which are mainly produced by Complex I and III of the mitochondrial electron transport chain (Rath et al, 2018). Alterations of mitochondrial function have been described in inflammatory intestinal disorders as well as in cancer. Indeed, a mitochondrial unfolded protein response (UPR[MT]) has been observed in IECs from patients suffering from inflammatory bowel disease (IBD) and in mouse models of intestinal inflammation (Rath et al, 2012). Likewise, increased ROS levels resulting from mutation or altered amounts of mitochondrial DNA or from Rac1 GTPase activity potentiate Wnt-driven tumor initiation in the intestine (Woo et al, 2012; Smith et al, 2020; Myant et al, 2013). How these mitochondrial mediators

[1]Laboratory of Cancer Biology, GIGA Cancer, University of Liege, Sart-Tilman, Liège, Belgium. [2]Metastasis Research Laboratory, GIGA Cancer, University of Liege, Sart-Tilman, Liège, Belgium. [3]Laboratory of Cancer Signaling, GIGA Cancer, University of Liege, Sart-Tilman, Liège, Belgium. [4]WELBIO-Walloon Excellence in Life Sciences and Biotechnology, WEL Research Institute, WELBIO Department, Wavre, Belgium. [5]GIGA Genomics Platform, University of Liege, Liege, Belgium. [6]Metabolism and Nutrition Research group, Louvain Drug Research Institute (LDRI) and Institute of Experimental and Clinical Research (IREC), UCLouvain, Université catholique de Louvain, Brussels, Belgium. [7]Section of Biomolecular Medicine, Division of Systems Medicine, Department of Metabolism, Digestion and Reproduction, Imperial College London, London, United Kingdom. [8]Institute for Pathology-University Hospital of Cologne, Cologne, Germany. [9]Laboratory of Tumour and Development Biology, GIGA Cancer, University of Liege, Sart-Tilman, Liege, Belgium. [10]Department of Pathology, University Hospital of Liege (CHU), Sart-Tilman, Liege, Belgium. [11]Platform for Metabolic Analyses, Structure Fédérative de Recherche Necker, INSERM US24/CNRS UMS 3633, Paris, France. [12]GIGA Flow Cytometry and Cell Imaging Platform, University of Liege, Liege, Belgium. [13]Unit of Cell and Tissue Biology, GIGA Neurosciences, University of Liege, Liege, Belgium. [14]Institute of Genetics and Cancer, Edinburgh Cancer Research, Edinburgh, Scotland. [15]Present address: Department of Hematology and Stem Cell Transplantation, University Duisburg-Essen, University Hospital Essen, Essen, Germany. ✉E-mail: Alain.chariot@uliege.be

trigger both metabolic and transcriptional reprogramming to regulate intestinal tumor development and whether or not the genetic status of these tumors has any influence in these biological processes remain unclear.

Lipids are essential for cell membrane integrity and are synthesized in distinct cellular compartments. Most of them are produced within the endoplasmic reticulum (ER) and redistributed to other cellular membranes (Sprong et al, 2001; Wong et al, 2017). Intracellular lipid transport between organelles is mediated by vesicular transport through the fusions of vesicles from a donor to an acceptor compartment. This process can also occur via several cytosolic proteins in a monomeric manner between membranes of organelles (Prinz, 2010). Monomeric exchange involves specific cytoplasmic proteins with specific lipid-binding domains required to promote lipid exchanges (Holthuis and Levine, 2005). These proteins include the evolutionarily conserved members of the steroidogenic acute regulatory protein-related lipid transfer (START) domain family (Soccio and Breslow, 2003; Alpy and Tomasetto, 2005; Clark, 2012). The START domain contains a 210-amino acids sequence which folds into a α/β helix-grip hydrophobic pocket for binding to phospholipids, sterols or sphingolipids (Alpy and Tomasetto, 2005; Tsujishita and Hurley, 2000). In mammals, START domains are found in 15 members (Stard1-Stard15) which can be further classified into six subfamilies based on the sequence and ligand similarities (Soccio and Breslow, 2003; Alpy and Tomasetto, 2005). Phosphatidylcholines (PC), a class of phospholipids, are transported by Stard2/PC-TP, Stard7 and Stard10 (Kanno et al, 2007; Kang et al, 2010; Horibata et al, 2010; Olayioye et al, 2005). PC account for up to 45% of phospholipids found in both inner and outer membranes of mitochondria, which lacks the enzymatic activity to synthetize it (Tamura et al, 2014). Therefore, PC transfer proteins are believed to be critical for mitochondria homeostasis.

Stard7, also referred to as GTT1, was originally identified as a transcript more expressed in the choriocarcinoma cell line JEG-3 than in normal and benign trophoblastic tissues (Durand et al, 2004). Its mRNA sequence has two distinct in frame translation initiation AUG codons. This leads to the production of two Stard7 proteins, a short 35 kDa cytoplasmic polypeptide and a second product, Stard7-I, which has an additional 75 amino acids stretch with a mitochondrial localization signal at its N-terminal domain (Horibata et al, 2010). The mature form of Stard7 is generated by cleavage of this N-terminal mitochondrial-targeting sequence (Horibata et al, 2010). Stard7 deficiency is associated with altered mitochondrial size, decreased aerobic respiration, increased ROS production and mitochondrial DNA damage which, in turn, causes altered barrier integrity of epithelial cells in vitro and in vivo (Yang et al, 2017). As Stard7 is also expressed in a variety of cancer cell lines, it is believed to play a role in cell proliferation by providing PC for mitochondrial biogenesis (Durand et al, 2004). As a Wnt target gene in several cell types, Stard7 may actually act as a critical Wnt effector in tumor development (Rena et al, 2009; Lee et al, 2007). However, experimental data to support this hypothesis is currently lacking.

Several START members have been linked with cancer. Indeed, Stard10 is overexpressed in 35–40% of ErbB2 positive breast cancers and cooperates with ErbB receptor signaling (Olayioye et al, 2004). Similarly, the Stard3 gene, which codes for a cholesterol transfer protein, is part of the ErbB2 amplicon and is critical for the

growth and survival of ErbB2 positive breast cancer cells (Akiyama et al, 1997; Sahlberg et al, 2013; Wilhelm et al, 2017). Stard3, when fused to protein phosphatase 1 regulatory inhibitor subunit 1B gene (PPP1R1B), promotes gastric cancer development through AKT (Yun et al, 2014). Whether and how Stard7 regulates tumor development in vivo is unknown.

In this study, we report that mitochondrial stress due to the loss of Stard7 in IECs triggers mTORC1 activation in both healthy and Apc-mutated intestinal epithelia. Remarkably, Stard7 acts as a tumor-promoting protein in a mouse model of inflammation-driven tumor development but as a tumor suppressor gene in Wnt-driven tumor initiation. Despite opposite roles for Stard7 in both experimental models of intestinal cancers, Stard7-deficient IECs showing or not constitutive Wnt signaling undergo a similar mTORC1/Atf4-dependent transcriptional reprogramming characterized by elevated levels of serine biosynthesis regulatory enzymes. Importantly, the enhanced tumor development seen in Apc-mutated mice lacking Stard7 in IECs relies, at least in part, on a specific microbiota signature. Finally, the capacity of IECs to trigger a cellular response to oxidative stress critically depends on their Apc status. Collectively, our study highlights the key role of the Apc status in controlling the effects of mitochondrial stress on intestinal tumor development.

# Results

## Stard7 is dispensable in intestinal homeostasis

The role of lipid transfer proteins such as STARD7 in intestinal cancers is poorly understood. To experimentally explore this issue, we first established the expression profile of Stard7 in the mouse intestine. Stard7 is expressed in all parts of the intestine, although at higher levels in the colon (Fig. 1A). To gain insight into the role of Stard7 in the biology of the intestine, we generated a mouse model in which Stard7 was genetically inactivated in IECs by crossing Stard7$^{Lox/lox}$ mice with the Villin-CRE strain (Stard7$^{\Delta IEC}$) (Fig. 1B). The percentage of Ki67$^+$ cells did not change in crypts from the small intestine or from the colon lacking Stard7 (Stard7$^{Lox/lox}$ referred to as "WT" and Stard7$^{\Delta IEC}$ referred to as "KO") (Fig. EV1A,B, respectively). Moreover, the number of Mucin 2$^+$ cells, a goblet cell marker, was also similar in small intestines lacking or not Stard7 (Fig. EV1C). Moreover, the architecture as well as the number of Alcian blue$^+$ cells per crypt, which also reflects goblet cell differentiation, was also similar in the colon of both Stard7$^{Lox/lox}$ and Stard7$^{\Delta IEC}$ mice (Fig. EV1D). Stard7 was also dispensable in Paneth cell differentiation, as judged by similar number of Lysozyme$^+$ cells and protein levels of Lysozyme in the small intestine of both Stard7$^{Lox/lox}$ and Stard7$^{\Delta IEC}$ mice (Fig. EV1E). Moreover, mRNA levels of Wnt target genes (Lgr5, c-Myc, Sox9 and Cd44) remained unchanged in IECs lacking Stard7 (Fig. EV1F). This conclusion also applied to stem cells markers (Olfm4, Bmi1, Epha2) and to the tuft cell marker Dclk1 (Fig. EV1F).

As barrier integrity was damaged upon Stard7 deficiency in the lung (Yang et al, 2017), we also looked at proteins involved in tight junctions but did not find any defect in both localization and level of E-cadherin and Claudin 3 upon Stard7 deficiency in the small intestine and in the colon (Fig. EV2A,B). Finally, the body weight of 21 months old Stard7$^{Lox/lox}$ and Stard7$^{\Delta IEC}$ mice were comparable as

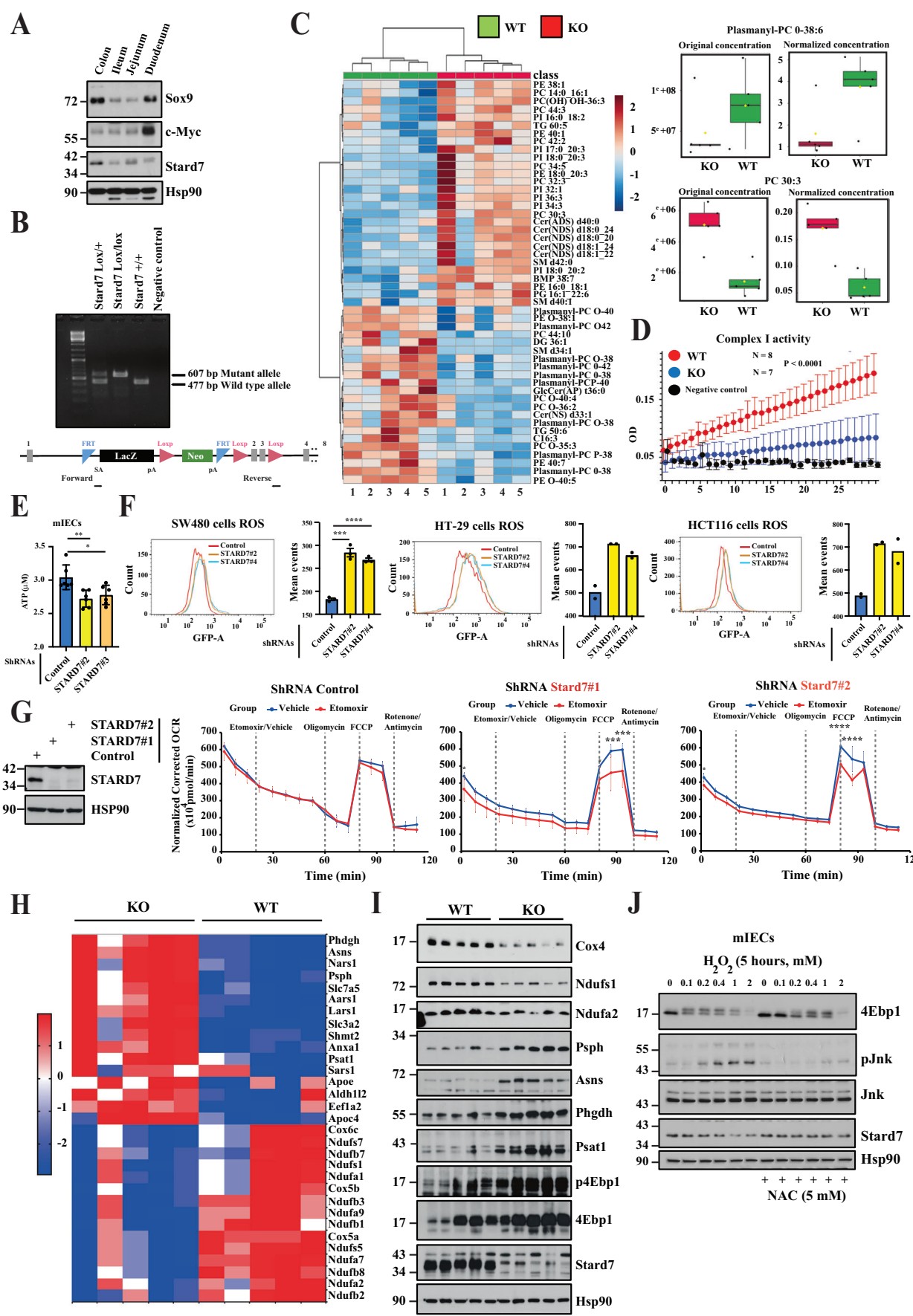

**Figure 1.   Stard7 deficiency in intestinal epithelial cells triggers a mitochondrial stress and potentiates an mTORC1/atf4 signature.**

(A) Expression profile of Stard7 in the mouse intestinal epithelium. Extracts from the indicated distinct parts of the intestine were subjected to western blot analyses. These analyses were carried out using extracts from 5 mice (2 males and 3 females, 8–12 weeks old). A representative blot is illustrated. (B) Generation and genotyping of a mouse model genetically inactivated for Stard7 in intestinal crypts. Loxp sites are flanking exons 2 and 3 of the Stard7 gene. The Stard7$^{Lox/lox}$ mouse was crossed with the Villin-CRE strain to generate Stard7$^{\Delta IEC}$ mice. PCR analyses were conducted with genomic DNAs using both forward and reverse primers to detect mutant and/or wild-type alleles. (C) Stard7 deficiency in intestinal epithelial cells show deregulated lipid levels. Extracts from the colon of mice of the indicated genotype (Stard7$^{Lox/lox}$ referred to as "WT" and Stard7$^{\Delta IEC}$ referred to as "KO", respectively, $n = 5$ for each genotype, 4 WT males and 1 WT female, 3 KO males and 2 KO females, 8–12 weeks old) were subjected to a lipidomic analysis. Stard7 deficiency leads to the accumulation of multiple phosphatidylcholine (PC), phosphatidylinositol (PI) and phosphatidyletha-nolamine (PE) derivatives as well as decreased levels of both plasmanyl and plasmenyl lipids. Heatmaps and box plot were created using the public database MetaboAnalyst https://www.metaboanalyst.ca/MetaboAnalyst/home.xhtml. Each box shows the 25th (Q1) to 75th (Q3) percentiles, with a line at the median (50th percentile) and whiskers extending to near the data's max/min (within 1.5x Interquartile Range (IQR)). (D) IECs lacking epithelial Stard7 show a defective Complex I activity. Complex I activity was measured with extracts from IECs of mice lacking or not Stard7 in intestinal epithelial cells (means ± S.D., two-way ANOVA, ****$P < 0.0001$, $n = 8$ and 7 for WT and KO mice, respectively, 6 WT males and 2 WT female, 5 KO males and 2 KO females, 8–12 weeks old). (E) Stard7 deficiency in mouse intestinal epithelial cells (mIECs) impairs ATP production. ATP production was measured with extracts from control and Stard7-depleted mIECs, as indicated. Data were from two independent experiments carried out in triplicates (means ± S.D., unpaired Student $T$ test with Welch's correction, *$P = 0.0213$, **$P = 0.0067$). Data in this figure as well as in following figures were generated with biological replicates. (F) STARD7 deficiency in colon cancer-derived cell lines enhances ROS levels. Extracts from control and STARD7-depleted SW480, HT-29 and HCT116 cells were used to quantify ROS levels by FACS analyses (see methods for details) (means ± S.D., Dunnett's multiple comparisons test, SW480 cells, ****$P < 0.0001$, $n = 3$). (G) Mouse intestinal epithelial cells lacking Stard7 rely on FAO as a source of energy. Basal, maximal and spare oxygen consumption rate (OCR - pmol/min) measurements were carried out with control and Stard7-depleted mIECs treated or not with Etomoxir as indicated (see "Methods" for details). All results were normalized according to cell numbers. Results shown are from three independent experiments carried out with triplicates. Two-sided statistical analysis was performed using one-way analysis of variance followed by Tukey's multiple comparisons (***$P < 0.001$, ****$P < 0.0001$). Western blot analyses were carried out with protein extracts to assess STARD7 levels in all experimental conditions (left panels). Four independent experiments were carried out and a representative blot is illustrated. (H, I) Defective expression of Ndu proteins but enrichment of candidates downstream of the mTORC1/Atf4 signaling cascade in IECs lacking Stard7. A HeatMap generated with proteomic data and western blot analyses are illustrated (F, G, respectively). Protein extracts were generated from IECs of Stard7$^{Lox/lox}$ and Stard7$^{\Delta IEC}$ mice (5 WT males, 3 KO males and 2 KO females, 8–12 weeks old). p4Ebp1 phosphorylation was assessed on Threonine 70 (G). (J) $H_2O_2$ triggers mTORC1 activation in a ROS-dependent manner in IECs. mIECs were pre-incubated or not with NAC for 1 hour and subsequently left untreated or stimulated with $H_2O_2$ for 5 hours at the indicated concentrations. Extracts were subjected to western blot analyses using the indicated antibodies. Three independent experiments were carried out and representative blots are illustrated. Source data are available online for this figure.

were mRNA levels of all tested pro-inflammatory cytokines, which rules out the possibility that Stard7 deficiency leads to any spontaneous inflammatory disorders (Fig. EV2C,D, respectively). Collectively, our results demonstrate that Stard7 in dispensable in intestinal homeostasis.

As Stard7 promotes PC transfer from ER to mitochondria, we hypothesized that Stard7 deficiency may trigger some mitochondrial stress. We first assessed mitochondrial morphology in intestinal tissues from Stard7$^{Lox/lox}$ and Stard7$^{\Delta IEC}$ mice through transmission electronic microscopy. We noticed that the number of mitochondria per µm$^2$ of cytoplasm significantly decreased in cells lacking Stard7 (Fig. EV3A). In contrast, the mean mitochondrial area increased upon Stard7 deficiency (Fig. EV3A). As a result, the total mitochondrial area in the cytoplasm did not significantly change in cells lacking Stard7 (Fig. EV3A). We next conducted lipidomic analyses with extracts from the colon of 8–12 weeks old mice lacking or not Stard7 in IECs. We noticed that levels of multiple phosphatidylcholine (PC), phosphatidylinositol (PI) and phosphatidylethanolamine (PE) derivatives increased in IECs lacking Stard7, which presumably results from a defective PC transport (Fig. 1C). Importantly, levels of both plasmanyl and plasmenyl lipids decreased in IECs lacking Stard7 (Fig. 1C). To more precisely define the consequences of Stard7 deficiency on the lipid composition of distinct subcellular compartments, we repeated these lipidomic analyses using ER, mitochondrial, cytoplasmic and lysosomal extracts from the intestine of both WT and KO mice. We first conducted western blot analyses with our organelle-specific extracts to validate their quality. As expected, Stard7 was found in mitochondria, in the cytoplasm and also in mitochondria-associated membranes (MAMs) while the Voltage-dependent anion channel 1 (Vdac1) was exclusively detected in both mitochondria and in MAMs (Fig. EV3B). In agreement with

its function as a subunit of the Cytochrome c oxidase (Complex IV), Cox4 was found in the mitochondria while Rab11, a marker of recycling endosomes, was detected in ER fractions as well as in the lysosomal and plasma membrane fractions (Fig. EV3B). As expected, Hsp90 was detected in the cytoplasm while both Perk and Ip3r were found in the ER, in MAMs as well as in the in the lysosomal and plasma membrane fractions (Fig. EV3B). In agreement with a role of Stard7 in PC transport from the ER to mitochondria, multiple PC derivatives significantly accumulated in the ER of IECs lacking Stard7 (Fig. EV3B). This conclusion did not apply to all PC derivatives as levels of 3 of them (i.e. PC 35:1, PC O-42:3 and PC 40:0) were higher in the mitochondrial extracts from the intestine of Stard7 KO mice (Fig. EV3B). Levels of multiple PC derivatives also significantly increased in lysosomal/plasma membrane but not in cytoplasmic extracts upon Stard7 deficiency (Fig. EV3B). Strikingly, the distribution of other candidates such as Ceramides and Plasmanyl lipids also changed in both ER and mitochondria extracts from the intestine of KO versus WT mice (Fig. EV3B). This lipid redistribution may, at least in part, results from enhanced mitochondria-associated membranes contact (MAMs) seen in mIECs lacking Stard7 (Fig. EV3C), as previously demonstrated in breast cancer cells (Dondajewska et al, 2025). Therefore, Stard7 deficiency has profound consequences on the landscape of lipid composition in multiple cellular organelles.

To assess the consequences of Stard7 deficiency on mitochondrial functions, we assessed Complex I activity in IECs from both Stard7$^{Lox/lox}$ and Stard7$^{\Delta IEC}$ mice. The loss of Stard7 in IECs impaired Complex I activity, which demonstrates that Stard7 deficiency triggers a mitochondrial stress (Fig. 1D). Likewise, Stard7 deficiency impaired ATP production in mouse immortalized intestinal epithelial cells (mIECs) lacking Stard7 and enhanced reactive oxygen species (ROS) production in all tested colon

cancer-derived cell lines, presumably because of this mitochondrial stress (Fig. 1E,F, respectively). Interestingly, mIECs lacking Stard7 critically rely on fatty acid oxidation (FAO) as the maximal oxygen consumption rate (OCR) dropped in Stard7-depleted but not in control cells when treated with Etomoxir, a FAO inhibitor (Fig. 1G). To explore the molecular consequences of Stard7 deficiency in the intestine, we next conducted proteomic analyses with extracts from the colon of both Stard7$^{Lox/lox}$ and Stard7$^{\Delta IEC}$ mice to define protein candidates whose expression is deregulated in mice lacking Stard7 in IECs. Multiple candidates involved in oxidative phosphorylation, including multiple NADH dehydrogenase subunits (Ndufs), all acting in Complex I, as well as Cox4, Cox5a/b and Cox6c were not properly expressed in IECs from Stard7$^{\Delta IEC}$ mice (Fig. 1H,I). On the other hand, multiple enzymes involved in serine biosynthesis, namely Psph and Phgdh as well as Activating Transcription Factor 4 (Atf4) target genes such as Asns, Psat1, Nars and Sars were more expressed in IECs from Stard7$^{\Delta IEC}$ mice (Fig. 1H,I). Atf4 is an mTORC1 effector and a specific transcriptional signature can be defined downstream of this pathway (Torrence et al, 2021). Consistently, 4Ebp1 phosphorylation on Thr 70, a hallmark of mTORC1 activity, was enhanced upon Stard7 deficiency (Fig. 1I). To better link ROS production to enhanced mTORC1 activation seen upon Stard7 deficiency, we treated or not mIECs with $H_2O_2$. 4Ebp1 phosphorylation was enhanced by $H_2O_2$ in these cells as was the phosphorylation of the pro-apoptotic kinase Jnk (Fig. 1J). Interestingly, the anti-oxidant N-Acetyl Cysteine (NAC) attenuated $H_2O_2$-dependent 4Ebp1 phosphorylation, which further supports the conclusion that ROS promotes mTORC1 activation in mIECs (Fig. 1J). Collectively, our data suggest that Stard7 deficiency causes mitochondrial stress and potentiates an mTORC1/Atf4 signaling cascade that does nevertheless not influence intestinal homeostasis.

## STARD7 is overexpressed in colon cancer and supports Wnt signaling

To assess the potential role of STARD7 in intestinal tumor development, we first explored its expression profile in human intestinal cancers. STARD7 expression is enhanced in clinical cases of intestinal tumors when compared to normal adjacent tissues. Indeed, a morphometric quantification carried out on immunohistochemistry data from human normal intestines as well as dysplasia and carcinomas cases also highlighted STARD7 overexpression in colon carcinomas (Fig. EV4A). Interestingly, high STARD7 mRNA levels were associated with lower survival rates in patients suffering from intestinal cancers (Fig. EV4B). Therefore, these data prompted us to address the potential role of STARD7 in intestinal tumor development. We first characterized several colon cancer-derived cell lines in which STARD7 was depleted. SOX9 and c-MYC, two Wnt target genes, were downregulated at mRNA and protein levels in all tested colon cancer-derived cell lines lacking STARD7, suggesting that STARD7 acts upstream of Wnt signaling (Fig. EV4C,D, respectively). Importantly, while β-Catenin deficiency led to a decrease in both SOX9 and c-MYC levels, it did not alter protein levels of STARD7 in a variety of colon cancer-derived cell lines, except in Colo205 cells (Fig. EV4E,F). Moreover, ICRT3, which inhibits β-Catenin/TCF interaction and their transcriptional activity, decreased both c-MYC and SOX9 but not STARD7 protein levels in the tested colon cancer cell lines, except in SW480 cells (Fig. EV4G). Tamoxifen-inducible β-Catenin$^{c.a.}$ mice, which express

a truncated and stabilized form of β-Catenin in IECs, show constitutive Wnt signaling and develop adenomas upon Tamoxifen administration (Harada et al, 1999). As expected, Sox9 protein levels were severely increased in these mice, in contrast to Stard7 levels (Fig. EV4H). Therefore, STARD7 expression is not regulated by Wnt signaling in colon cancer-derived cell lines as well as in mouse IECs but promotes the expression of Wnt effectors in established colon cancer cell lines.

## Stard7 contributes to inflammation-driven tumor development in the intestine

A recently published paper demonstrated that Stard7$^{+/-}$ mice were more sensitive to experimental colitis induced by Dextran Disulfate Sodium (DSS), at least because of impaired intestinal barrier integrity (Uddin et al, 2024). To assess whether this conclusion also applies to mice in which Stard7 is specifically and only inactivated in IECs, we subjected both Stard7$^{Lox/lox}$ and Stard7$^{\Delta IEC}$ mice to DSS to trigger colitis. As expected, DSS severely decreased E-cadherin protein levels due to damage in the single layer of IECs and this decrease appeared slightly more pronounced in DSS-treated Stard7$^{\Delta IEC}$ mice (Fig. EV5A). Moreover, DSS triggered apoptosis in both genotypes, as evidenced by elevated levels of activated Caspase 3 (Fig. EV5A). We did not find any statistically significant decrease in the colon length of DSS-treated Stard7$^{\Delta IEC}$ versus Stard7$^{Lox/lox}$ mice, even if mRNA levels of TNF but not IL-1β were slightly increased upon Stard7 deficiency (Fig. EV5B,C, respectively). Therefore, Stard7 expression in IECs does not protect from DSS-induced colitis.

As it is currently unknown whether Stard7 is involved in intestinal tumor development, we next treated both Stard7$^{lox/lox}$ and Stard7$^{\Delta IEC}$ mice with a combination of Azoxymethane (AOM) and Dextran Disulfate Sodium (DSS) to trigger inflammatory bowel diseases (IBD)-related colon tumorigenesis (Fig. 2A). The number of tumors in the colon severely decreased in mice lacking Stard7 in IECs (Fig. 2A). Moreover, Stard7 deficiency in IECs decreased the number of both low and high grade tumors in contrast to the increased area of the normal epithelium in analyzed sections (Fig. 2B). Representative sections used for these histological analyses are illustrated in Fig. EV6. To explore molecular mechanisms through which Stard7 promotes AOM/DSS-dependent tumor development, we conducted unbiased transcriptomic analyses combined with Gene Set Enrichment Analyses (GSEA). While both Unfolded Protein Response (as evidenced by elevated Atf4 mRNA levels for example) and mTORC1-dependent signaling pathways were potentiated upon Stard7 deficiency, the oxidative phosphorylation (OXPHOS) as well as the cellular response to oxidative stress were impaired in IECs lacking Stard7 (Fig. 3A). Likewise, 4Ebp1 phosphorylation on Thr 70 was also enhanced without Stard7 (Fig. 3B,C), at least because 4Ebp1 protein levels were dramatically increased in both the small intestine and colon of Stard7$^{\Delta IEC}$ mice treated with AOM/DSS (Fig. 3B,C, respectively). Consistent with a defective cellular response to oxidative stress, protein levels of the anti-oxidant protein Tigar were decreased in the colon of Stard7$^{\Delta IEC}$ mice treated with AOM/DSS (Fig. 3C) (Bensaad et al, 2006). Likewise, levels of pH2AX, a DNA damage marker, were enhanced upon Stard7 deficiency (Fig. 3C). Yet, cleaved Caspase 3 levels were similar in the colon of both AOM/DSS-treated genotypes, indicating that a Caspase 3-dependent cell death pathway was not triggered upon Stard7 deficiency in vivo (Fig. 3C). In agreement with our transcriptomic

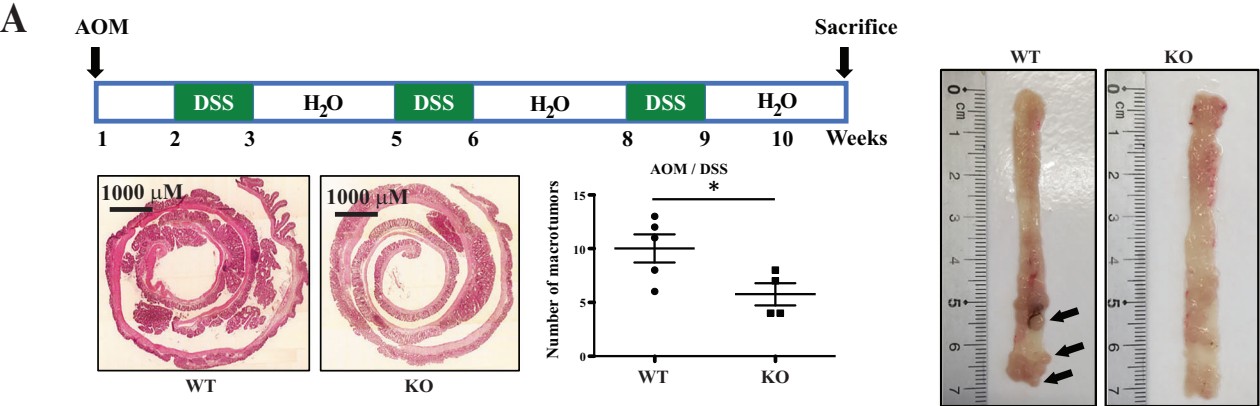

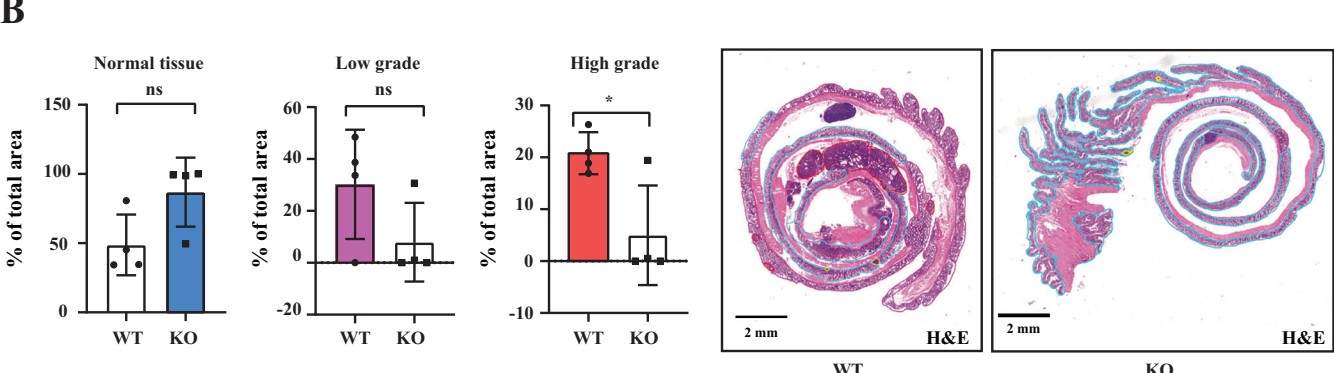

**Figure 2.    STARD7 promotes inflammation-driven tumorigenesis in the intestine.**

(A) Illustration of colons from both Stard7$^{lox/lox}$ and Stard7$^{\Delta IEC}$ mice, following AOM/DSS administration. In the middle, quantification of tumor numbers in the colon of both genotypes ($n = 5$ and 4 for Stard7$^{lox/lox}$ and Stard7$^{\Delta IEC}$ mice, respectively (means ± S.D., $T$ test with Welch's correction, *$P = 0.0361$)). Mice (males only) were 9 weeks old at the beginning of the experiment. (B) Decreased numbers of both low and high grade tumors upon Stard7 deficiency in AOM/DSS-treated mice ($n = 4$ for both Stard7$^{lox/lox}$ and Stard7$^{\Delta IEC}$ mice, means ± S.D., T test with Welch's correction, normal tissue: ns $= 0.0623$, low grade: ns $= 0.1417$, high grade: *$P = 0.0384$, ns $=$ not significant). Source data are available online for this figure.

analyses, multiple Atf4 target genes such as enzymes involved in serine biosynthesis, namely Psph and Phgdh as well as Asns were more expressed in IECs lacking Stard7 (Fig. 3D). Interestingly, colon tumors from Stard7$^{\Delta IEC}$ mice treated with AOM/DSS showed lower protein levels of Cyclin D1, c-Myc and Sox9 upon Stard7 loss, suggesting impaired expression of Wnt effectors in this model of inflammation-driven cancer (Fig. 3E). Importantly, mRNA levels of IL-17, IL-6 but not TNF were decreased in IECs from Stard7$^{\Delta IEC}$ mice treated with AOM/DSS, indicating that tumor development upon chronic inflammation in the intestine was impaired upon Stard7 deficiency in IECs at least through the decreased expression of some pro-inflammatory cytokines (Fig. 3F). Collectively, our results suggest that Stard7 is acting as tumor-promoting protein in a model of inflammation-driven intestinal cancer, at least by promoting the expression of Wnt-induced candidates and some pro-inflammatory cytokines.

## Stard7 is a tumor suppressor gene in Wnt-driven tumor initiation

Stard7 is promoting the expression of Wnt effectors in established colon cancer cell lines showing constitutive Wnt signaling but it is

nevertheless unclear whether Stard7 is required in Wnt-driven tumor initiation. To assess this issue, we crossed Stard7$^{Lox/lox}$ or Stard7$^{\Delta IEC}$ mice with the Apc$^{+/Min}$ strain which spontaneously develops intestinal adenomas due to constitutive Wnt signaling (Su et al, 1992). Unexpectedly, both male and female Apc$^{+/Min}$/Stard7$^{\Delta IEC}$ mice showed a more pronounced weight loss when compared to Apc$^{+/Min}$/Stard7$^{Lox/lox}$ mice (Fig. 4A,B). As a result, the median survival of Apc$^{+/Min}$/Stard7$^{\Delta IEC}$ mice was shorter than Apc$^{+/Min}$/Stard7$^{Lox/lox}$ mice (24.12 versus 32.86 weeks), due to a dramatic increase in tumor burdens in the distal colon (Fig. 4C–E, respectively). Apc$^{+/Min}$/Stard7$^{\Delta IEC}$ mice actually showed a higher number of both polyps and flat adenomas in the colon (Fig. 4F). A detailed histological analysis indicated that areas of healthy tissues in the colon decreased in Apc$^{+/Min}$/Stard7$^{\Delta IEC}$ mice (Fig. 4G). Consistently, the number of tumors in the colon of Apc$^{+/Min}$/Stard7$^{\Delta IEC}$ mice increased no matter what their grade was (lower, mild or high grades) (Fig. 4G).

To explore the underlying mechanisms through which Stard7 negatively regulates intestinal tumor development upon constitutive Wnt signaling, we hypothesized that the loss of function of the Apc tumor suppressor gene, which triggers chromosomal instability

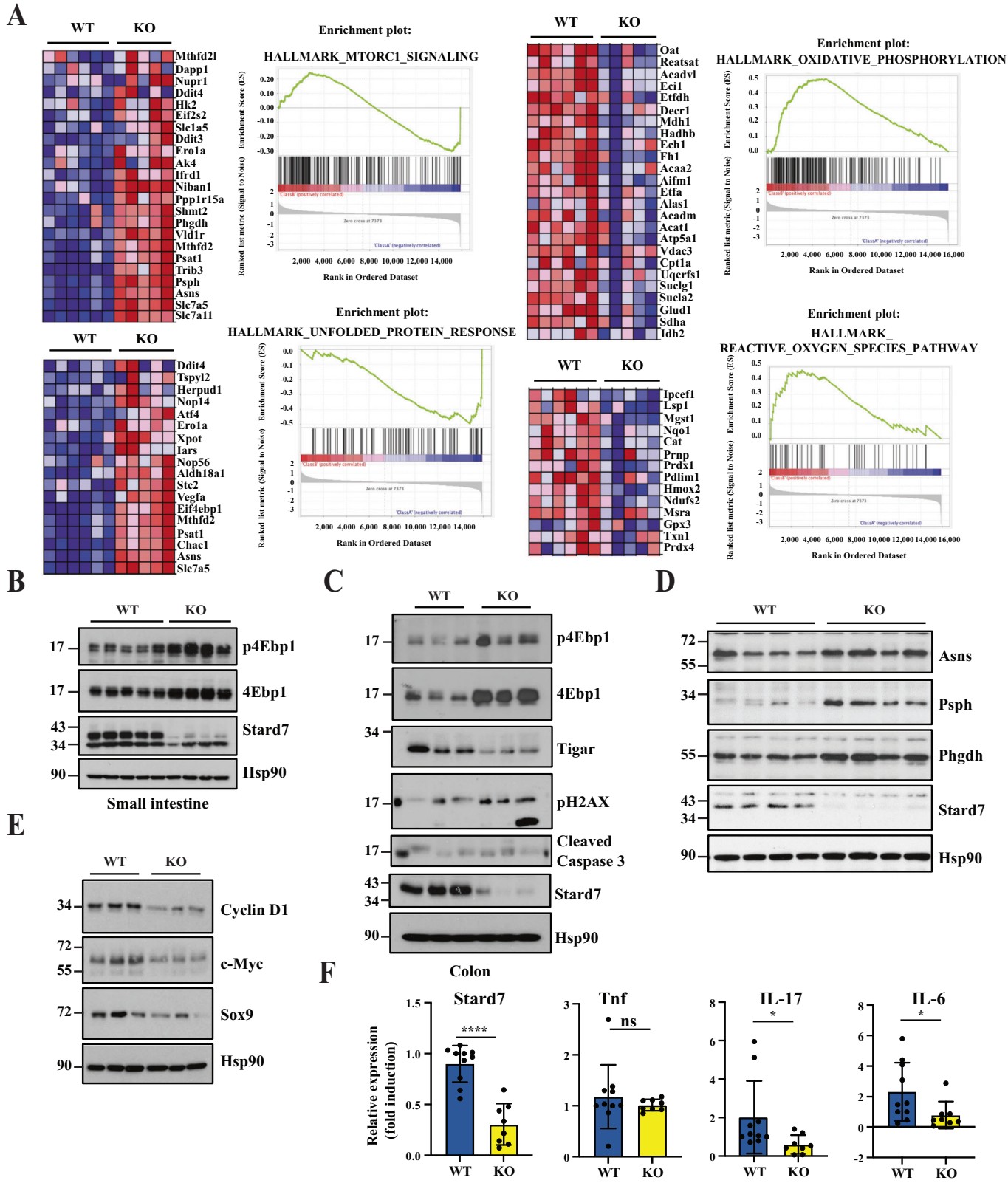

◄

**Figure 3. Stard7 acts as an oncogene in a model of inflammation-driven tumor development through mTORC1 inhibition.**

(A) Defective signaling pathways upon Stard7 deficiency identified through GSEA analyses. Genes with the highest enrichment score are shown as individual gene expression-based heat maps. HeatMaps demonstrate upregulation of both mTORC1 and UPR signaling pathways (candidates in red) as well as downregulation of oxidative phosphorylation as well as cellular response to ROS species (candidates in blue) in IECs lacking Stard7. The color code in HeatMaps corresponds to the color code illustrated in enrichment plots. Mice (males only) were 9 weeks old at the beginning of the experiment. (B, C) Stard7 negatively regulates 4Ebp1 protein levels in both small intestine and colon of mice treated with AOM/DSS. Total extracts were generated with IECs from small intestines (B) and colon (C) of mice of the indicated genotypes, after the AOM/DSS treatment. p4Ebp1 phosphorylation was assessed on Threonine 70. (D) Increased levels of candidates induced by Atf4 upon epithelial Stard7 deficiency. Total extracts from colon of AOM/DSS-treated mice of the indicated genotypes were subjected to western blot analyses. (E) Stard7 promotes the expression of Wnt effectors in inflammation-driven tumorigenesis. Total SDS extracts from colon tumors of the indicated mouse genotypes treated with AOM/DSS were subjected to western blot analyses. (F) Stard7 deficiency in IECs decreases mRNA levels of multiple pro-inflammatory cytokines upon AOM/DSS treatment. mRNA levels of the indicated candidates in IECs of AOM/DSS-treated mice of the indicated genotypes (Stard7$^{Lox/lox}$ (WT) and Stard7$^{\Delta IEC}$ mice (KO)) were quantified by quantitative Real-Time PCR analyses. mRNA levels of these candidates in one randomly selected Stard7$^{Lox/lox}$ mice were set to 1 and levels in other mice were relative to that after normalization with Gapdh mRNA levels ($n = 10$ for Stard7$^{Lox/lox}$ mice and $n = 8$ for Stard7$^{\Delta IEC}$ mice (means ± S.D., $T$ test with Welch correction, Stard7: \*\*\*$P < 0.001$, IL-17: \*$P = 0.0446$, IL-6: \*$P = 0.0418$, Tnf: ns = not significant). Source data are available online for this figure.

(Fodde et al, 2001), combined with the enhanced production of ROS due to *Stard7* deficiency, would enhance the rate of mutations in IECs. To assess this issue, we conducted exome sequencing experiments with intestinal tumors from both genotypes and noticed that the mutational rate was more elevated in tumors from Apc$^{+/Min}$/Stard7$^{\Delta IEC}$ mice (Fig. 5A). As expected, the number of total single nucleotide polymorphisms (SNPs), including in coding sequences, was more elevated in Apc$^{+/Min}$ compared to intestinal extracts from healthy mice, which reflects the genomic instability resulting from Apc loss (Fig. 5B) (Fodde et al, 2001). Importantly, Stard7 deficiency in IECs of Apc$^{+/Min}$ mice increased the number of SNPs, including in coding sequences (Fig. 5B). T > C mutations were statistically more frequent in IECs from Apc$^{+/Min}$/Stard7$^{\Delta IEC}$ (Fig. 5C). Therefore, the higher number of tumors found in the intestine of Apc$^{+/Min}$/Stard7$^{\Delta IEC}$ results, at least in part, from an increased mutational rate.

We next looked at mRNA levels of pro-inflammatory cytokines known to promote colon cancer development, namely, TNF, IL-1β, IL-6 and IL-17A (Wang et al, 2014; Dmitrieva-Posocco et al, 2019; Grivennikov et al, 2009; Greten et al, 2004). As expected, Stard7 mRNA levels dramatically decreased in Apc$^{+/Min}$/Stard7$^{\Delta IEC}$ mice (Fig. 5D). Interestingly, TNF, IL-17A and IL-1β but not IL-6 mRNA levels increased in IECs of Apc$^{+/Min}$/Stard7$^{\Delta IEC}$ mice, indicating that enhanced tumor development seen in Apc$^{+/Min}$ mice lacking Stard7 in IECs results, at least in part, to increased levels of multiple pro-inflammatory cytokines (Fig. 5D). Therefore, Stard7 is acting as a tumor suppressor gene in Wnt-driven tumor initiation in the intestine.

## Stard7 deficiency in Apc$^{+/Min}$ mice enhances mTORC1 activation and leads to a transcriptional reprogramming

To learn on molecular mechanisms through which StarD7 negatively regulates Wnt-driven tumor development in the intestine, we first looked at the activation of key oncogenic pathways. Although Stard7 plays opposite role in inflammatory-driven versus Wnt-dependent tumor development in the intestine, mTORC1 activation assessed through 4Ebp1 phosphorylation was elevated upon Stard7 deficiency in the colon of Apc$^{+/Min}$ mice, as previously observed in the AOM/DSS model (Fig. 5E). Of note, Akt activation was not regulated by Stard7 in Apc$^{+/Min}$ mice (Fig. 5E). Therefore, Stard7 is a negative regulator of mTORC1 activation in both mouse models of inflammatory-driven versus Wnt-dependent

tumor development in the intestine. We next looked at upstream kinases known to regulate mTORC1 activity and found that AMPK, a negative regulator of mTORC1 (Gwinn et al, 2008), was less phosphorylated in the colon from Apc$^{+/Min}$/Stard7$^{\Delta IEC}$ mice (Fig. 5F). Of note, Lkb, an AMPK kinase (Woods et al, 2003), was not dramatically less phosphorylated upon Stard7 deficiency in Apc$^{+/Min}$ mice (Fig. 5F). Therefore, mTORC1 activation is enhanced in the colon of Apc$^{+/Min}$ mice lacking Stard7 due to a defective Ampk signaling axis. Interestingly, Gpx4 levels were dramatically enhanced in the colon from Apc$^{+/Min}$/Stard7$^{\Delta IEC}$ mice. Likewise, levels of 4-HNE, a toxic product of lipid peroxidation defined as a reliable marker of ferroptosis, were also elevated in tissues from Apc$^{+/Min}$/Stard7$^{\Delta IEC}$ mice, as judged by immunohistochemistry analyses (Fig. 5G). We found that 4-HNE positive cells were more detected in tumors rather than in adjacent normal tissues in these mice (Fig. 5G). As a result, the higher percentage of 4-HNE positive cells found in tissues from Apc$^{+/Min}$/Stard7$^{\Delta IEC}$ mice resulted from enhanced tumor development. Therefore, Stard7 deficiency in IECs leads to ferroptosis upon constitutive Wnt signaling, presumably due to impaired Coenzyme Q transport (Fig. 5F) (Deshwal et al, 2023).

To learn more on the consequences of Stard7 deficiency in Wnt-driven tumor initiation on transcriptional reprogramming, we conducted RNA Sequencing experiments combined with GSEA with intestinal tumors from both Apc$^{+/Min}$/Stard7$^{\Delta IEC}$ and Apc$^{+/Min}$/Stard7$^{Lox/lox}$ mice. mRNA levels of candidates linked to mTORC1 activation such as UPR effectors were more expressed in tumors from Apc$^{+/Min}$/Stard7$^{\Delta IEC}$ mice (Fig. 6A). We confirmed through quantitative Real-Time PCR analyses that mRNA levels of Asparagine Synthetase (Asns), Asparaginyl-TRNA Synthetase 1 (Nars), Phosphoserine Synthetase (Psph), Phosphoserine Amino-transferase 1 (Psat1), Phosphoglycerate Dehydrogenase (Phgdh) and Methylenetetrahydrofolate Dehydrogenase (NADP$^+$ Dependent) 2 (Mthfd2), which are all target genes of the UPR effector Atf4 (Siu et al, 2002; Harding et al, 2003), were increased in tumors from Apc$^{+/Min}$/Stard7$^{\Delta IEC}$ mice (Fig. 6B). Interestingly, the expression of candidates involved in oxidative phosphorylation was impaired upon Stard7 deficiency in Apc$^{+/Min}$ mice, similarly to what we saw in the AOM/DSS model (Fig. 6A). Of note, Gpx4 mRNA levels were downregulated in tissues from Apc$^{+/Min}$/Stard7$^{\Delta IEC}$, which is in contrast to elevated Gpx4 protein levels found in these tissues, which possibly reflects the involvement of post-translational modifications of Gpx4 occurring upon Stard7

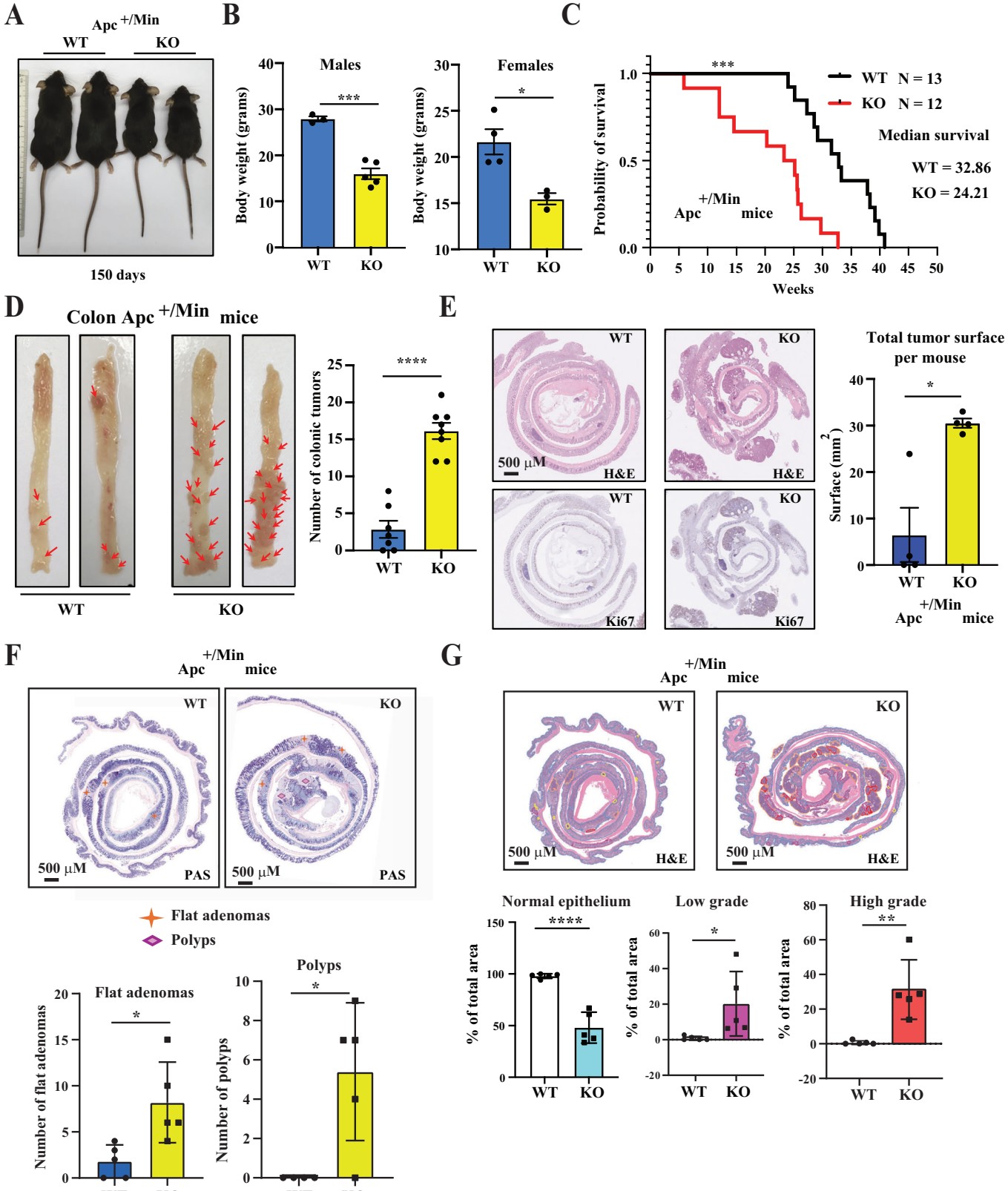

◄ **Figure 4. Stard7 limits Wnt-driven tumor initiation in the intestine.**

(A, B) Lack of Stard7 in intestinal epithelial cells enhances weight loss in mice showing constitutive Wnt signalling. Illustration of 150 days old Apc$^{+/Min}$/Stard7$^{Lox/lox}$ ("WT") and Apc$^{+/Min}$/Stard7$^{ΔIEC}$ ("KO") mice (A). The body weight of 150 days old Apc$^{+/Min}$/Stard7$^{Lox/lox}$ (3 males and 4 females) and Apc$^{+/Min}$/Stard7$^{ΔIEC}$ mice (5 males and 3 females) was quantified (means ± S.D., $T$ test with Welch correction, $n = 4$ mice, ***$P = 0.0002$, *$P = 0.0138$) (B). (C) Mice lacking Stard7 in IECs show a shorter survival upon constitutive Wnt signalling. A Kaplan–Meier survival graph for both Apc$^{+/Min}$/Stard7$^{Lox/lox}$ (WT) ($n = 13$ mice, 5 males and 8 females) and Apc$^{+/Min}$/Stard7$^{ΔIEC}$ mice (KO) ($n = 12$, 5 males and 7 females) was established (log-rank test, ***$P = 0.0003$). (D, E) Lack of Stard7 in IECs enhances tumor formation in the distal colon of Apc$^{+/Min}$ mice. Illustration of the colon of 150 days old Apc$^{+/Min}$/Stard7$^{Lox/lox}$ (WT, 3 males and 4 females) and Apc$^{+/Min}$/Stard7$^{ΔIEC}$ (KO, 4 males and 4 females) mice (D). The number of colonic tumors was quantified in 150 days old mice of the indicated genotypes (means ± S.D., Student $T$ test with Welsh correction, $n = 7$ mice, ****$P < 0.0001$) (D). IHC analyses are shown to visualize colonic tumors in both genotypes (E). The total tumor surface was also quantified in both genotypes (means ± S.D., $n = 4$ mice, 1 male and 3 females for WT and 2 males and 2 females for KO mice), Student $T$ test with Welch correction, *$P = 0.0241$) (E). (F) Stard7 deficiency in IECs of Apc$^{+/Min}$ mice enhances the number of tumors in the distal colon. On the top, representative images of the intestine in both WT and KO mice (Apc$^{+/Min}$/Stard7$^{Lox/lox}$ and Apc$^{+/Min}$/Stard7$^{ΔIEC}$ mice, respectively). PAS Periodic Acid-Schiff. At the bottom, quantification of both polyps and flat adenomas in the colon of both genotypes (orange stars and purple diamonds, respectively) (means ± S.D., $n = 5$ mice for both genotypes (3 males and 2 females for WT mice, 4 males and 1 female for KO mice), Flat adenomas: *$P = 0.0272$, Polyps: *$P = 0.0262$). (G) Elevated number of tumors of all grades in the colon upon Stard7 deficiency in IECs. At the bottom, quantification of the indicated grades of tumors in the colon of both genotypes (Apc$^{+/Min}$/Stard7$^{Lox/lox}$ (WT) and Apc$^{+/Min}$/Stard7$^{ΔIEC}$ (KO) mice). On the top, representative images of the intestine in both WT and KO mice (means ± S.D., $n = 5$ mice for both genotypes, Student $T$ test with Welch correction, Normal epithelium: ****$P < 0.0001$, Low grade: *$P = 0.0437$, High grade: **$P = 0.0040$). Source data are available online for this figure.

deficiency (Fig. 5F). Importantly, the AMPK target Acetyl-CoA carboxylase (ACC), which catalyzes the rate-limiting reaction in the biosynthesis of long chain fatty acids (Wakil et al, 1983), was not properly phosphorylated and was therefore more active in Apc$^{+/Min}$ mice lacking Stard7 in IECs (Fig. 6C). Moreover, candidates involved in the response to ROS were also dysregulated in tumors from Apc$^{+/Min}$/Stard7$^{ΔIEC}$ mice (Fig. 6A). In this context, levels of Thioredoxin-interacting protein (Txnip), which binds and inhibits the anti-oxidant function of Thioredoxin and triggers cell death upon oxidative stress (Nishiyama et al, 1999; Lu and Holmgren, 2012), were decreased in Apc$^{+/Min}$ mice lacking Stard7 in IECs while Thioredoxin Reductase 1 (TrxR1) was properly expressed (Fig. 6C). Of note, CTP:phosphocholine cytidylyltransferase α (CCTα), the rate-determining enzyme in de novo synthesis of PC, was less expressed in IECs from Apc$^{+/Min}$/Stard7$^{ΔIEC}$ mice, possibly as a result of PC accumulation in the ER due to a defective transfer to mitochondria (Fig. 6C).

We also conducted proteomic analyses with extracts from the colon of both 100 days old Apc$^{+/Min}$/Stard7$^{Lox/lox}$ and Apc$^{+/Min}$/Stard7$^{ΔIEC}$ mice to define protein candidates whose expression is deregulated in Apc$^{+/Min}$ mice lacking Stard7 in IECs. Similarly to our RNA Sequencing data, multiple candidates involved in oxidative phosphorylation, including multiple Ndufs, all acting in Complex I, as well as Cox4, Cox5a, Cox6c and Mrpl47 were not properly expressed in IECs from Apc$^{+/Min}$/Stard7$^{ΔIEC}$ mice (Fig. 6D,E). Likewise, the loss of Stard7 in IECs impaired Complex I activity in Apc$^{+/Min}$ mice, similarly to what we had seen in healthy mice (Figs. 6F and 1D, respectively). On the other hand and in agreement with our transcriptomic analyses, multiple enzymes involved in serine biosynthesis, namely Psph and Phgdh as well as Atf4 target genes such as Asns, Psat1, Nars and Sars were more expressed in IECs from Apc$^{+/Min}$/Stard7$^{ΔIEC}$ mice (Fig. 6D,E). Consistently, our metabolomic analyses carried out with extracts from the colon of 100 days old Apc$^{+/Min}$/Stard7$^{Lox/lox}$ and Apc$^{+/Min}$/Stard7$^{ΔIEC}$ mice showed that serine levels were higher in IECs lacking Stard7 (Fig. EV7). This conclusion actually applied to multiple other amino acids, indicating that Stard7 deficiency potentiates a one-carbon metabolism signature (Fig. EV7). Therefore, the loss of epithelial Stard7 in Apc-mutated intestines enhances tumor development, at least by potentiating serine biosynthesis/one carbon metabolism through an mTORC1/Atf4-dependent signaling pathway. Interestingly, intestinal extracts from

Apc$^{+/Min}$/Stard7$^{ΔIEC}$ mice also showed enhanced levels of Tricarboxylic Acid (TCA) cycle intermediates such as Lactate, Fumarate, Malate and Citrate, thus indicating that this metabolic pathway is deregulated upon Stard7 deficiency (Fig. EV7). On the other hand, levels of multiple fatty acids decreased in IECs from Apc$^{+/Min}$/Stard7$^{ΔIEC}$ mice (Fig. EV7). Collectively, our results indicate that the loss of epithelial Stard7 in Apc$^{+/Min}$ mice has dramatic consequences on mitochondrial metabolism.

We also conducted lipidomic analyses with extracts from the colon of 100-day-old Apc$^{+/Min}$ mice lacking or not Stard7 in IECs and noticed that changes in lipids levels seen upon Stard7 deficiency in this strain were very similar to changes seen in healthy mice (Figs. 7A and 1C, respectively). Indeed, levels of multiple phosphatidylcholine (PC), phosphatidylinositol (PI) and phosphatidylethanolamine (PE) derivatives increased in IECs lacking Stard7, which presumably results from a defective PC transport (Fig. 7A). On the other hand, levels of both plasmanyl and plasmenyl lipids decreased in IECs lacking Stard7 (Fig. 7A). Therefore, Stard7 deficiency causes profound consequences on the lipidomic signature of IECs in an Apc-independent manner.

## Apc-mutated intestinal tumors lacking Stard7 are sensitive to mTORC1 inhibition

Given the fact that intestinal tumors arising from constitutive Wnt signaling and lacking Stard7 show enhanced mTORC1 signaling, we hypothesized that these malignancies would be very sensitive to mTORC1 pharmacological inhibition. To address this issue, we treated Apc$^{+/Min}$/Stard7$^{Lox/lox}$ and Apc$^{+/Min}$/Stard7$^{ΔIEC}$ mice with the mTORC1 inhibitor Rapamycin and assessed the consequences on mTORC1 activity as well as on tumor number and size. As expected, Rapamycin abolished 4Ebp1 phosphorylation in intestinal tumors from Apc$^{+/Min}$/Stard7$^{ΔIEC}$ mice after 7 days of treatment (Fig. EV8A). Interestingly, all Atf4 target genes, namely Asns, Psat1 and Psph were downregulated in IECs from Rapamycin-treated Apc$^{+/Min}$/Stard7$^{ΔIEC}$ mice (Fig. EV8A). Of note, Rapamycin did not influence Ndufs1 expression (Fig. EV8A). Importantly, both tumor number and tumor size were dramatically decreased in Apc$^{+/Min}$/Stard7$^{ΔIEC}$ mice treated with Rapamycin for 20 days, indicating that mTORC1 critically contributes to tumor development in Apc-mutated mice lacking Stard7 expression in IECs (Fig. EV8B–D).

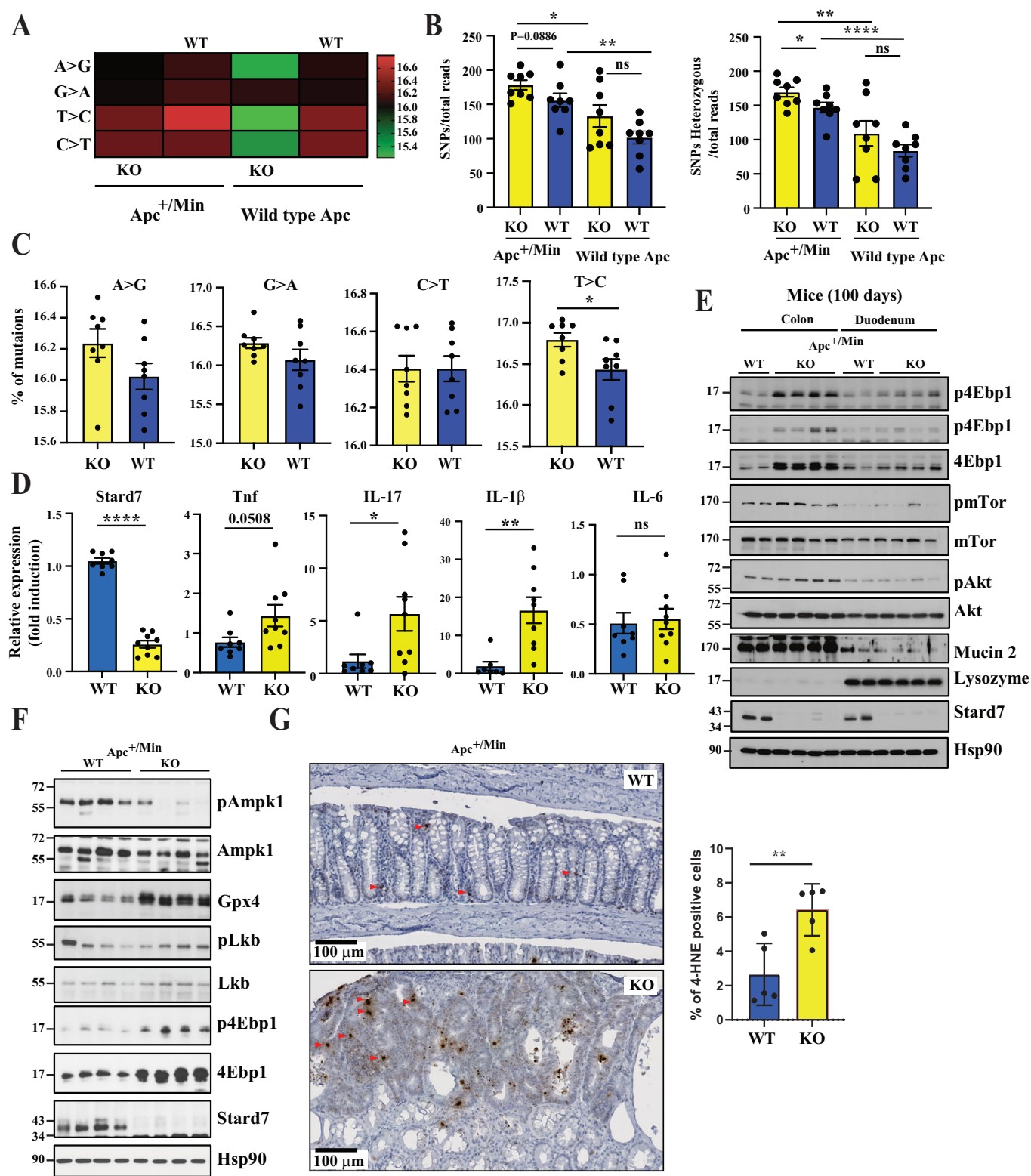

Collectively, our results demonstrate that Stard7 expression in IECs negatively regulates intestinal tumor development upon constitutive Wnt signaling, at least by limiting the activation of the mTORC1/Atf4 signaling cascade and downstream serine biosynthesis.

## mTORC1 activity is controlled by ROS in IECs lacking Stard7 expression

Having established that Stard7 negatively regulates mTORC1 activation in IECs, we next explored the underlying molecular

◄ 

**Figure 5.  Higher mutational rate of intestinal tumors showing constitutive Wnt signalling and lacking Stard7 expression.**

(A, B) Tumors from 100-day-old Apc$^{+/Min}$/Stard7$^{ΔIEC}$ mice (KO) show more mutations than tumors from 100-day-old Apc$^{+/Min}$/Stard7$^{Lox/lox}$ mice (WT), as judged by exome sequencing analyses (A). The total number of SNPs or SNPs found in coding sequences was quantified in the indicated genotypes. The mutational rate was also established in IECs from both Stard7$^{Lox/lox}$ (WT) and Stard7$^{ΔIEC}$ mice (KO) (wild type Apc) (A, B) (means ± S.D., Unpaired $T$ test, $n = 8$ for all genotypes, SNP/total reads: Apc$^{+/Min}$ background, KO versus WT mice: $P = 0.0886$; KO mice, Apc$^{+/Min}$ background versus wild type Apc: *$P = 0.0220$; WT mice, Apc$^{+/Min}$ background versus wild type Apc, **$P = 0.0014$; wild type Apc, KO versus WT mice, ns, $P = 0.2373$, not significant. SNPs Heterozygous/total reads: Apc$^{+/Min}$ background, KO versus WT mice: *$P = 0.0446$; KO mice, Apc$^{+/Min}$ background versus wild type Apc: **$P = 0.0085$; WT mice, Apc$^{+/Min}$ background versus wild type Apc, ****$P < 0.0001$; wild type Apc, KO versus WT mice, ns, $P = 0.1147$, not significant. Apc$^{+/Min}$ background, WT mice: 5 males and 3 females, Apc$^{+/Min}$ background, KO mice: 5 males and 3 females. Wild type Apc background: WT mice: 3 males and 5 females, wild type Apc background, KO mice: 4 males and 4 females. (C) Higher rate of T-C mutations in intestinal tumors from Apc$^{+/Min}$/Stard7$^{ΔIEC}$ mice (KO) than from Apc$^{+/Min}$/Stard7$^{Lox/lox}$ mice (WT) (means ± S.D., Unpaired $T$ test, $n = 8$ mice for each genotype, A > G: $P = 0.1269$; G > A: $P = 0.1092$; C > T: $P = 0.1448$; T > C: *$P = 0.033$). (D) Stard7 deficiency in IECs enhances mRNA levels of multiple pro-inflammatory cytokines. mRNA levels of the indicated candidates in IECs of 100 days old mice of the indicated genotypes (Apc$^{+/Min}$/Stard7$^{Lox/lox}$ (WT) and Apc$^{+/Min}$/Stard7$^{ΔIEC}$ mice (KO)) were quantified by quantitative Real-Time PCR analyses. Levels of these candidates in one randomly selected Apc$^{+/Min}$/Stard7$^{Lox/lox}$ mice were set to 1 and levels in other mice were relative to that after normalization with Gapdh mRNA levels (means ± S.D., $T$ test with Welch correction, $n > 4$ mice for each genotype (5 WT males and 3 WT females; 5 KO males and 4 KO females), Stard7: ****$P < 0.0001$, Tnf: $P = 0.0508$, IL-17: *$P = 0.0270$; IL-1β: **$P = 0.0023$; IL-6: $P = 0.7731$, ns = not significant. (E) Enhanced mTORC1 but not Akt activation in IECs from Apc$^{+/Min}$ mice lacking Stard7 in IECs. Total cell extracts were generated with colon and duodenum of 100-day-old mice of the indicated genotype (Apc$^{+/Min}$/Stard7$^{Lox/lox}$ (WT) and Apc$^{+/Min}$/Stard7$^{ΔIEC}$ (KO) mice) (2 WT males, 2 KO males and 2 KO females) and western blot analyses were conducted using the indicated antibodies. p4Ebp1 phosphorylation was assessed on Threonine 70 and on Threonines 37 and 46 (top and second panel from the top, respectively). (F) Defective Ampk phosphorylation but enhanced Gpx4 protein levels in IECs from Apc$^{+/Min}$ mice lacking Stard7 in IECs. IECs were extracted from colon and duodenum of 100 days old mice of the indicated genotype (Apc$^{+/Min}$/Stard7$^{Lox/lox}$ (WT) and Apc$^{+/Min}$/Stard7$^{ΔIEC}$ (KO) mice) (3 WT males, 1 WT female; 2 KO males, 2 KO females) and western blot analyses were conducted using the indicated antibodies. p4Ebp1 phosphorylation was assessed on Threonine 70. (G) Enhanced ferroptosis upon Stard7 deficiency in IECs in mice showing constitutive Wnt signaling. Immunohistochemistry analyses were conducted in the colon of the indicated genotypes to quantify levels of 4-HNE. On the left, representative images of colon sections are illustrated. On the right, the percentage of positive cells was calculated as the number of positive cells divided by the total number of detected cells ($n = 5$ mice for each genotype, means ± S.D., Student $T$ test, **$P = 0.0076$) (3 WT males and 2 WT females; 4 KO males and 1 KO female). Source data are available online for this figure.

mechanism. As Stard7 promotes PC transfer from ER to mitochondria, we hypothesized that Stard7 deficiency may trigger some mitochondrial stress. Because we had seen more ROS production upon STARD7 deficiency in colon cancer-derived cell lines showing constitutive Wnt signaling (Fig. 1F), we hypothesized that enhanced ROS production seen in IECs cells lacking Stard7 would be a key factor driving enhanced mTORC1 activation. To actually explore the link between ROS production and mTORC1 activation in vivo, we treated or not 100 days old Apc$^{+/Min}$/Stard7$^{Lox/lox}$ and Apc$^{+/Min}$/Stard7$^{ΔIEC}$ mice with the anti-oxidant N-Acetyl Cysteine (NAC) and assessed the consequences on mTORC1 activation. NAC severely impaired 4Ebp1 phosphorylation in intestines from Apc$^{+/Min}$/Stard7$^{ΔIEC}$ mice, indicating that ROS promotes mTORC1 activation (Fig. 7B, left panels). As a result, Sars, Psph, Phgdh, Asns and Psat1 expression were downregulated by NAC (Fig. 7B). Surprisingly, NAC did not interfere with the number of tumors in the distal colon of both Apc$^{+/Min}$/Stard7$^{Lox/lox}$ and Apc$^{+/Min}$/Stard7$^{ΔIEC}$ mice (Fig. 7B, right panels). NAC may have mTORC1-independent and pro-tumoral effects in addition to anti-tumoral mTORC1-dependent effects (Fig. 7B, right panels). Therefore, ROS controls mTORC1 activation with consequences on the mTORC1/Atf4 downstream signature in the intestine from Apc$^{+/Min}$/Stard7$^{ΔIEC}$ mice.

## Apc status controls the cellular response of IECs to ROS

As Stard7 deficiency leads to similar consequences on mTORC1 activation in both healthy and Apc-mutated mice, the activation of this signaling pathway could not explain the distinct role of Stard7 in AOM/DSS- versus Wnt-dependent tumor development in the intestine. Therefore, we hypothesized that the Apc genetic status, which is distinct in both experimental models, may underlie the distinct consequences of Stard7 deficiency on tumor development. To assess this issue, we treated or not ex-vivo organoids with H$_2$O$_2$

and assessed the consequences on mTORC1 activation as well as on levels of oncogenic effectors. H$_2$O$_2$ triggered 4Ebp1 phosphorylation in both wild type and mutated Apc ex-vivo organoids (Fig. 7C). However, H$_2$O$_2$-dependent Jnk phosphorylation was only observed in healthy ex-vivo organoids, suggesting that Apc-mutated ex-vivo organoids resist to cell death triggered by H$_2$O$_2$ (Fig. 7C). Likewise, Wnt effectors such as Sox9 and c-Myc, which is known to inhibit JNK-dependent cell apoptosis in vivo (Huang et al, 2017), were induced by H$_2$O$_2$ in Apc-mutated but not wild type ex-vivo organoids. Moreover, the profile of Ampk1 phosphorylation was also relying on the Apc status (Fig. 7C). Collectively, these results suggest that the response of IECs to oxidative stress depends on the Apc mutational status.

## The microbiota signature relies on epithelial Stard7 expression in Apc$^{+/Min}$ mice

To further explore mechanisms underlying the enhanced tumor development in the distal colon of Apc$^{+/Min}$ lacking Stard7 expression in IECs and because gut microbiota promotes tumor development in Apc$^{+/min}$ mice (Li et al, 2012), we next explored whether gut bacteria contribute to this phenotype. To assess this issue, we treated or not both Apc$^{+/Min}$/Stard7$^{Lox/lox}$ mice and Apc$^{+/Min}$/Stard7$^{ΔIEC}$ mice with broad-spectrum antibiotics (Abx) known to deplete commensal bacteria in mice (Göktuna et al, 2016). Antibiotics severely decreased the number of tumors in the distal colon of Apc$^{+/Min}$/Stard7$^{ΔIEC}$ mice, thus demonstrating that tumor development in these mice is partially microbiota-dependent (Fig. 8A). To further identify the bacteria that contribute to tumor development in Apc$^{+/Min}$/Stard7$^{ΔIEC}$ mice, we compared intestinal microbiota signatures in both Apc$^{+/Min}$/Stard7$^{Lox/lox}$ and Apc$^{+/Min}$/Stard7$^{ΔIEC}$ mice (Fig. 8B). β-diversity analysis revealed a modest but statistically significant separation using Aitchison ($P = 0.049$) and weighted UniFrac distances ($P = 0.043$), whereas non-phylogenetic

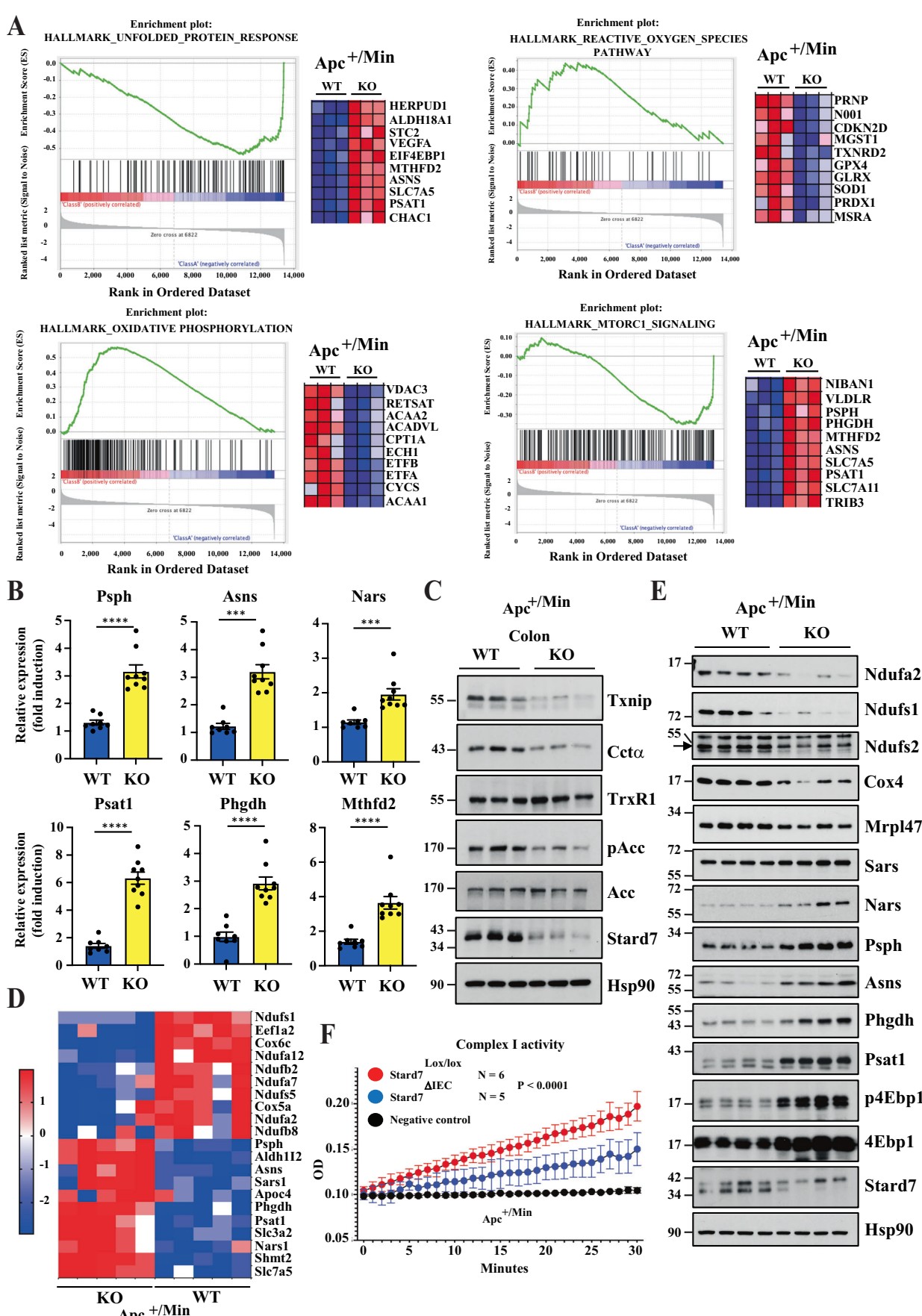

**Figure 6. Transcriptional reprogramming upon epithelial Stard7 deficiency in Apc⁺/Min mice.**

(A) IECs lacking Stard7 and showing constitutive Wnt signalling show deregulated transcriptional signatures related to multiple signaling pathways. GSEA analyses were conducted with RNA Sequencing data obtained with total RNAs from IECs of 100 days old Apc$^{+/Min}$/Stard7$^{Lox/lox}$ (WT) and Apc$^{+/Min}$/Stard7$^{\Delta IEC}$ (KO) mice (3 WT males, 2 KO males, 1 KO female). Genes with the highest enrichment score are shown as individual gene expression-based heat maps. HeatMaps demonstrate upregulation of both mTORC1 and UPR signaling pathways (candidates in red) as well as downregulation of oxidative phosphorylation as well as cellular response to ROS species (candidates in blue) in IECs lacking Stard7. The color code in HeatMaps corresponds to the color code illustrated in enrichment plots. (B) Enhanced mRNA levels of Atf4 target genes in IECs from Apc$^{+/Min}$/Stard7$^{\Delta IEC}$ mice. Quantitative Real-Time PCR analyses were conducted with total RNAs from IECs of 100 days old Apc$^{+/Min}$/Stard7$^{Lox/lox}$ (WT) and Apc$^{+/Min}$/Stard7$^{\Delta IEC}$ (KO) mice. Levels of the indicated candidates in one randomly selected Apc$^{+/Min}$/Stard7$^{Lox/lox}$ mice were set to 1 and levels in other mice were relative to that after normalization with Gapdh mRNA levels (means ± S.D., $T$ test with Welch correction, $n > 8$ mice for each genotype, Psph: ****$P < 0.0001$; Asns: ****$P < 0.0001$; Nars: ***$P = 0.0010$; Psat1: ****$P < 0.0001$; Phgdh: ****$P < 0.0001$; Mthfd2: ****$P < 0.0001$) (5 WT males, 3 WT females; 5 KO males, 4 KO females). (C) Defective expression of candidates involved in the cellular response to oxidative stress and in oxidative phosphorylation in IECs lacking Stard7 and showing constitutive Wnt signalling. Extracts from IECs of 100 days old Apc$^{+/Min}$/Stard7$^{Lox/lox}$ (WT) and Apc$^{+/Min}$/Stard7$^{\Delta IEC}$ (KO) mice were subjected to western blot analyses using the indicated antibodies (1 WT male, 3 WT females; 1 KO male, 2 KO females). (D, E) Defective expression of Ndu proteins but enrichment of candidates downstream of Atf4 in IECs lacking Stard7 and showing constitutive Wnt signalling. A HeatMap generated with proteomic data and western blot analyses are illustrated (D, E, respectively). Protein extracts were generated from IECs of 100 days old Apc$^{+/Min}$/Stard7$^{Lox/lox}$ (WT) and Apc$^{+/Min}$/Stard7$^{\Delta IEC}$ (KO) mice. p4Ebp1 phosphorylation was assessed on Threonine 70 (E). The arrow depicts the specific band for Ndufs2 (E). For the HeatMap, 3 WT males, 2 WT females; 4 KO males, 1 KO female). For western blot analyses, 3 WT males 1 WT female, 2 KO males, 2 KO females). (F) IECs from Apc$^{+/Min}$ mice and lacking epithelial Stard7 show a defective Complex I activity. Complex I activity was measured with extracts from IECs of Apc$^{+/Min}$ mice lacking or not Stard7 (means ± S.D., two-way ANOVA, ****$P < 0.0001$, $n = 6$ WT and 5 KO mice) (4 WT males, 2 WT females, 4 KO males, 2 KO females). Source data are available online for this figure.

(Bray–Curtis: $P = 0.06$; Jaccard: $P = 0.075$) and presence/absence-based phylogenetic metrics (unweighted UniFrac: $P = 0.20$) did not reach statistical significance (Fig. 8C). This pattern indicates that groups differed mainly through subtle abundance shifts in phylogenetically related taxa, rather than through major changes in overall community membership. At Phylum level, samples from Apc$^{+/Min}$/Stard7$^{\Delta IEC}$ mice were enriched in *Bacteroidota* ($P = 0.03$) and showed lower levels of *Firmicutes* ($P = 0.05$) (Fig. 8D), resulting in a significantly increased *Bacteroidota/Firmicutes* ratio ($P = 0.026$) (Fig. 8D). Despite these shifts, comprehensive differential abundance analysis using four independent statistical methods (ANCOM-BC2, ALDEx2, MaAsLin2, and DESeq2) did not identify any taxa that remained significant after correction for multiple testing. Several families and genera exhibited notable trends when examining raw p-values. Taxonomic resolution at family level revealed the decrease in *Firmicutes* was mainly driven by a decrease in *Lachnospiraceae* ($P = 0.07$) and the increase in *Bacteroidota* was primarily due to a significant increase in *Prevotellaceae* ($P = 0.015$) (Fig. 8E). Interestingly, at genus level, we found that *Paraprevotella* was increased in Apc$^{+/Min}$/Stard7$^{\Delta IEC}$ mice and absent in all but one Apc$^{+/Min}$/Stard7$^{Lox/lox}$ mouse ($P = 0.015$) (Fig. 8F).

Although of a predictive character, functional community profiling using Tax4fun (Abhauer et al, 2015) suggested that the microbiota composition found in samples of Apc$^{+/Min}$/Stard7$^{\Delta IEC}$ mice could potentially be associated with more cancers and with both one carbon metabolism and decreased AMPK signaling signatures. This is in agreement with our transcriptomic and proteomic analyses (Fig. 8G). In conclusion, the loss of Stard7 in IECs may, at least in part, potentiate tumor development in the distal colon of Apc$^{+/Min}$ mice by reprogramming microbiota composition in the intestine.

## Discussion

We report here that the loss of the lipid transfer protein Stard7, which triggers a mitochondrial stress and enhances ROS production, has a context-dependent role in intestinal tumor development. Indeed, Stard7 contributes to inflammation-driven intestinal tumor development but unexpectedly limits Wnt-driven tumor initiation in the distal colon.

Stard7 expression in IECs is dispensable in intestinal homeostasis, as all tested epithelial markers as well as tight junction proteins are properly expressed in Stard7$^{\Delta IEC}$ mice. A recently published study demonstrated that haplotypic expression of Stard7 impairs the expression of tight junctions proteins such as Claudins, which is in contrast to what we report with our Stard7$^{\Delta IEC}$ strain (Uddin et al, 2024). To assess whether this phenotype was intrinsic to the epithelial cell compartment, Stard7 was genetically inactivated using the Sonic Hedgehog (Shh)-CRE in that study. Although some defects in tight junctions were still seen in these mice, this is not the final demonstration that these defects result from the loss of Stard7 in all IECs. The hedgehog ligand Shh is indeed detected in the proliferative pseudostratified epithelium of the early gut tube before villus morphogenesis (Walton and Gumucio, 2020). However, when villi are emerging, Shh expression is only retained in proliferative cells of the intervillus strain (Walton and Gumucio, 2020). Importantly, Shh is also detected in multiple other organs, including the skin where any damage that results from a genetic inactivation may have strong consequences on the microbiome and consequently on colitis triggered by DSS (Dokoshi et al, 2024). Therefore, any phenotype that results from the genetic inactivation of Stard7 using the Shh-CRE is not expected to perfectly mimic defects using the Villin-CRE where only IECs are targeted.

The severe phenotype seen upon epithelial Stard7 deficiency in our mouse models of intestinal cancer results from impaired mitochondrial functions. Indeed, both healthy and Apc$^{+/Min}$ mice lacking Stard7 in IECs suffer from an impaired mitochondrial Complex I activity, at least due to a defective expression of multiple candidates involved in oxidative phosphorylation. This mitochondrial stress presumably results from a defective PC transport from ER to mitochondria. It is nevertheless important to mention that a defective Coenzyme Q seen upon Stard7 deficiency may also contribute to the observed phenotypes. Indeed, increased Gpx4 and 4-HNE levels seen in IECs from Apc$^{+/Min}$/Stard7$^{\Delta IEC}$ mice is a hallmark of ferroptosis, which occurs when Coenzyme Q is not properly transported from mitochondria to the plasma membrane (Deshwal et al, 2023).

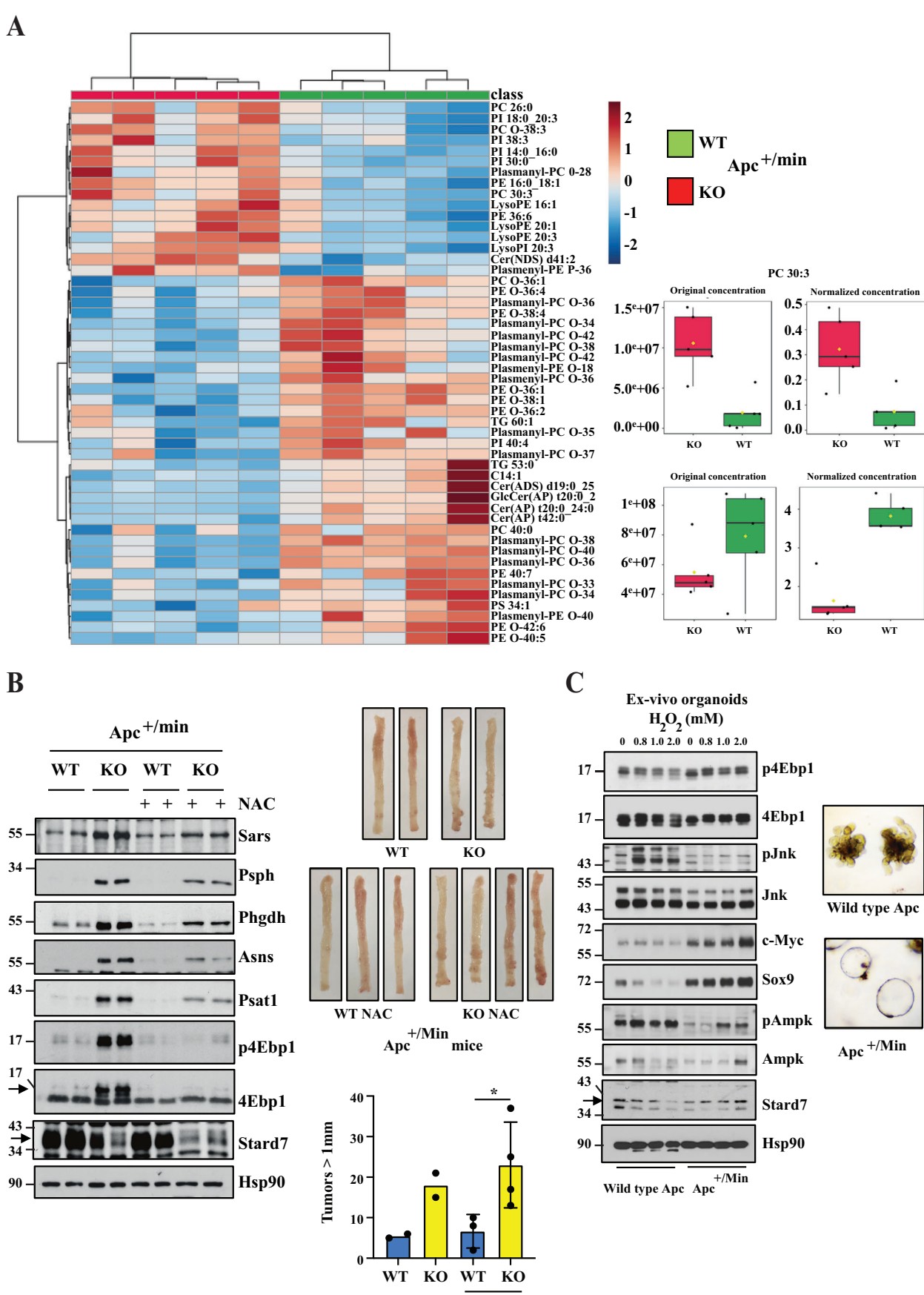

**Figure 7. Stard7 deficiency in IECs impairs mitochondrial activity in Apc$^{+/Min}$ mice and potentiates mTORC1 activation in a ROS-dependent manner.**

(A) Stard7 deficiency in IECs from Apc$^{+/Min}$ mice show deregulated lipid levels. Extracts from the colon of 100 days old Apc$^{+/Min}$ mice of the indicated genotype were subjected to a lipidomic analysis. Stard7 deficiency in this strain leads to the accumulation of multiple phosphatidylcholine (PC), phosphatidylinositol (PI) and phosphatidylethanolamine (PE) derivatives as well as decreased levels of both plasmanyl and plasmenyl lipids ($n = 5$ for both genotypes) (3 WT males, 2 WT females; 4 KO males, 1 KO females). Heatmaps and box plot were created using the public database MetaboAnalyst https://www.metaboanalyst.ca/MetaboAnalyst/home.xhtml. Each box shows the 25th (Q1) to 75th (Q3) percentiles, with a line at the median (50th percentile) and whiskers extending to near the data's max/min (within 1.5x Interquantile Range (IQR)). (B) The mTORC1/Atf4-dependent signature found upon epithelial Stard7 deficiency in Apc$^{+/Min}$ mice is controlled by ROS. 100 days old mice of the indicated genotype (Apc$^{+/Min}$/Stard7$^{Lox/lox}$ (WT) and Apc$^{+/Min}$/Stard7$^{\Delta IEC}$ (KO) mice) were treated or not with NAC for 3 weeks and extracts from IECs of the resulting mice were subjected to western blot analyses using the indicated antibodies. p4Ebp1 phosphorylation was assessed on Threonine 70. Western blot analyses were carried out with extracts from a total of 4 untreated WT and KO mice as well as from 5 WT and KO mice treated with NAC. Representative western blots are illustrated. The arrow depicts the specific band for 4Ebp1 and Stard7. Representative intestines from both genotypes treated or not with NAC are illustrated on the top. At the bottom, a quantification of the number of tumors ( > 1 mm) is illustrated in all experimental conditions (means ± S.D., ANOVA test, Sidak's multiple comparisons test, $n = 2$ mice for untreated animals, $n = 3$ WT and $n = 4$ KO for treated animals. NAC-treated WT versus NAC-treated KO mice: *$P = 0.0469$. 3 WT males, 2 WT females; 3 KO males, 3 KO females. (C) The cellular response to H$_2$O$_2$ relies on the Apc status. Ex-vivo organoids generated with IECs from healthy (wild type Apc) or Apc$^{+/Min}$ mice were stimulated or not with H$_2$O$_2$ for 5 hours at the indicated concentrations and extracts were subjected to western blot analyses using the indicated antibodies. p4Ebp1 phosphorylation was assessed on Threonine 70. Three independent experiments were carried out and representative western blots are illustrated. The arrow depicts the specific band for Stard7. Note that both Jnk1/2 isoforms are detected. Representative ex-vivo organoids of the indicated genotypes are illustrated as well. Source data are available online for this figure.

The enhanced ROS production seen in all colon cancer-derived cell lines lacking STARD7 most likely results from a mitochondrial stress. STARD7 deficiency also triggers ER stress which is also known to promote ROS production, as demonstrated in hepatocarcinoma-derived HepG2 cells as well as in triple negative breast cancer-derived MDA-MB231 cells (Flores-Martin et al, 2016; Dondajewska et al, 2025). However, this conclusion does not apply to all cell types as levels of UPR effectors known to be induced (BIP, XBP1s, IRE1α) or activated (pEIF2α) upon ER stress did not change upon STARD7 deficiency in ERα$^+$ breast cancer-derived T47D and MCF7 cells nor in IECs from 100 days old Apc$^{+/Min}$/Stard7$^{\Delta IEC}$ mice (Fig. EV9) (Dondajewska et al, 2025). The UPR signature found in our GSEAs carried out with extracts lacking Stard7 actually resulted from elevated levels of Atf4 and downstream targets rather than from the activation of all UPR branches. Mitochondrial stress, which results from the loss of Stard7 in IECs, critically contributes to the described phenotypes. Indeed, mitochondrial stress is known to induce Atf4 (Quiros et al, 2017). This Atf4-dependent transcriptional reprogramming is seen in IECs from Apc$^{+/Min}$/Stard7$^{\Delta IEC}$ mice. Importantly, this Atf4-dependent signature is controlled by mTORC1 in both healthy and Apc$^{+/Min}$ mice, indicating that Stard7 deficiency in IECs potentiated an mTORC1/Atf4 signaling axis. Enhanced mTORC1 activation, a hallmark of IECs lacking Stard7 and showing or not constitutive Wnt signaling, is ROS-dependent as NAC interferes with mTORC1 activation and consequently with the Atf4-dependent transcriptional reprogramming. As Rapamycin severely decreases the number of tumors in the distal colon of Apc$^{+/Min}$/Stard7$^{\Delta IEC}$ mice, this demonstrates that mTORC1 is a key effector driving tumor development in Apc-mutated intestinal epithelial cells lacking Stard7. Although NAC inhibits mTORC1 activation, it does not interfere with the number of intestinal tumors in Apc$^{+/Min}$/Stard7$^{\Delta IEC}$ mice, indicating that NAC has additional pro-tumoral functions that counteract its inhibitory effects on mTORC1 activity. In this context, antioxidants have been reported to promote intestinal tumor development in Apc$^{+/Min}$ mice (Zou et al, 2021). Although the underlying mechanisms remain unclear, it was postulated that low doses of NAC (i.e. doses we also used in our study) could neutralize damaging ROS produced by mitochondria

while having no effects on proliferative ROS produced through a RAC1-dependent pathway (Zou et al, 2021; Cheung et al, 2016).

While the link between ROS and mTORC1 has been established in our experimental models, it remains nevertheless unclear how ROS activates mTORC1. As expected, Ampk, known to inhibit mTORC1, is less active upon Stard7 deficiency in Apc$^{+/Min}$ mice. Even if Lkb activity was not dramatically changed upon Stard7 deficiency in IECs, ROS can actually inhibit Lkb kinase activity through the formation of a covalent adduct on Cys210 within the activation loop (Wagner et al, 2006). Likewise, ROS also inhibit AMPK through the oxidation of Cys130 and 174 on its α subunit, a chemical modification which impairs the interaction between AMPK and LKB (Shao et al, 2014). Importantly, our data also suggest that bacterial products from the specific microbiota signature found in the intestine of Apc$^{+/Min}$/Stard7$^{\Delta IEC}$ mice may also inhibit Ampk activation. In this context, it is also important to note that the enhanced mTORC1 activation seen in Apc$^{+/Min}$/Stard7$^{\Delta IEC}$ mice has been seen in the colon and to a less extent in the duodenum. This observation, combined with the specific microbiota signature found in the colon of these mice, opens the possibility that oncometabolites produced by some bacterial strains may also contribute to mTORC1 activation. Therefore, mTORC1 activity is potentially regulated by multiple pathways upon Stard7 deficiency in Apc$^{+/Min}$ mice. Importantly, mTORC1 also has a context-dependent role on intestinal tumor development with opposite roles than Stard7 (Brandt et al, 2018). Indeed, mTORC1 promotes Wnt-driven tumor initiation but limits colon cancer development upon chronic inflammation. Therefore, mTORC1 signaling is a key element through which Stard7 differentially controls tumor development in the intestine. Such conclusion is further supported by the fact that tumors in the distal colon of Apc$^{+/Min}$/Stard7$^{\Delta IEC}$ mice are extremely sensitive to mTORC1 pharmacological inhibition. Downstream of the potentiated mTORC1/Atf4 signaling axis seen in IECs from Apc$^{+/Min}$/Stard7$^{\Delta IEC}$ mice is the one carbon metabolism signature which includes multiple enzymes involved in serine biosynthesis. As a result, intestinal tumors from Apc$^{+/Min}$/Stard7$^{\Delta IEC}$ mice are expected to be less sensitive to serine deprivation, as previously described for K-Ras-driven intestinal tumors (Maddocks et al, 2017).

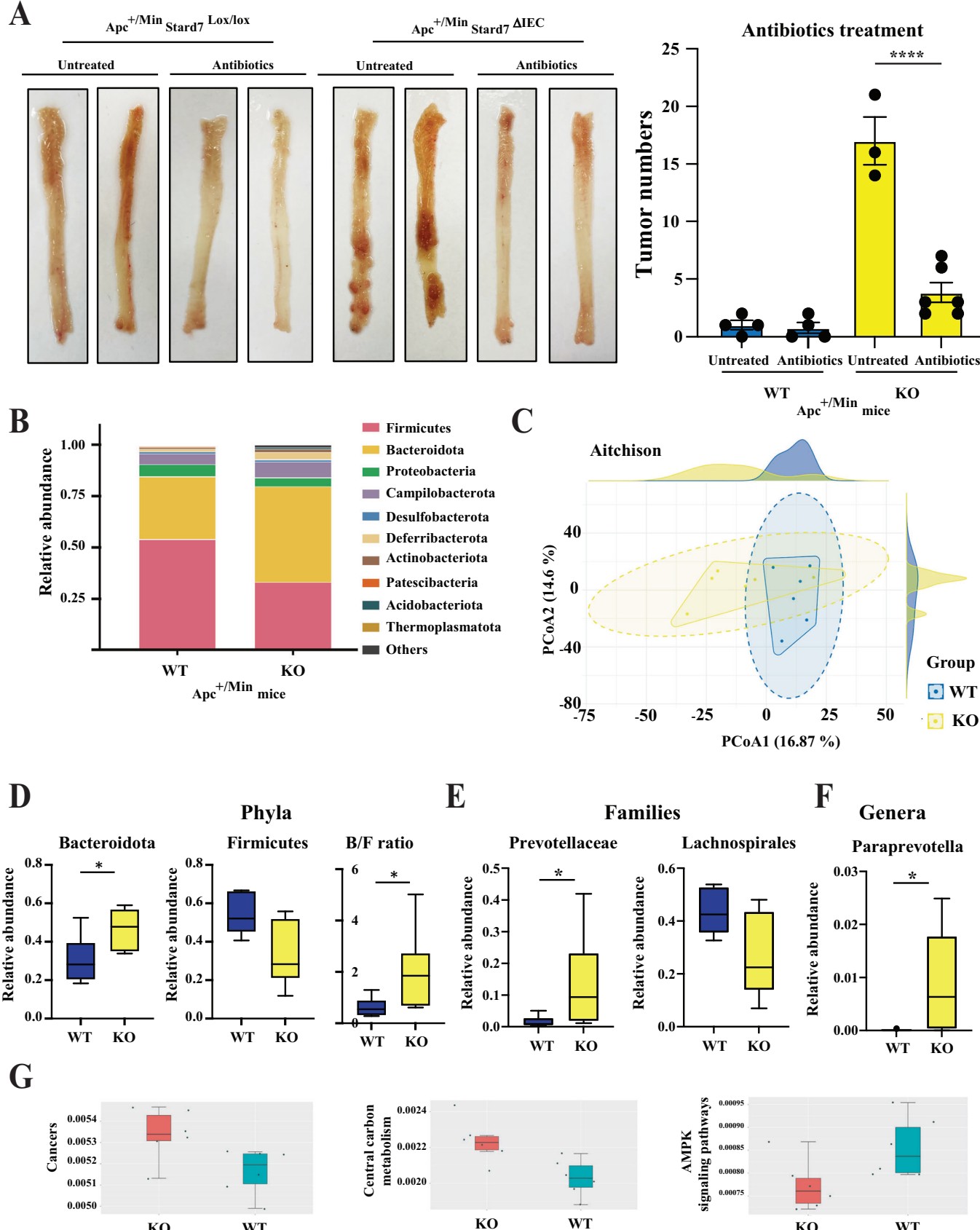

◀

**Figure 8. Epithelial Stard7 deficiency in Wnt-driven intestinal tumors is associated to a specific microbiota signature.**

(A) The enhanced tumor development in the distal colon of Apc$^{+/Min}$/Stard7$^{ΔIEC}$ (KO) mice is microbiota-dependent. 100 days old mice of the indicated genotype (Apc$^{+/Min}$/Stard7$^{Lox/lox}$ (WT) and Apc$^{+/Min}$/Stard7$^{ΔIEC}$ (KO) mice) ($n \geq 3$) were treated or not with a combination of antibiotics (see methods for details) and the number of tumors were counted in the resulting mice (4 WT males, 4 WT females; 4 KO males, 5 KO females). Representative colons of all experimental conditions are illustrated on the left and the quantification of the number of tumors is illustrated on the right (means ± S.D., ANOVA test, Tukey's multiple comparisons test. Untreated versus antibiotics-treated KO mice, ****$P < 0.0001$). (B–F) Epithelial Stard7 deficiency in 100 day old Apc$^{+/Min}$ mice leads to the establishment of a specific microbiota signature in the intestine. Cecal microbiota composition was determined using 16S rRNA amplicon sequencing (V4 region) ($n = 6$) and the top phyla are shown in bar plots per genotype (B). Mice are individually represented in a principal coordinate analysis (PCoA) with a Weight Unifrac distance metric (C). Principal Coordinates Analysis (PCoA) based on weighted UniFrac distances showing modest but significant separation between WT (blue) and KO (yellow) mice. Individual samples are shown as points; shaded polygons represent group hulls, and dashed ellipses depict 95% confidence intervals around group centroids. Marginal density plots display the distribution of samples along PCoA1 and PCoA2. Differentially abundant taxa are shown at the level of Phylum (D) (means + S.D., Mann–Whitney test, Bacteroidota: *$P = 0.0411$; Firmicutes: $P = 0.0649$; B/F ratio: *$P = 0.0260$, $n = 6$ mice for both genotypes), Family (E) (means + S.D., Mann– Whitney test, Prevotellaceae: *$P = 0.0152$; Lachnospiraceae: $P = 0.0931$. $n = 6$ mice for both genotypes) and genus (F) (means + S.D., Mann–Whitney test, Paraprevotella: *$P = 0.0167$). For box and whisker plots, the center line indicates the median (50th percentile); the box bounds correspond to the 25th and 75th percentiles (interquartile range). Whiskers extend to the minimum and maximum values. 4 WT males, 2 WT females; 4 KO males, 2 KO females. (G) Deregulated signaling pathways and biological processes associated with the specific microbiota signature found in Apc$^{+/Min}$/Stard7$^{ΔIEC}$ mice ($n = 6$ mice per genotype). For box and whisker plots, the center line indicates the median (50th percentile); the box bounds correspond to the 25th and 75th percentiles (interquartile range). Whiskers extend to the minimum and maximum values. Source data are available online for this figure.

The mitochondrial stress seen upon Stard7 deficiency leads to a severe metabolic reprogramming first characterized by a defective OXPHOS. We report that ACC phosphorylation, which is AMPK-dependent, is impaired upon Stard7 deficiency, which makes ACC more active. This observation fits with the fact that Ampk phosphorylation is defective in IECs lacking Stard7 and a more active ACC may reflect a compensatory mechanism in cells dealing with a defective OXPHOS. Stard7 deficiency is also characterized by the accumulation of multiple amino acids and TCA intermediates as well as by decreased levels of fatty acids. While increased amino acid levels result from the potentiated one carbon metabolism signature seen upon Stard7 deficiency, it is currently unclear why fatty acids are decreasing in IECs from Apc$^{+/Min}$/Stard7$^{ΔIEC}$ mice. This observation has to result from an imbalance between fatty acid oxidation and fatty acids supply. Mitochondria lacking Stard7 may face OXPHOS deficiency by potentiating FAO as an alternative source of energy. In this context, we demonstrated that the maximum oxygen consumption rate severely decreased upon Stard7 deficiency in cells treated with the FAO inhibitor Etomoxir, which experimentally supports this hypothesis. Interestingly, glycolysis is used as another alternative source of energy is myoblasts lacking Stard7, which suggests that the metabolic reprogramming seen upon Stard7 deficiency is cell type-dependent (Rojas et al, 2024). Beside these deregulations in metabolism, IECs lacking Stard7 also show an accumulation of PC, PE and PI, which is presumably related to the key role of Stard7 in PC transport. On the other hand, Stard7 deficiency in IECs is also associated with decreased levels of both plasmanyl and plasmenyl lipids, independently of the Apc status. Beside a role in membrane dynamics, plasmanyl and plasmenyl lipids, defined as Plasmalogens, also have a protective role against an oxidative stress (Nagan and Zoeller, 2001). Therefore, the decreased levels of both plasmanyl and plasmenyl lipids may render Stard7-deficient cells more sensitive to ROS-dependent cellular damage.

Both lipidomic and transcriptomic analyses strongly indicate that all deregulations seen upon Stard7 deficiency result from a mitochondrial stress as the causal event, independently of the Apc status. Yet, the role of Stard7 on tumor development unexpectedly relies on the Apc status. Our experiment carried out in ex-vivo organoids derived from intestinal crypts demonstrated that ROS induces cell death in wild type Apc-expressing but not in Apc-mutated ex-vivo organoids. This result perfectly fits with our in vivo data in which we show that epithelial Stard7 deficiency in Apc$^{+/Min}$ but not in AOM/DSS-treated mice potentiates tumor development. The accumulation of ROS-dependent mutations in wild type-expressing cells is toxic but rather promotes tumor development in Apc-mutated cells. While the source of ROS dictates the outcome on tumor development (Cheung et al, 2016), our study therefore demonstrates that the genetic status has also to be taken into account to understand the role of ROS in cancer. Our lipidomic analyses indicate that Stard7 deficiency leads to a profound reprogramming of the lipid landscape in cellular organelles such as ER, mitochondria and lysosomes. This observation suggests that the loss of Stard7 has a strong impact on PC levels in both ER and mitochondria with some consequences on the composition of other lipids found in multiple membranes. Our results also suggest that Stard7 may regulate tumor development through additional ROS-independent pathways and involving lipids as signaling molecules. While this hypothesis is interesting to experimentally address, it is important to note that the changes in the lipid landscape seen in cells lacking Stard7 were very similar in both wild type and Apc-mutated intestinal epithelial cells. Yet, Stard7 has opposite roles in tumor development in both experimental models, which strengthens the notion that the Apc status has to be taken into account for any experiment aiming at defining the role of lipids as signaling proteins in cancer.

Our results demonstrate that the loss of Stard7 in IECs impairs mRNA levels of pro-inflammatory cytokines in the inflammation-driven model of intestinal cancer but enhances their mRNA levels in Apc$^{+/min}$ mice. We and others demonstrated that the loss of Apc disrupts the architecture of the single layer of intestinal epithelial cells, which leads to the establishment of a pro-inflammatory signature, at least through TLR activation (Göktuna et al, 2016). Therefore, both Wnt-driven tumor development as well as the inflammation-driven cancer model both rely on an inflammatory signature to support tumor development. However, the key difference for both model is the Apc status. As this specific mutational status helps intestinal cells to better cope with ROS levels, several pro-survival oncogenic signaling pathways are robustly activated, which leads to the production of chemokines and to the attraction of Tumor-Associated Macrophages (TAMs), which are the main source of pro-inflammatory cytokines. This is

most likely the reason why Apc$^{+/Min}$/Stard7$^{\Delta IEC}$ mice show elevated levels of these cytokines. These pro-inflammatory cytokines contribute to tumor development in both experimental models but their distinct expression profile in control versus Stard7-inactivated epithelial cells reflects the opposite outcome on tumor development in both models due to distinct Apc status. It is also interesting to note that tumors from AOM/DSS-treated Stard7$^{\Delta IEC}$ mice show lower levels of the anti-oxidant protein Tigar, a candidate whose deficiency enhances the production of cytokines by pancreatic cancer cells (Cheung et al, 2024). Therefore, it is unlikely that lower levels of Tigar seen upon Stard7 deficiency in our inflammatory-driven tumor development model contributes to the decreased mRNA levels of pro-inflammatory cytokines, even if both mouse models of pancreatic and colon cancers may have organ-specific differences.

The fact that STARD7 acts as a tumor suppressor gene in Wnt-driven tumor development was unexpected given the fact that we and others showed that STARD7 is overexpressed in clinical cases of colon cancer known to show constitutive Wnt signaling in most cases (Zhao et al, 2023). While STARD7 overexpression is colon cancer is now well established, the link with Wnt signaling is not clearly established. Both STARD7 and TCF4 expression were reported to be positively correlated (Zhao et al, 2023). However, our data obtained in multiple experimental models do not support the notion that Wnt signaling promotes STARD7 expression. We rather show that STARD7 acts upstream of Wnt signaling by controlling the expression of several Wnt effectors, even if the underlying mechanism remains unclear.

While Apc$^{+/Min}$ mice do not perfectly mimic human intestinal cancers due to the limited number of tumors in the distal colon, we show here that Apc$^{+/Min}$/Stard7$^{\Delta IEC}$ mice represent a much better experimental model as it is phenotypically more similar to human colon malignancies. Such conclusion also applied to Apc$^{+/Min}$ mice lacking Glutathione S-transferase Pi (GSTP) in which higher levels of some pro-inflammatory cytokines were seen, similarly to Apc$^{+/Min}$/Stard7$^{\Delta IEC}$ mice (Ritchie et al, 2009). The higher number of tumors in the distal colon of Apc$^{+/Min}$/Stard7$^{\Delta IEC}$ mice is associated with enriched levels of Bacteroidota and lower levels of Firmicutes. The ratio of both bacterial phyla appears to be critical in colon cancer progression (Pandey et al, 2023). Collectively, our data demonstrate that transformed and ROS-producing IECs lacking Stard7 potentiate the activation of oncogenic signaling pathways in the intestine and are associated with modifications of the gut microbiota.

# Methods

### Reagents and tools table

| Reagent/resource | Reference or source | Identifier or catalog number |
|---|---|---|
| **Experimental models** | | |
| HCT116 cell line | ATCC | CCL-247 |
| HT-29 cell line | ATCC | HTB-38 |
| SW480 cell line | ATCC | CCL-228 |
| COLO205 cell line | ATCC | CCL-222 |

| Reagent/resource | Reference or source | Identifier or catalog number | |
|---|---|---|---|
| DLD1 cell line | ATCC | CCL-221 | |
| HCT15 cell line | ATCC | CCL-225 | |
| **Recombinant DNA** | | | |
| **Antibodies** | | | |
| STARD7 | PA5-30772 | Invitrogen | 1/1000 |
| STARD7 | HPA064978 | Sigma | 1/2000 |
| Chop | #2895 | Cell Signaling Technologies | 1/1000 |
| pEif2α | #3398 | Cell Signaling Technologies | 1/1000 |
| Eif2α | #5324 | Cell Signaling Technologies | 1/1000 |
| Claudin3 | # 34-1700 | ThermoFisher Scientific | 1/1000 |
| E-cadherin | # 610181 | BD Biosciences | 1/1000 |
| TrxR1 (B-2) | sc-28321 | Santa Cruz | 1/500 |
| TXNIP (D5F3E) | #14715 | Cell Signaling Technologies | 1/1000 |
| p-AKT (S473) (D9E) | #4060 | Cell Signaling Technologies | 1/1000 |
| Perk | #3192 | Cell Signaling Technologies | 1/1000 |
| IP3R | #8568 | Cell Signaling Technologies | 1/1000 |
| Cleaved caspase 3 | 9664s | Cell Signaling Technologies | 1/1000 |
| HSP90 | sc-13119 | Santa Cruz | 1/1000 |
| c-Myc | 5605s | Cell Signaling Technologies | 1/1000 |
| Lysozyme | A0099 | DAKO | 1/5000 |
| SOX9 | AB5535 | Millipore | 1/1000 |
| Mucin2 | SC-15334 | Santa Cruz | 1/500 |
| Cyclin D1 | #2978 | Cell Signaling Technologies | 1/1000 |
| Phospho-4EBP1 T70 | #9455 | Cell Signaling Technologies | 1/1000 |
| 4EBP1 | #9644 | Cell Signaling Technologies | 1/1000 |
| Phospho-4EBP1 T37/46 | #2855 | Cell Signaling Technologies | 1/1000 |
| Phospho-mTOR | #2974 | Cell Signaling Technologies | 1/1000 |
| mTOR | #2983 | Cell Signaling Technologies | 1/1000 |
| AKT | #4691 | Cell Signaling Technologies | 1/1000 |
| Phospho-AMPKα | #2535 | Cell Signaling Technologies | 1/1000 |

| Reagent/resource | Reference or source | Identifier or catalog number | |
|---|---|---|---|
| AMPKa | #2532 | Cell Signaling Technologies | 1/1000 |
| GPX4 | #59735 | Cell Signaling Technologies | 1/1000 |
| LKB1 | #3047 | Cell Signaling Technologies | 1/1000 |
| Phospho-LKB1 | #3482 | Cell Signaling Technologies | 1/1000 |
| CCTa | #6931 | Cell Signaling Technologies | 1/1000 |
| Phospho-ACC | #11818 | Cell Signaling Technologies | 1/1000 |
| ACC | #3662 | Cell Signaling Technologies | 1/1000 |
| Ndufa2 | PA5-96946 | Invitrogen | 1/1000 |
| Ndufs1 | #70264 | Cell Signaling Technologies | 1/1000 |
| Ndufs2 | abx338914 | Abbexa | 1/1000 |
| COXIV | #4850 | Cell Signaling Technologies | 1/1000 |
| Mrpl47 | PA5-101365 | Invitrogen | 1/1000 |
| SARS | MBS9406267 | MyBiosource, Inc | 1/1000 |
| NARS | Sc-271059 | Santa Cruz | 1/1000 |
| PSPH | ab96414 | Abcam | 1/1000 |
| ASNS | #703164 | Invitrogen | 1/1000 |
| Phgdh | 13428 | Cell Signaling Technologies | 1/1000 |
| PSAT1 | 10501-1-AP | Proteintech | 1/1000 |
| Phospho-S6 | #4858 | Cell Signaling Technologies | 1/1000 |
| S6 | #2217 | Cell Signaling Technologies | 1/1000 |
| Phospho-JNK | #9251 | Cell Signaling Technologies | 1/1000 |
| JNK | #9252 | Cell Signaling Technologies | 1/1000 |
| β-catenin | Sc-7199 | Santa Cruz | 1/1000 |
| Rab11 | #3539 | Cell Signaling Technologies | 1/1000 |
| Vdac1 | #4866 | Cell Signaling Technologies | 1/1000 |
| Ire1α | Ab37073 | Abcam | 1/1000 |
| Xbp1S | #40435 | Cell Signaling Technologies | 1/1000 |

| Reagent/resource | Reference or source | Identifier or catalog number | |
|---|---|---|---|
| Bip | #3183 | Cell Signaling Technologies | 1/1000 |
| Tigar | sc-166290 | Santa Cruz | 1/1000 |
| pH2AX | #9718 | Cell Signaling Technologies | 1/1000 |
| **Oligonucleotides and other sequence-based reagents** | | | |
| PCR primers | This study | Table EV2 | |
| shRNAs | This study | Table EV1 | |
| **Chemicals, enzymes and other reagents** | | | |
| ICRT3 | HY-103705 | MedChemExpress | |
| Rapamycin | R-5000 | LC Laboratories | |
| NAC | A7250 | Sigma | |
| **Software** | | | |
| **Other** | | | |

## Cell cultures and generation of ex-vivo organoids

Human adenocarcinoma HCT116, DLD1, HT-29 and SW480 cell lines were purchased from the American Type Culture Collection (ATCC, Manassas, VA, USA). These cells were characterized by ATCC, using a comprehensive database of short tandem repeat (STR) DNA profiles and were never cultured for more than 20 passages in our laboratory. The Lenti-X 293 T cell line was obtained from Clontech Laboratories (Palo Alto, CA, USA). All cell lines were tested for mycoplasma contamination on a regular basis and were maintained in McCoy's 5 A (Lonza, Basel, Switzerland) supplemented with 10% Fetal Bovine Serum (FBS) (Sigma-Aldrich, St Louis, MO, USA), L-glutamine and antibiotics (Lonza). mIECs were maintained in Dulbecco's Modified Eagle Medium (Capricorn Scientific) supplemented with 10% FBS (Gibco), Penicillin-Streptomycin Mixture (Lonza) and L-Glutamine (Capricorn Scientific).

For the generation of ex-vivo organoids, around 10 cm of duodenum samples from 8–10 weeks old healthy (i.e. wild type Apc-expressing) or 100 days old $Apc^{+/Min}$ mice were used for the generation of ex-vivo organoids. Intestinal contents were flushed away with PBS. Intestines were opened longitudinally and remaining intestinal contents were removed. Villus were removed by cover slip, and remaining parts were cut into 5 mm length pieces and kept in PBS on ice. All next steps were carried out as soon as possible under a cell culture hood. Samples were washed with ice-cold sterile PBS 3 times, then washed once at room temperature with sterile HBSS medium without calcium and magnesium (Gibco) supplemented with 30 mM EDTA. Samples were then incubated in 10 ml of fresh HBSS-EDTA medium for 5 minutes at room temperature on a rolling machine, vortexed for 5 seconds and supernatants were kept on ice as fraction 1. Samples were incubated a second time with 10 ml of fresh HBSS-EDTA medium for 5 minutes at room temperature on a rolling machine, vortexed for 15 seconds and supernatants were kept on ice as fraction 2. Samples were incubated a third time with 10 ml of fresh HBSS-EDTA

medium for 5 minutes at room temperature on a rolling machine, vortexed for 15–30 seconds and supernatants were kept on ice as fraction 3. All fractions were checked under the microscope. Fractions with crypts (theoretically fractions 2 and 3) were passed through a 70 mm cell strainer into 50 ml falcon tubes filled with 10 ml of 1% BSA-PBS. Cell strainers were washed twice with ice-cold PBS. Samples were then centrifuged at $200 \times g$ for 5 minutes at 4 °C. Supernatants were removed and pellets were washed with ice-cold PBS twice. After the second wash, pellets were transferred into 15 ml falcon tubes and washed once with PBS. Pellets were then resuspended in Matrigel® (Corning) and seeded on 24-well culture plates. Plates were incubated at 37 °C for 15–20 minutes to solidify Matrigel® and IntestiCult™ Intestinal Organoid Growth Medium (Stemcell) was added (for the generation of ex-vivo organoids from healthy mice). Ex-vivo organoids from duodenum of Apc$^{+/Min}$ mice were maintained in DMEM/F12 supplemented with Penicillin/Streptomycin, 1× B27 and 1× N2 supplements, EGF (20 ng/ml), Noggin (100 ng/ml). They were passaged every 6–7 days.

## Mouse strains and treatments

The Stard7$^{Lox/lox}$ strain (Stard7$^{tm1a(EUCOMM)Wtsi}$) was obtained from the Wellcome Trust Sanger Institute. Both exons 2 and 3 were floxed with Loxp sites, as described in the following link: https://www.ncbi.nlm.nih.gov/nucleotide/JN950450.1?report=genbank&log$=nuclalign&blast_rank=2&RID=KF0V8WPC016. Apc$^{+/Min}$ mice were obtained from Jackson Laboratory (Strain #:002020) as was also the Villin-CRE strain (B6.Cg-Tg(Vil1-cre)997Gum/j, Strain #: 004586). Villin-Cre-ER$^{T2}$ Ctnnb1$^{+/lox(ex3)}$ (β-cat$^{c.a.}$) mice were previously described (Harada et al, 1999; el Marjou et al, 2004). Villin-Cre-ER$^{T2}$Ctnnb1$^{+/lox(ex3)}$ mice were gavaged 5 consecutive days with 1 mg of tamoxifen (Sigma, St Louis, MO, USA) to induce β-Catenin activation in enterocytes, as described previously (Göktuna et al, 2014). All mouse strains were housed at the animal facility of the University of Liege, according to rules requested by the ethical committee. Cages were ventilated, softly lit and subjected to a light dark cycle. The relative humidity was kept at 45 to 65%. Mouse rooms and cages were always kept at a temperature range of 20–24 °C.

For mice treatments, a Rapamycin (LC Laboratories Cat#R-5000) stock solution was prepared at 50 mg/mL in Ethanol 100%. For mice injections, a freshly prepared working solution of Rapamycin at 2 mg/ml was made with a final concentration of Ethanol 4%, PEG400 5%, and Tween 80 5%. 105 days old Apc$^{+/Min}$ mice were daily treated with intraperitoneal injections of Rapamycin for 20 days (4 mg/kg) to assess consequences on tumor sizes. For western blot analyses, mice were treated with Rapamycin (8 mg/kg) for 7 days. For NAC (Sigma Cat#A9165) treatments, a 1 g/L solution in drinking water (pH 7–8) ad libero was given to 100 days old Apc$^{+/Min}$ mice for 3 weeks. The solution was also changed every 3 days. For antibiotics treatments of 100 days old Apc$^{+/Min}$ mice, 0.5 g Ciprofloxacin, 1 g Ampicillin and 0.5 g Metronidazole per liter were added in the drinking water for 3 weeks. The solution was changed every 3 days. For all treatments, control animals were treated with the vehicle alone. For AOM/DSS treatments, mice were 8–10 weeks old. A single dose of AOM (10 mg/kg) (Santa Cruz, sc-358746) was given at day 0. A DSS (MP Biochemicals Cat #9011-18-1) 2% solution in the drinking water was given for 5 days 3 times at weeks 2, 5 and 8 (see Fig. 2A for

details). Mice were sacrificed 11 weeks after the single AOM injection.

For single DSS treatments, 8–12 week old males were treated or not with a 3% DSS solution in the drinking water for 6 consecutive days. Mice were observed and checked daily according to ethical protocols. Mice were sacrificed after the DSS treatment.

## Tissue processing and intestinal epithelial cell isolation

After euthanasia, small intestine and colon were extracted from mice, washed with PBS and cut longitudinally. In all, 5–10 cm of the proximal tissue (duodenum) was rolled and used for histological purposes and 5 cm was used for intestinal epithelial cell extraction. In brief, intestine was incubated twice for 10 minutes at 37 °C in a HBSS-EDTA buffer (30 mM). Cells were harvested by intensive vortexing and next centrifuged for 5 minutes at 4 °C at 1500 rpm, washed twice in PBS and snap frozen.

## Lentiviral vector production and cell transduction

Transfection mixture was prepared in 800 μL OptiMEM (Gibco) by adding following plasmids: 12 μg of psPAX2 (AddGene #12260), 5 μg of pVSV-G (AddGene #138479) and 12 μg of expression vector carrying shRNA sequence, in the presence of 80 μL of TransIT-LT1 Transfection Reagent (Mirus, Mir2360). Expression vectors were: pLVshCtrl (Sigma, SHC002), pLVshSTARD7#2 (Sigma, TRCN000028081), pLVshSTARD7#4 (Sigma, TRCN0000155648). A list of other shRNAs used in this study is provided in Table EV1. After 15 minutes of incubation at room temperature, the transfection mixture was added dropwise on top of the 60–70% confluent LentiX cells on T75 cm³. Medium was changed within 6–16 hours, with 8 mL of a complete culture medium. After 72 hours of viral particle production, medium with LV was collected, centrifuged at $800 \times g$ for 10 minutes, 0.22 μL filtered and either added directly to cells to be transduced or frozen at −80 °C.

For cell transduction, on day 0 (D0), cells were transduced using the medium containing lentiviral particles in the presence of 8 μg/mL of Polybrene transfection reagent (Millipore). After 24 hours (D1), the medium was changed to the selection medium with 1 μg/mL Puromycin (InVivoGen) for 48 hours. Afterwards (D3-D4), cells were split using the Trypsin/EDTA solution (BioWest) and seeded for experiments. All experiments were done on D5-D6 post transduction.

## Transcriptomic analyses by RNA sequencing

RNA sequencing was performed on libraries prepared with total RNA samples from the mouse intestinal epithelium (colon) of 100 days old Apc$^{+/Min}$/Stard7$^{Control}$ mice and Apc$^{+/Min}$/Stard7$^{ΔIEC}$ mice ($n = 3$) and from the mouse intestinal epithelium of AOM/DSS-treated Stard7$^{Lox/lox}$ ($n = 6$) and Stard7$^{ΔIEC}$ mice ($n = 5$). Total RNAs were extracted using the TriPure isolation reagent (Roche, cat#11667165001) according to the manufacturer's protocol. Total RNAs were treated with TURBO™ DNase (ThermoFisher Scientific, AM2238). RNA integrity was verified on a Bioanalyser 2100 with RNA 6000 Nano chips (Agilent Technologies). RNA integrity number score was above 7 for every sample. Libraries were prepared using Truseq® stranded mRNA Sample Preparation Kits (Illumina) following manufacturer's instructions. Libraries were

validated using QIAxcel Advanced System and quantified by qPCR using the KAPA library quantification kit. Libraries were multiplexed and sequenced on an Illumina NextSeq500 sequencer to generate more than ~25,000,000 paired-end reads ($2 \times 150$ bases) per library. Raw reads were demultiplexed and adapter-trimmed using Illumina bcl2fastq conversion software v2.20. Reads were processed within the nf-core/rnaseq-1.4.2 pipeline (Ewels et al, 2020) using STAR aligner, the mouse reference GRCm39 and the gene annotations from Ensembl release 107. Quality of the sequencing data was successfully controlled using QC modules of the pipeline and a report has been compiled with MultiQC.

## Real-time PCR analyses

Total RNAs were extracted from cultured cells using the column based extraction E.Z.N.A.® Total RNA Kit I (Omega Biotek) according to manufacturer's protocol and from the mouse intestinal epithelium as described here before. cDNAs were obtained using the RevertAid H Minus First Strand cDNA Synthesis Kit using oligo(dT)18 primers and 1 µg of total RNAs as a template. Real-time PCR reactions were carried out on Light Cycler480 (Roche) with TB Green Premix Ex Taq II Tli RNase H Plus (Takara Bio) and specific primers designed with the PrimerBlast software (NCBI). Ct values were used to calculate fold change of expression using the $2^{-\Delta\Delta Ct}$ method. GAPDH was used as the housekeeping control. Primer sequences can be found in Table EV2.

## Exome sequencing

Intestinal tumors from 100 days old Apc$^{+/Min}$/Stard7$^{\Delta IEC}$ and Apc$^{+/Min}$/Stard7$^{Lox/lox}$ mice ($n = 8$) as well as from IECs of 10 weeks old Stard7$^{\Delta IEC}$ and Stard7$^{Lox/lox}$ mice ($n = 8$) were used to establish the mutational signature in all genotypes. Samples were prepared according to instructions provided by BGI. Samples were processed and all steps of exome sequencing experiments were carried out by BGI. For bioinformatical analyses, data were filtered to remove adaptors, contamination and low-quality reads from raw reads. Reads were then aligned to the mouse reference genome (GRCm38) using the BWAsoftware. The sequencing quality, including data production, statistics, sequencing depth distribution and coverage uniformity was assessed. SAMtools, SOAPsnp, or GATK were used for SNP calling. SNPs were annotated to the corresponding gene functional units in RefSeqGene database, including nucleotide and amino acid changes. SNP validation and comparison was done with dbSNP database, 1000 Genomes Project database, publicly available exome databases (ESP), ENCODE, ClinVar, GWAS, PVFD* and BGI-GaP*. Statistics of SNPs in each functional element was done. SAMtools or GATK were used for InDel calling. Each InDel was annotated to the corresponding gene functional units in RefSeq-Gene database, including nucleotide and amino acid changes. InDel validation and comparison was carried out with dbSNP database, 1000 Genomes Project database, publicly available exome databases (ESP), ENCODE, ClinVar, GWAS, PVFD* and BGI-GaP*. The statistics of InDels in each functional element was finally done.

## SDS-PAGE and western blot analyses

Cells were washed twice in ice-cold PBS and scraped in 1% SDS lysis buffer supplemented with cOmplete Protease Inhibitor (Roche) and PhosStop (Roche). Protein concentrations were measured using the Pierce BCA Protein Assay Kit (Thermo Scientific) according to manufacturer's instructions. Samples were denatured by boiling for 7 minutes in a Laemmli Buffer with β-mercaptoethanol and 20 µg of total proteins were loaded per well of SDS-PAGE gel (8–14%). Samples were separated under reducing conditions and transferred to PVDF membrane (Immobilon-P, Millipore), using wet transfer chambers. Membranes were blocked in 5% skimmed milk for 45 minutes, cut and incubated with primary antibodies overnight on rotor at 4 °C. The following day, membranes were washed five times in TBS-T and incubated for 1 hour at room temperature with HRP conjugated secondary antibodies donkey anti-rabbit (GE Healthcare, NA934V) or sheep anti-mouse (GE Healthcare, NA931V). After subsequent five washing steps, membranes were visualized with the Pierce ECL Western (Thermo Scientific) or SuperSignal West Femto Maximum Sensitivity Substrate (Thermo Scientific). A list of primary antibodies is available in the reagents and tools table. The ImageJ/Fiji program was used for densitometry analyses (see the Appendix file entitled « Densitometry analyses of western blots »). Data was exported to Excel and the protein of interest/Hsp90 ratio was calculated.

## Histological and immunohistochemical analyses

STARD7-IHC-based Tissue microarray analyses (TMA) were done at the University Hospital of Cologne using the anti-STARD7 antibody (PA5-30772, Thermo Fisher Scientific), as illustrated in Fig. EV4A. Slides were scanned with a Panoramic 250 slide scanner. Morphometric quantification was carried out using the ImageJ program (National Institutes of Health, Bethesda, USA). Briefly, samples from human patients were categorized into three different groups: normal ($n = 32$), dysplasia ($n = 37$), and carcinoma ($n = 32$). The ratio of StarD7-positive signals versus whole background was calculated in each sample, and two-tailed Student's $T$ tests were utilized for statistical analysis, with $P < 0.05$ considered significant.

For histological analyses illustrated in Figs. 2B and 4E–G, organs were fixed in a 4% paraformaldehyde (PFA) solution for 24 hours. Histological examination was performed on paraffin-embedded sections stained with hematoxylin, eosin (H&E) and Alcian Blue (BA). The grading of adenoma and adenocarcinoma was determined based on the histology score, as described (Paul et al, 2022). Representative images are provided in Fig. EV6. The areas of normal epithelium, high and low grade adenomas were quantified using the QuPath software. Low grade dysplasia was characterized by hyperchromatic nuclei, nuclear stratification and elongated cell morphology, whereas high grade dysplasia exhibited marked nuclear hyperchromatic, increased nuclear pleomorphism, and loss of cell polarity. The areas of high grade adenoma, low grade adenoma, and normal epithelium were calculated as a percentage of the total analyzed area. Statistical comparisons were performed using a Student's $T$ test.

For immunohistochemistry analyses, following deparaffinization, tissue sections were autoclaved for 11 minutes at 126 °C in Tris/EDTA buffer/0.01% Tween20 (pH 9) for antigen retrieval. Endogenous peroxidase activity was blocked by incubating the slides in 3% $H_2O_2$ for 10 minutes. Slides were then incubated with a blocking solution containing 2.5% goat serum for 30 minutes at room temperature (RT). Sections were subsequently incubated with

an anti-Ki67 SolA15 antibody (1:400, 14-5698-82, Invitrogen/Bioscience) for 1 hour at room temperature. After washing in PBS, the secondary reaction was performed using the Rat ImmPRESS® HRP (Vector Laboratories) kit according to the manufacturer's instructions. Positive cells were visualized using the DAB Substrate Kit HRP (Vector Laboratories) and the samples were counterstained with hematoxylin. The percentage of positive cells was determined through computerized image analysis with QuPath 0.4.3. Statistical comparisons were performed using a Student's T test.

For the quantification of 4-HNE in colon sections, tissues were stained with DAB and counterstained with hematoxylin. Positive cells were identified using QuPath (v0.4.4) with the "Positive Cell Detection" algorithm. Epithelial regions were manually annotated and cells were classified as positive or negative based on DAB optical density.

## Immunofluorescence analyses

For immunofluorescences with TMRE (Tetramethylrhodamine, ethyl ester) and ER-Tracker™ green, control or STARD7-depleted mIECs cells were seeded on Cellview cell culture dishes (35 mm, 4 compartments) (Greiner bio-one, Cat#627870) overnight in the growth media. Cells were washed once with PBS then once with HBSS and incubated for 30 minutes at room temperature with 50 nM TMRE (Invitrogen, Cat#T669), 1 μM ER-Tracker™ green (Invitrogen, Cat# E34251) and 1 μM Hoechst 33342 (Thermo Scientific, Cat# 62249) in HBSS covered from light. Cells were subsequently washed three times for 5 minutes at room temperature in PBS. All samples were acquired with a LSM980 Airyscan 2 super-resolution system (Carl Zeiss, Oberkochen, Germany), in SR mode (pixel size 0.035 μm), equipped with a Plan-Apochromat ×63/1.4 oil objective. We imaged TMRE (red), ER-tracker ™ green and Hoechst (blue) in 2D, respectively using a 561 nm laser at 1.5%, a longpass filter LP570 and gain at 600 V; a 488 nm laser at 4.0%, a 495–550 nm bandpass filter and gain at 600 V and finally, a 405 nm laser at 4.0%, a shortpass filter SP505 nm and gain at 700 V. We acquired between 6 and 10 random field images, for a total of at least 100 cells per sample. The images were analyzed using the QuPath software, as described (Bankhead et al, 2017). All cells were manually annotated and the Mander's coefficient was calculated for each cell, using the script originally developed by MicroscopyRA (available at: https://gist.github.com/Svidro/68dd668af64ad91b2f76022015dd8a45#file-colocalization-of-channels-per-detection-0-2-0-groovy, which we adapted to our dataset). We analyzed the co-localization between channels 1 and 2 (TMRE for mitochondria and ER Tracker, respectively). To remove the background signal, the following thresholds were applied: channel 1 Background = 200 and channel 2 Background = 900.

## Proteomic analyses

Pellets from IECs were resuspended in 50 μl 6 M guanidine hydrochloride 100 mM Tris pH 8.5 containing 1.5 mg/mL TCEP and 1 mg/mL chloroacetamide and digested with 1 μg LysC (FUJIFILM Wako Pure Chemicals U.S.A. Corporation) for 4 hours at 37 °C. Subsequently, samples were diluted to 300 μl with LC-MS Water and digested overnight using 1 μg of porcine Trypsin (Thermo Scientific) followed by desalting using stage-tips before MS analyses. Peptides were resuspended in 0.1% TFA in Water and

peptide content was estimated 280 nm Absorption using a Nanodrop 2000 (Thermo Scientific). 1 μg was then injected and separated on an Ultimate 3000 Nano using a C18 packed emitter (Aurora, IonOptiks, Australia), with a gradient from 4 to 29% acetonitrile in 90 min, with a 10 minutes 80% wash. In total, 0.5% acetic acid was present throughout. Peptides were analysed in data-independent acquisition (DIA) mode on a Thermo Fusion Lumos. The mass spectrometer was operated in DIA mode, acquiring a MS 350–1650 Da at 120 k resolution followed by MS/MS on 45 windows with 0.5 Da overlap (200-2000 Da) at 30 k with a NCE setting of 27. Data was searched using DIA-NN (1.8.1) against the Uniprot Human database using the default setting for library-free search.

## Biochemical fractionations

Organelle-enriched protein extracts were isolated as previously described (Wieckowski et al, 2009). The entire small intestine (duodenum, jejunum and ileum) was washed with PBS, cut longitudinally and collected in cold PBS into 50 ml tubes. The intestine was then cut in ~1 cm pieces and put into HBSS/EDTA solution for 15 minutes at 37 °C with gentle shaking. Tissue was vortexed for maximum 30 seconds and the supernatant containing intestinal epithelial cells was collected into new 50 ml tubes. Cells were centrifuged at 1200 rpm for 5 minutes and the resulting supernatant was discarded. Cells were washed twice with PBS (without $Ca^{2+}$ and $Mg^{2+}$) and centrifuged at 600× g for 5 minutes at 4 °C. Cells were again centrifuged at 600× g for 5 minutes at 4 °C. PBS was discarded and cells were resuspended in 10 ml of ice-cold buffer containing 225 mM mannitol, 75 mM sucrose, 0.1 mM EGTA and 30 mM Tris-HCl pH 7.4. Cells were homogenized with cold Teflon homogenizer and cell integrity was monitored under the microscope (80–90% of cell damage has been reached). The homogenate was centrifuged at 600× g for 5 minutes at 4 °C. The pellet was discarded and the homogenate was again centrifuged at 600 g for 5 minutes at 4 °C. The supernatant was again centrifuged at 7000× g for 10 minutes at 4 °C. The supernatant was a cytosolic fraction with lysosomes and microsomes, while the resulting pellet contained mitochondria. The supernatant was further processed for separation of cytosolic, lysosomal and ER fractions. To achieve this goal, the supernatant was centrifuged at 20,000× g for 30 minutes at 4 °C. After centrifugation, the pellet consisted of lysosomal and plasma membrane fractions. Next, the centrifugation of the obtained supernatant at 100,000× g for 1 hour allowed the isolation of the ER (pellet) and the cytosolic fraction (supernatant). The crude mitochondria pellet was resuspended in 20 ml of cold buffer containing 225 mM mannitol, 75 mM sucrose and 30 mM Tris-HCl pH 7.4 and centrifuged at 7000× g for 10 minutes at 4 °C. The supernatant was discarded and the pellet was resuspended in 20 ml of ice-cold buffer containing 225 mM mannitol, 75 mM sucrose and 30 mM Tris-HCl pH 7.4 and centrifuged at 10,000× g for 10 minutes at 4 °C. The mitochondria pellet was resuspended in 2 ml of ice-cold buffer (250 mM mannitol, 5 mM HEPES (pH 7.4) and 0.5 mM EGTA) and layered on 8 ml of Percoll medium. The same solution (250 mM mannitol, 5 mM HEPES (pH 7.4) and 0.5 mM EGTA) was gently layered on the top to fill up the centrifuge tube and was centrifuged at 95,000× g for 30 minutes (Beckman Coulter Optima L-100 XP Ultracentrifuge (SW40 rotor,

Beckman, Fullerton, CA, USA). The band containing purified mitochondria was localized at the bottom of the ultracentrifuge tube. Mitochondria-associated membranes (MAMs) were visible as a diffused white band in the middle of the tube. MAMs were collected with a Pasteur pipette and then diluted ten times with a buffer containing 250 mM mannitol, 5 mM HEPES (pH 7.4) and 0.5 mM EGTA. MAMs were centrifuged at 6300× $g$ for 10 minutes at 4 °C (to remove mitochondria contamination). The supernatant was then centrifuged at 100,000× $g$ for 1 hour at 4 °C. The resulting pellet containing MAMs was resuspended and used for western blot analyses.

## ROS quantification

Control or STARD7-depleted cells were incubated for 10 minutes at 37 °C in a carboxy-H2DFFDA-containing solution at a concentration of 24 µM (Invitrogen). The medium was then removed. Cells were trypsinized and centrifuged at 1500 rpm for 5 minutes at room temperature. The supernatant was subsequently removed and cells were washed in PBS once and resuspended in 100 µl of PBS. Cells were analysed on FACS Canto II and the data were generated using the FlowJo program. For all FACS analyses, the gating strategy used was forward and side scatter gating to remove debris and other events of non-interest (doublets) while preserving cells based on size and or complexity.

## Lipidomic analyses

In total, 200 µL of isopropanol was added to the cell pellet, sonicated on ice-cold bath for 10 minutes, vortexed for 10 minutes at 4 °C and centrifuged for protein precipitation. The supernatants containing lipid extracts were then individually transferred to new vials and separated on a Dionex UltiMate 3000 LC System (Thermo Scientific, Waltham, Massachusetts, EUA) using a Kinetex C18 EVO 2.6 µm, 100 A, 150 × 0.3 mm LC Column. Mobile phases consist of (A) H$_2$O—5 mM ammonium formate, 0.1% Formic acid, (B) 60% Acetonitrile: 40% Methanol—5 mM ammonium formate, 0.1% Formic acid, and (C) Isopropanol—5 mM ammonium formate, 0.1% Formic acid. Detailed information on the gradient (time and Flow) are available upon request. Mass spectrometry data were acquired on a Q Exactive™ Plus Hybrid Quadrupole-Orbitrap™ Mass Spectrometer (Thermo Scientific, Waltham, Massachusetts, EUA) using data dependent acquisition in positive and negative ion mode. The parameters used were: Full MS: Resolution: 70,000; AGC target: 1e6; Max IT: 100 ms; Scan range: 90 to 1350 $m/z$. dd-MS$^2$: Resolution: 17,500; AGC target: 1e5; Max IT: 120 ms; Loop count: 5; TopN: 5; Isolation window: 1.5 $m/z$. Data were processed using compound discovery (Thermo Scientific, Waltham, Massachusetts, EUA) and lipids identification was performed using Lipidex (Hutchins et al, 2018). For lipidomic analyses using subcellular fractions (Fig. EV3B), data were analysed in 2 ionization modes (negative and positive ion modes).

## Complex I enzyme activity

The Microplate Assay Kit (Colorimetric) from Abcam (Cat#ab 109721) was used to measure Complex I activity, according to the manufacturer's protocol.

## Measurement of ATP production

The quantification of ATP levels was performed according to the protocol of the ATP assay kit from Sigma-Aldrich (Cat#MAK473). Around 2 × 10$^4$ cells were seeded in 96-well plates and 90 µl of Reaction Mixture was added in each well the next day. After gentle mix and tapping, the plate was read using a TriStar$^2$ LB942 multimode reader. The concentration was determined according to a standard curve as follows: ATP (µM) =($R$sample − $R$blank/$Slope$ ($µM^{-1}$))$x$ $DF$ where RSample = Luminescence (RLU) value of the sample, RBlank = Luminescence (RLU) value of the blank, DF = Sample dilution factor (DF = 1 for undiluted samples) obtained from ATP standards according to the manufacturer's recommendation. Resulting data were normalized to protein concentrations for each sample.

## Extracellular flux assays

Oxygen consumption rate (OCR - pmol/min) measurements were performed with a Seahorse XF96 cell culture microplates (Agilent, Santa Clara, CA). For mitochondrial OCR (pmol/min) analysis, control and Stard7-depleted mIECs seeded and allowed to adhere overnight. Twenty-four hours before the assay, cells were starved in DMEM containing 0.5 mM glucose, 1 mM glutamine, 1% FBS, and 0.5 mM L-carnitine. On the day of the assay, culture medium was replaced with Seahorse XF assay medium supplemented with 2 mM glucose and 0.5 mM L-carnitine, adjusted to pH 7.4. Palmitate (0.17 mM, conjugated to BSA) was added immediately before starting the assay. Fatty acid oxidation was assessed using the Seahorse XF Analyzer with sequential injections of Etomoxir (5 µM), Oligomycin (1 µM), FCCP (1 µM), and Rotenone/Antimycin A (0.5 µM). All results were normalized according to the cell number evaluated by Hoechst 33342 (2 µg/mL) incorporation after cold methanol/acetone fixation followed by a well-scanning. Results shown are from three independent experiments carried out with triplicates. Two-sided statistical analysis was performed using one-way analysis of variance followed by Tukey's multiple comparisons.

## Transmission electron microscopy

Cells were fixed for 1 hour at 4 °C in a solution composed of 2.5% glutaraldehyde in 0.1 M Sorensen's buffer (0.2 M NaH$_2$PO$_4$, 0.2 M Na$_2$HPO$_4$, pH 7.4). After several washes in the same buffer, the samples were post-fixed for 60 minutes with 2% osmium tetroxide, washed in deionised water, dehydrated through graded ethanol (70, 95, and 100%) and embedded in epon for 48 hours at 60 °C. Ultrathin sections (700 Å thick) were obtained by means of an ultramicrotome (Reichert Ultracut E) equipped with a diamond knife. The ultrathin sections were mounted on palladium/copper grids coated with collodion and contrasted with uranyl acetate and lead citrate for 5 minutes each before being examined under a Jeol JEM1400 transmission electron microscope at 80 kV. Random fields were photographed using an 11-megapixel camera system (Quemesa, Olympus). Morphometric measurements were performed with iTEM v5.2 (Olympus, Tokyo, Japan) and analyzed using Image J v1.52a software. We took images of 18 randomly-selected cells from each experimental condition with a JEOL 1400 TEM at ×2500 magnification. We counted the number of

mitochondria (1536 in control cells and 1389 in STARD7-depleted cells) and calculated the number of mitochondria per $\mu m^2$ of cytoplasm, the mean mitochondrial area as well as the total mitochondrial area in the cytoplasm in each experimental condition.

## Targeted LC-MS profiling for intracellular metabolites

For metabolomics analysis, extracts from IECs of 100 days old Apc$^{+/Min}$/Stard7$^{Lox/lox}$ and Apc$^{+/Min}$/Stard7$^{\Delta IEC}$ mice ($n = 3$) were analysed. Each sample was washed three times with cold PBS, collected into an Eppendorf tube, frozen in liquid nitrogen and stored at -80 °C until extraction. The extraction solution used was 50% methanol, 30% ACN, and 20% water. The volume of extraction solution added was calculated from the cell count ($2 \times 10^6$ cells per ml). After addition of extraction solution, samples were vortexed for 5 minutes at 4 °C, and immediately centrifuged at 16,000× $g$ for 15 minutes at 4 °C. The supernatants were collected and analyzed by liquid chromatography–mass spectrometry using SeQuant ZIC-pHilic column (Merck) for the liquid chromatography separation. Mobile phase A consisted of 20 mM ammonium carbonate plus 0.1% ammonia hydroxide in water. Mobile phase B consisted of ACN. The flow rate was kept at 100 ml/min, and the gradient was 0 minute, 80% of B; 30 minutes, 20% of B; 31 minutes, 80% of B; and 45 minutes, 80% of B. The mass spectrometer (QExactive Orbitrap, Thermo Fisher Scientific) was operated in a polarity switching mode and metabolites were identified using TraceFinder Software (Thermo Fisher Scientific). To obtain a robust statistical analysis, metabolomics data were normalized using the median normalization method (Hendriks et al, 2007). The data were further pre-processed with a log transformation. The MetaboAnalyst 4.0 software (Xia et al, 2015) was used to conduct statistical analysis and heatmap generation, and enrichment analysis. The algorithm for heatmap clustering was based on the Pearson distance measurement for similarity and the Ward linkage method for biotype clustering. Metabolites with similar abundance patterns were positioned closer together. As part of the routine targeted LC-MS pipeline, mQACC-aligned QA/QC procedures were applied, including SOP-based extraction/storage/analysis, regular instrument maintenance, weekly system suitability tests, randomized/blinded runs, and multiple QCs (pooled interstudy QC, extraction blanks, system stability blanks, solvent blanks). QC performance showed no significant drift in measured metabolite levels. Relative values for all measured metabolites are provided in Table EV3.

## Establishment of the microbiota signature

Contents of cecum of 100 days old Apc$^{+/Min}$/Stard7$^{Lox/lox}$ and Apc$^{+/Min}$/Stard7$^{\Delta IEC}$ mice ($n = 6$) were used for analyses. Total cecum contents were squeezed into sterile cryotube and immediately snap-frozen in liquid nitrogen. Tools and gloves were washed between each mouse. Foil covers were changed each time. DNA was extracted using the Magnetic Soil and Stool DNA Kit (TianGen, China, Catalog #: DP712). The microbiota signature in both genotypes was established through 16S Amplicon Metagenomic Sequencing of the V4 region by Novogen. Briefly, 16S rRNA V4 region (16SV4) was amplified using specific primers (515F- 806 R) with a barcode. All PCR reactions were carried out with 1.5 µl of

Phusion® High - Fidelity PCR Master Mix (New England Biolabs); 0.2 µM of forward and reverse primers, and about 10 ng of template DNA. Thermal cycling consisted of initial denaturation at 98 °C for 1 minute, followed by 30 cycles of denaturation at 98 °C for 10 seconds, annealing at 50 °C for 30 seconds and elongation at 72 °C for 30 seconds and 72 °C for 5 minutes. Same volume of 1x loading buffer (contained SYB green) were mixed with PCR products and subjected to an electrophoresis on 2% agarose gel for detection. PCR products were mixed in equal density ratios. Then, mixture PCR products were purified with Universal DNA Purification Kit (TianGen, China, Catalog #: DP214). Sequencing libraries were generated using NEB Next® Ultra™ II FS DNA PCR-free Library Prep Kit (New England Biolabs, USA, Catalog #: E7430L) following manufacturer's recommendations and indexes were added. The library was checked with Qubit and Real-Time PCR for quantification and bioanalyzer for size distribution detection. Quantified libraries were pooled and sequenced on Illumina platforms, according to effective library concentration and data amount required. For bioinformatical analyses, sequences were processed using QIIME2 (version 2023.2) (Bolyen et al, 2019). The pipeline included Primer removal and Denoising using DADA2 to obtain the amplicon sequence variant (ASV) table (Callahan et al, 2016). Singletons (ASV present <2 times) were discarded. Sequences were clustered based on a 0.99-percentage identity and chimeras were removed using the UCHIME algorithm (implemented in QIIME's vsearch plugin). Taxonomic classification was performed using a pre-trained naive Bayes classifier implemented in QIIME2 against the SILVA 138 reference database (silva138_AB_V4_classifier.qza) (Quast et al, 2013). Reads classified as mitochondria, chloroplast or aquatic bacteria were filtered out while unassigned ASVs are retained. Taxa that could not be identified on genus-level are referred to the highest taxonomic rank identified. Further analyses were performed using a R-based pipeline developed for this project. Raw QIIME2-exported feature tables, taxonomy files and phylogenetic trees were imported into phyloseq and processed using standard R packages. Alpha-diversity indices (Shannon, Faith's phylogenetic diversity, Pielou's evenness and observed features) were calculated on rarefied data at a depth of 47,000 reads per sample, ensuring equal sampling effort across groups. Beta-diversity was assessed on non-rarefied data using multiple distance metrics (Aitchison, Bray–Curtis, Jaccard, weighted and unweighted UniFrac) and tested using PERMANOVA. To evaluate taxonomic differences between groups, we performed differential-abundance analysis using four complementary statistical frameworks: ANCOM-BC2, ALDEx2, DESeq2 and MaAsLin2. Analyses were conducted at the ASV level and, when indicated, after taxonomic agglomeration to higher ranks (phylum, family and genus). Statistical significance was evaluated using Benjamini–Hochberg false-discovery rate (FDR) correction.

## Statistical analyses

Statistical tests used in this study are described in figure legends when not mentioned in the methods. Investigators were not blinded. For animal studies, there were no randomization. Mice analysed were littermates and age-matched whenever possible. For sample size, statistical methods were used (see figure legends). Regarding inclusion/exclusion criteria, no animals were excluded from the analyses.

**The paper explained**

**Problem**

Colon cancer is the third most common cancer and second leading cause of cancer death worldwilde. There is an urgent need for new targets to inform the design of new therapeutic approaches to be used in combination with existing drugs. Defining new targets also relies on generating and characterizing mouse models of colon cancer that closely mimic human colon malignancies. In this context, more than 80% of clinical cases of colon cancer show a constitutive activation of the Wnt signaling pathway. However, Apc$^{+/Min}$ mice, in which this oncogenic pathway is constitutively activated, exhibit comparatively few tumors in the distal colon.

**Results**

Apc$^{+/Min}$ mice lacking the lipid transfer protein Stard7 in intestinal epithelial cells (Apc$^{+/Min}$/Stard7$^{\Delta IEC}$ mice) exhibit accelerated colon tumour formation and thus effectively mimics human colon malignancies. The microbiota signature found in the colon of these mice is also present in patients with colon cancer. While Stard7 acts as a tumor suppressor gene in Wnt-driven tumor development, it functions as an oncogenic candidate in a model of inflammation-driven cancer. Therefore, the mutational status of intestinal tumors must be considered when investigating the role of any gene candidate in cancer development.

**Impact**

Apc$^{+/Min}$/Stard7$^{\Delta IEC}$ mice are a suitable model for exploring the contribution of intestinal microbiota to tumour development in the distal colon.

## Study approval

The studies carried out with our mouse models of intestinal cancers were approved by the ethical committee of the University of Liege (Files 1842 and 2723).

## Data availability

Microbiota signature: The data source can be found at https://doi.org/10.5281/zenodo.10118008. Mass spectrometry proteomics data: ProteomeXchange Consortium via the PRIDE [1] partner repository with the dataset identifier PXD046922. RNA Sequencing datasets: GEO (accession number GSE247166). Lipidomic data: Data with extracts from Apc-expressing or Apc$^{+/Min}$ mice (Figs. 1C and 7A, respectively): https://massive.ucsd.edu/ProteoSAFe/private-dataset.jsp?task=119a004148af4b6ebcc96511dcd829f0. Data with extracts from subcellular fractions: Negative ion mode: https://massive.ucsd.edu/ProteoSAFe/private-dataset.jsp?task=286cd0bf273242a98b5a2ddbd671b7b5. Positive ion mode: https://massive.ucsd.edu/ProteoSAFe/private-dataset.jsp?task=7592f428e18c451987952f9e00240d3d.

The source data of this paper are collected in the following database record: biostudies:S-SCDT-10_1038-S44321-026-00409-5.

## Peer review information

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

## Acknowledgements

The authors are grateful to the GIGA Imaging platform for the IF and FACS analyses as well as to the GIGA Histology platform. AC is a Research Director at the FNRS and is supported by grants from WELBIO (WELBIO-CR-2019C-02R), EOS (program no. 40007505), FNRS/TELEVIE, the Belgian Foundation against Cancer (FBC/2020/1323), the University of Liege (ARC UBICOREAR) and by the Leon Fredericq Foundation (Faculty of Medicine, CHU of Liege). GR and AB are Senior Research Assistant and Research Associate at the FNRS, respectively. MVH and PC are Senior Research Associate and Research Director at the FNRS, respectively. PDC is honorary research director at FRS-FNRS (Fonds de la Recherche Scientifique) and recipients of grants from FNRS (Projet de Recherche PDR-convention: FNRS T.0030.21, CDR-convention: J.0027.22, FRFS-WELBIO: WELBIO-CR-2022A-02, EOS: program no. 40007505), ARC (action de recherche concertée: ARC25/30-151.

## Author contributions

**Kateryna Shostak**: Conceptualization; Resources; Data curation; Formal analysis; Validation; Investigation; Methodology. **Yu Chen**: Formal analysis; Validation; Investigation; Methodology. **Chloé Maurizy**: Formal analysis; Validation; Methodology. **Gilles Rademaker**: Data curation; Formal analysis; Methodology. **Xinyi Xu**: Formal analysis; Validation; Investigation; Methodology. **Arnaud Blomme**: Methodology. **Pierre Close**: Resources. **Olivier Renson**: Formal analysis; Validation; Investigation. **Matthias Van Hul**: Data curation; Formal analysis; Validation; Methodology; Writing—review and editing. **Patrice D Cani**: Resources; Data curation; Formal analysis; Methodology; Writing—review and editing. **Sebastian Klein**: Formal analysis; Validation; Investigation; Methodology. **Alexandra Florin**: Methodology. **Reinhard Büttner**: Formal analysis; Validation; Methodology. **Didier Cataldo**: Methodology. **Philippe Delvenne**: Validation; Methodology. **Ivan Nemazanyy**: Formal analysis; Validation; Methodology. **Caroline Wathieu**: Methodology. **Alexandre Hego**: Data curation; Validation; Methodology. **Sandra Ormenese**: Resources; Methodology. **Olivier Peulen**: Formal analysis; Validation; Methodology. **Marc Thiry**: Formal analysis; Validation. **Roopesch Krishnankutty**: Formal analysis; Validation; Investigation; Methodology. **Jair Marques Jr**: Formal analysis; Validation; Investigation; Methodology. **Alex von Kriegsheim**: Formal analysis; Validation; Investigation; Methodology. **Alain Chariot**: Conceptualization; Supervision; Funding acquisition; Validation; Investigation; Visualization; Methodology; Writing—original draft; Project administration; Writing—review and editing.

Source data underlying figure panels in this paper may have individual authorship assigned. Where available, figure panel/source data authorship is listed in the following database record: biostudies:S-SCDT-10_1038-S44321-026-00409-5.

## Disclosure and competing interests statement

PDC is inventor on patent applications dealing with the use of specific bacteria and components in the treatment of different diseases. PDC was co-founder of Enterosys.

# Expanded View Figures

**Figure EV1.  Stard7 is dispensable in intestinal homeostasis.**

(**A, B**) Cell proliferation in the intestinal epithelium is not regulated by Stard7. Anti-Ki67 immunofluorescence or immunochemistry analyses (**A, B**, respectively) were conducted in mouse small intestinal or colon sections from 8 to 12-weeks old Stard7$^{Lox/lox}$ ("WT") and Stard7$^{\Delta IEC}$ ("KO") mice (all males). The percentage of Ki67$^+$ cells was quantified in both genotypes ($n = 10$ mice for immunofluorescence analyses) (**A**). For immunohistological analyses (**B**), 5 mice per genotype were used and 2–3 fields per slide were randomly selected. 16 and 14 fields, which include 98 and 95 crypts from Stard7$^{\Delta IEC}$ and Stard7$^{Lox/lox}$ mice, respectively, were analysed (means ± S.D.). (**C, D**) Stard7 deficiency in the mouse small intestine and in the colon does not influence its architecture nor the number of goblet cells (**C, D**, respectively). Immunofluorescence analyses for Mucin 2$^+$ cells are illustrated (**C**). Sections from 3 WT and 3 KO mice were analysed. 5 random fields of both WT and KO mice were taken. 37 WT villi and 41 KO villi were analysed (means ± S.D.). Immunohistochemistry analyses (H&E stainings, top panels) showing the colon of the indicated genotypes are illustrated (**D**). Alcian blue stainings (lower panels) to visualize goblet cells in the colon of the indicated genotypes (8 weeks old mice) are illustrated. Alcian blue$^+$ cells were counted in 281 crypts/villi from 14 fields and in 340 crypts/villi from 15 fields (5 WT and 5 KO mice, respectively, means ± S.D., Student *T* test, ns = not significant). (**E**) Stard7 deficiency in the mouse small intestinal epithelium does not impact on Paneth cell differentiation, as assessed by immunofluorescence and western blot (WB) analyses for Lysozyme (left and right panels, respectively). Extracts from a total of 5 mice per genotype were analysed and one representative blot is illustrated. The arrow depicts the specific band for Stard7. For anti-Lysozyme immunofluorescences carried out with sections from 3 WT and 3 KO mice, 7 and 8 random fields were taken (WT and KO mice, respectively). 78 WT crypts and 99 KO crypts were analysed (means ± S.D.). (**F**) Markers of epithelial subtypes are properly expressed in the intestinal epithelium of Stard7$^{\Delta IEC}$ mice. Quantitative Real-Time PCR analyses were conducted with extracts from the intestinal epithelium of the indicated mouse genotypes. mRNA levels in one randomly selected Stard7$^{Lox/lox}$ mice were set to 1 and levels in other mice were relative to that after normalization with Gapdh mRNA levels ($n \geq 7$ for both genotypes, means ± S.D., *T* test with Welch correction, Stard7: **$P = 0.0033$; Lgr5: $P = 0.7593$; c-Myc: $P = 0.9974$; Cd44: $P = 0.5355$; Sox9: $P = 0.4429$; Dclk1: $P = 0.1437$; Bmi1: $P = 0.1999$; Olfm4: $P = 0.2518$; Epha2: $P = 0.2895$, ns = not significant. Source data are available online for this figure.

          

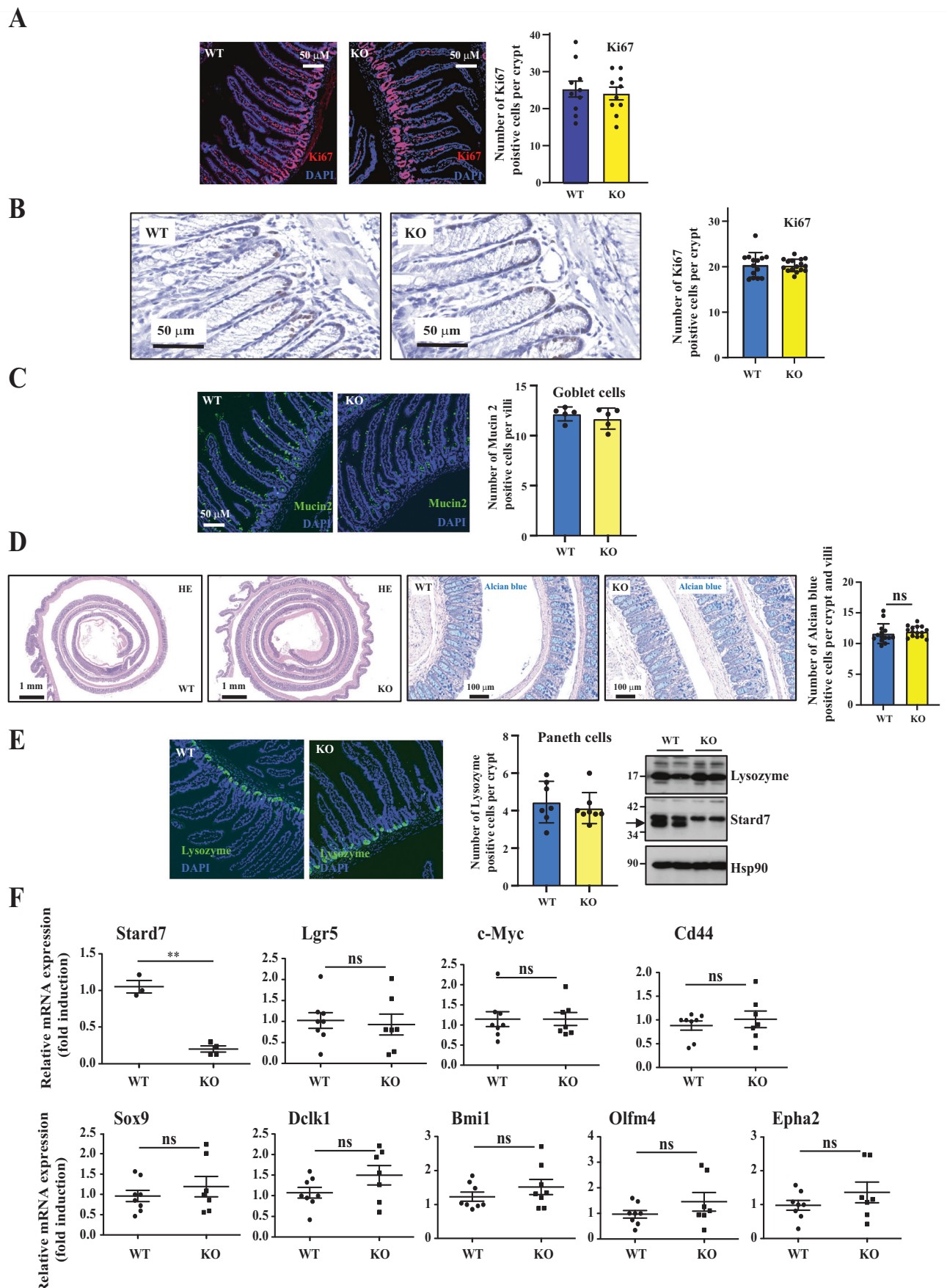

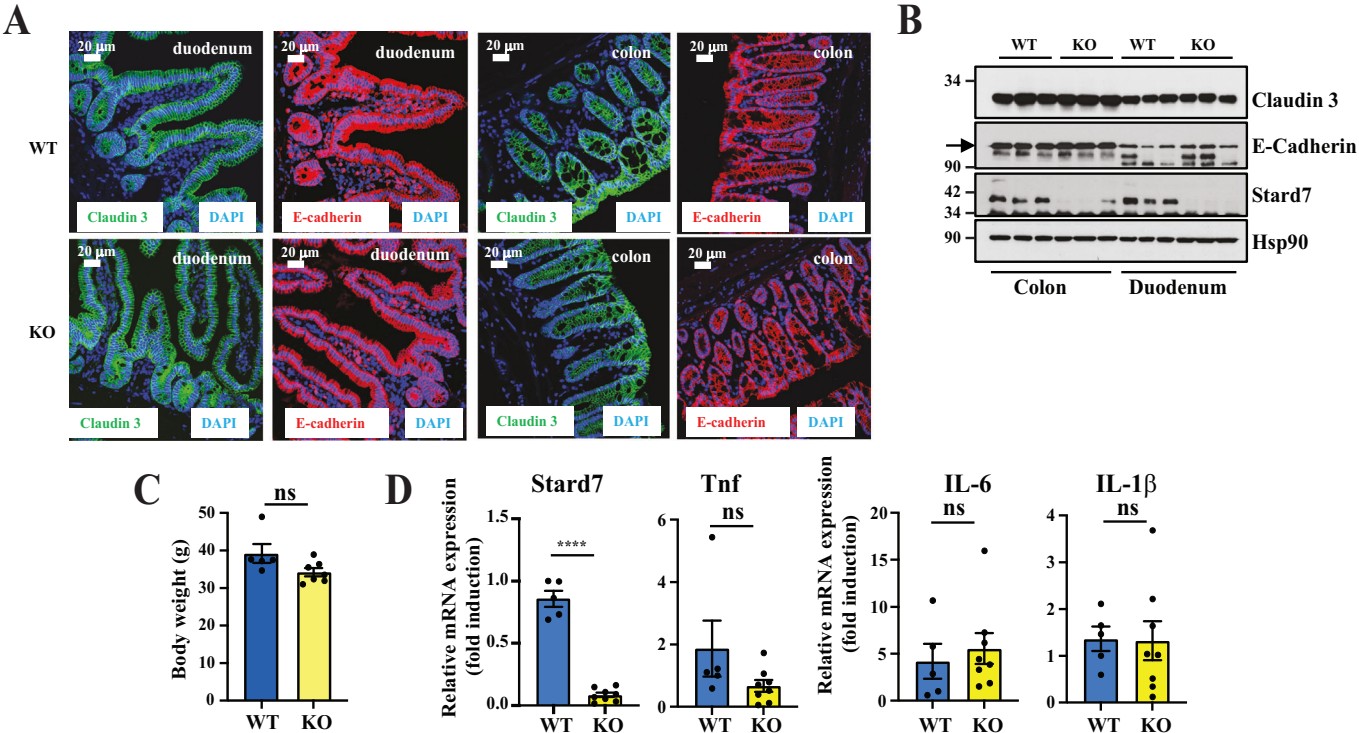

Figure EV2.  Stard7 deficiency does not impair the architecture of the intestine and does not lead to spontaneous inflammation.

(A, B) Stard7 deficiency does not impair the localization and expression levels of tight junctions proteins. Immunofluorescence analyses were conducted to detect the indicated proteins in the intestine of both Stard7$^{Lox/lox}$ and Stard7$^{ΔIEC}$ mice (A). Western blot analyses were also conducted with extracts from the intestinal epithelium (colon and duodenum) of both 8–12 weeks old Stard7$^{Lox/lox}$ and Stard7$^{ΔIEC}$ mice (3 WT males, 2 KO males and 1 KO female) (B). The arrow depicts the specific band for E-Cadherin. (C) Stard7 deficiency in the intestine does not change the body weight of aged mice. The body weight of 21-month-old mice of each indicated genotype was quantified ($n = 5$ and 8 for Stard7$^{Lox/lox}$ and Stard7$^{ΔIEC}$ mice (all males), respectively, means ± S.D., $T$ test with Welch correction, NS = no significance). (D) Stard7 deficiency in the intestine does not lead to any spontaneous inflammation. Quantitative Real-Time PCR analyses were conducted with extracts from the intestinal epithelium of the indicated mouse genotypes (21-month-old mice, all males). mRNA levels in one randomly selected Stard7$^{Lox/lox}$ mice were set to 1 and levels in other mice were relative to that after normalization with Gapdh mRNA levels ($n = 5$ and 8 for Stard7$^{Lox/lox}$ and Stard7$^{ΔIEC}$ mice, respectively, means ± S.D., $T$ test with Welch correction, Stard7: ****$P < 0.0001$; Tnf: $P = 0.2566$; IL-6: $P = 0.6957$; IL-1β: $P = 0.9350$, ns = not significant). Source data are available online for this figure.

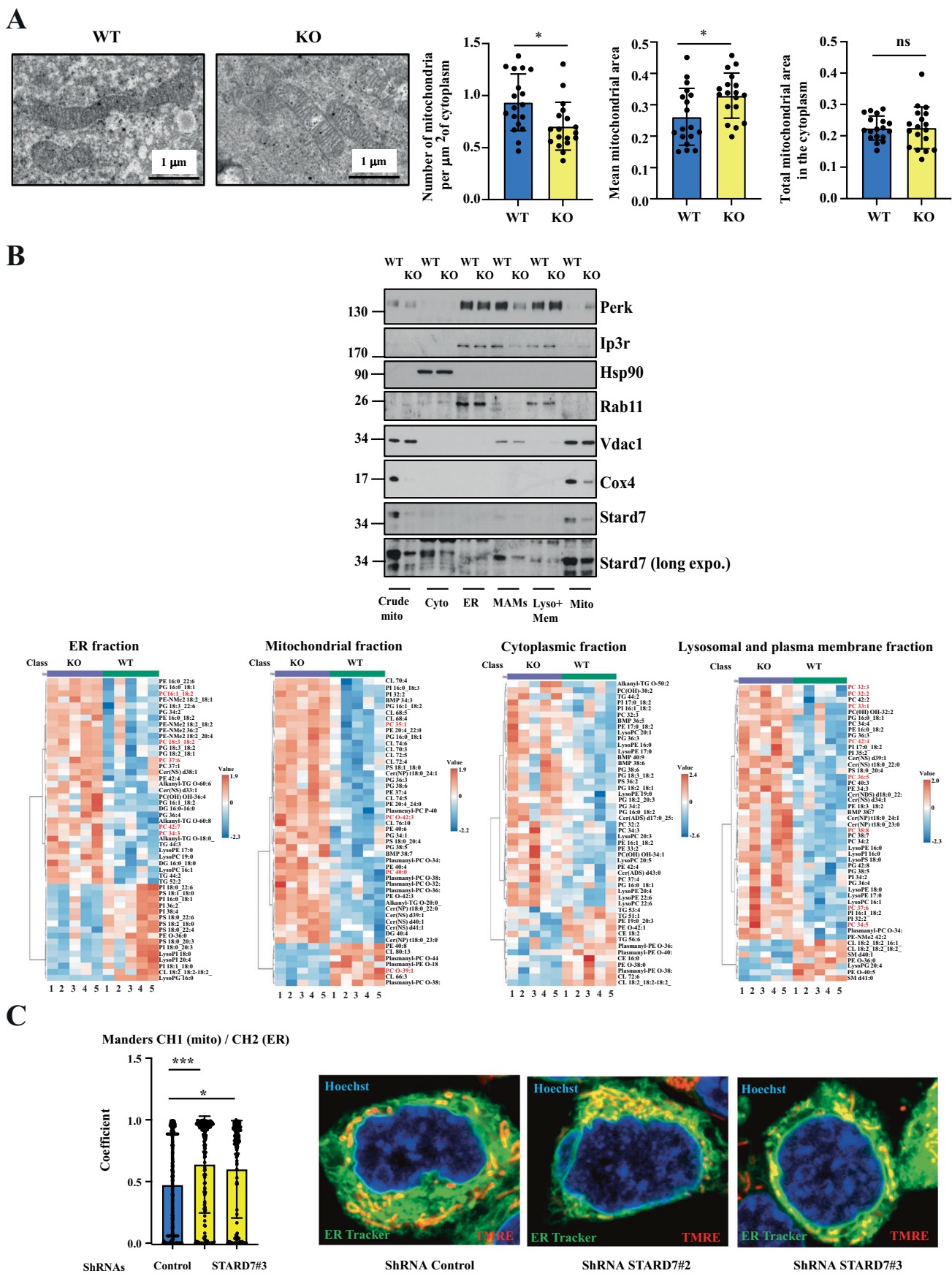

**Figure EV3. Dramatic lipids redistribution and enhanced mitochondrial-associated membrane contacts (MAMs) in intestinal epithelial cells lacking Stard7.**

(A) Intestinal epithelial cells lacking Stard7 show enlarged mitochondria. Tissues from 8-week-old Stard7^Lox/lox and Stard7^ΔIEC mice were used in transmission electronic microscopy to quantify mitochondrial areas in both experimental conditions (3 WT males; 2 KO males and 1 KO female). On the left, representative images in cells from Stard7^Lox/lox and Stard7^ΔIEC mice (WT and KO, respectively). On the right, the number of mitochondria per μm² of cytoplasm, the mean mitochondrial area as well as the total mitochondrial area in the cytoplasm were quantified in both experimental conditions (means ± SD, unpaired $T$ test with Welch correction. Histogram on the left: *$P = 0.0101$; Histogram on the middle: *$P = 0.0191$; Histogram on the right: $P = 0.9588$, ns = not significant). (B) Deregulation of the lipid landscape in organelles upon Stard7 deficiency. ER, mitochondrial, lysosomal and cytoplasmic extracts from the intestine of 8–12 weeks old WT and KO mice ($n = 5$ per genotype) (2 WT males, 3 WT females, 3 KO males, 2 KO females) were subjected to a lipidomic analysis. Phosphatidylcholine derivatives whose levels significantly change in KO versus WT mice are highlighted in red (Student $T$ test, $P < 0.05$, 5 mice per genotype). The quality of our organelle-specific protein extracts was assessed through western blot analyses (top panels). Extracts from two independent experiments were analysed and one representative western blot is illustrated. Crude mito = crude mitochondria, Cyto = cytoplasm, ER = endoplasmic reticulum, MAMs = mitochondria-associated membranes, Lyso + Mem = Lysosomal and plasma membrane fractions, Mito = mitochondria. (C) Enhanced MAMs in intestinal epithelial cells lacking Stard7. Immunofluorescence analyses were carried out in control versus Stard7-depleted mIECs to quantify the number of MAMs. On the left is illustrated the quantification of the Manders' coefficient M1 (fraction of mitochondria overlapping with the ER) in all experimental conditions. Analyses were carried out with 126, 165 and 123 control, STARD7#2 and STARD7#3 cells, respectively (means ± SD, Dunnett's multiple comparisons test. ShRNAs Control versus STARD7#2: ***$P = 0.0010$; ShRNAs Control versus STARD7#3: *$P = 0.0225$. Higher intensity of yellow corresponds to higher co-localization. Source data are available online for this figure.

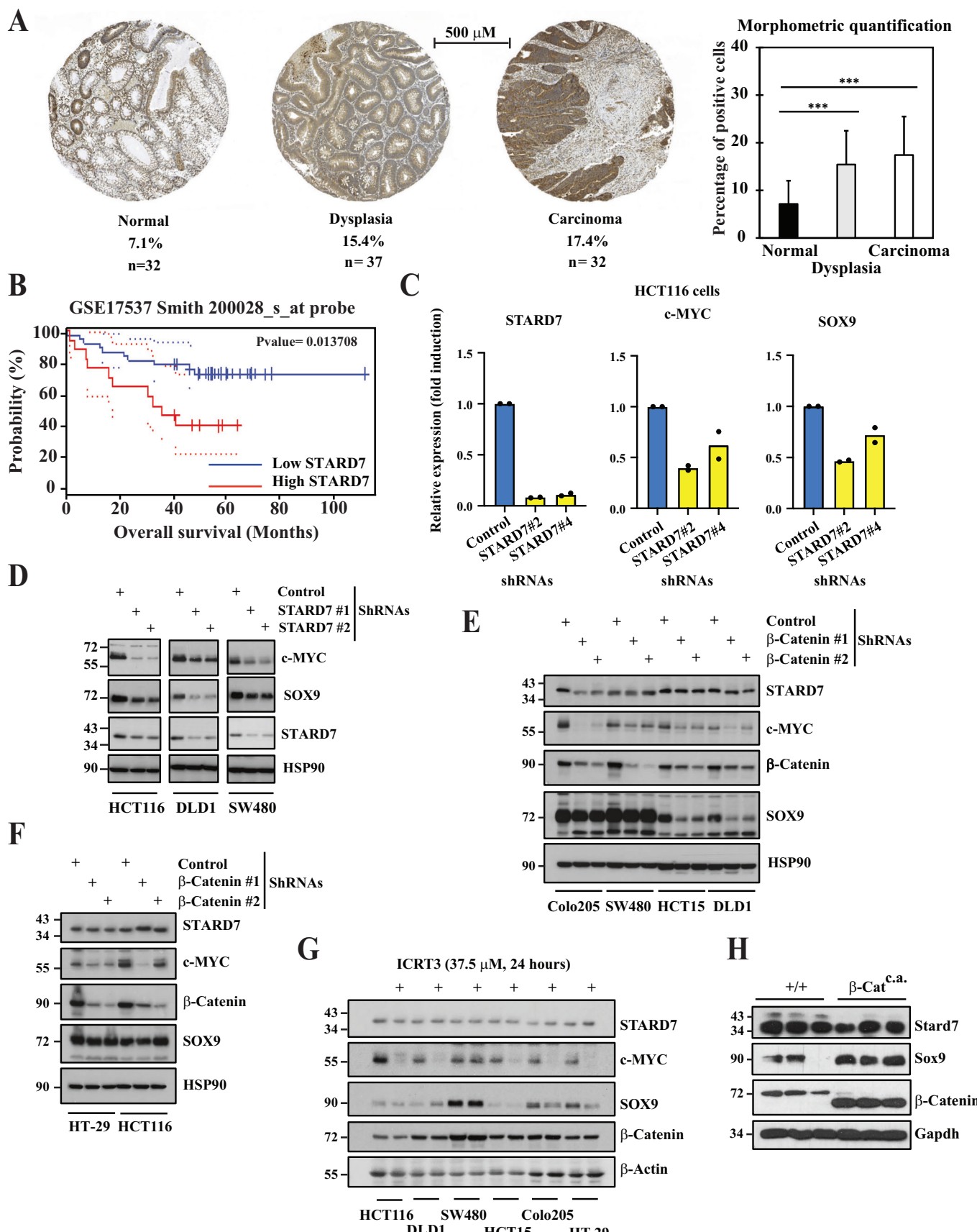

◀ **Figure EV4.  Enhanced expression of STARD7 in colon cancer.**

(A) Enhanced STARD7 expression in colon carcinomas. A morphometric quantification was carried out with immunohistochemistry (IHC) data obtained with clinical cases of normal intestinal tissues, dysplasia cases and carcinomas. The percentage of STARD7 positive cells is quantified in these three groups on the right (means + S.D., two-tailed Student T tests; Normal versus dysplasia: ****$P < 0.0001$; Normal versus Carcinoma: ****$P < 0.0001$). (B) Decreased survival for patients suffering from colon cancer showing high STARD7 mRNA levels (Kaplan–Meier survival plot). Vertical bars denote patients who were censored (i.e. patients still alive at last follow-up or lost to follow-up). (C) STARD7 controls levels of candidates induced by Wnt signaling. mRNAs extracted from control or STARD7-depleted HCT116 cells were subjected to quantitative Real Time PCRs to assess levels of STARD7, c-MYC and SOX9. Levels in control cells were set to 1 and levels in other experimental conditions were relative to that after normalization with GAPDH mRNAs. Data from two independent experiments performed in triplicates are shown. (D) STARD7 acts upstream of Wnt signalling. Protein extracts from control or STARD7-depleted colon cancer cell lines were subjected to western blot analyses using the indicated antibodies. Extracts from three independent experiments were analysed and one representative western blot is illustrated. (E, F) β-Catenin does not promote STARD7 expression in colon cancer cell lines. Extracts from control or β-Catenin-depleted cells were subjected to western blot analyses using the indicated antibodies. (G) Pharmacological inhibition of Wnt signalling does not downregulate STARD7 expression. Multiple colon cancer cell lines were treated or not with ICRT3 at the indicated concentration for 24 hours and extracts from the resulting cells were subjected to western blot analyses. (H) Stard7 expression is not upregulated in the mouse intestinal epithelium from mice showing constitutive Wnt signalling. Extracts from intestinal epithelial cells of control or β-Catenin[c.a.] mice were subjected to western blot analyses. Source data are available online for this figure.

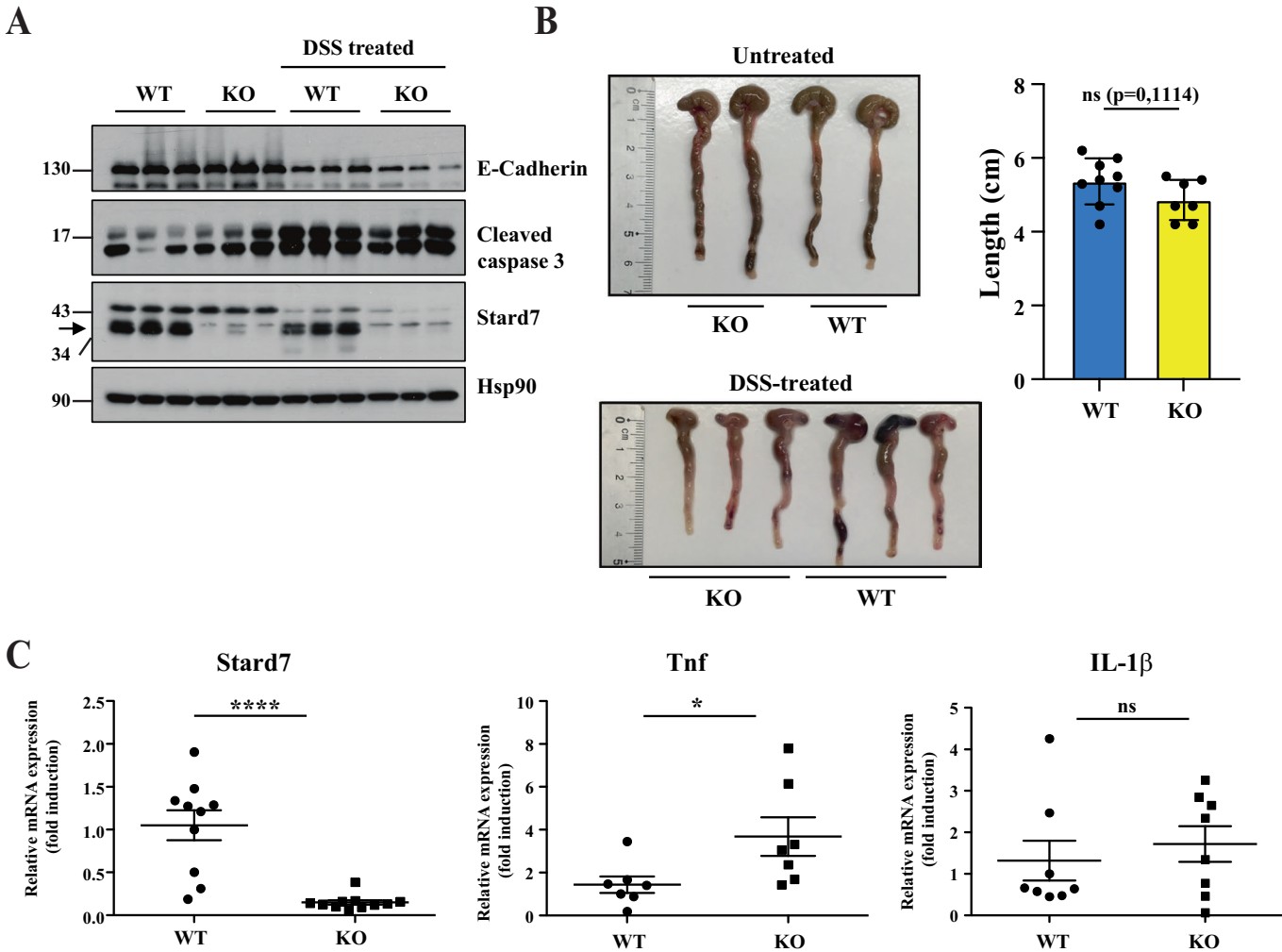

**Figure EV5. Stard7 expression in intestinal epithelial cells does not protect from DSS-induced colitis.**

(A) Stard7 deficiency in IECs does not interfere with cell apoptosis and slightly potentiates the decrease in E-cadherin protein levels upon DSS treatment. In all, 8–12 weeks old mice (all males) were treated or not with a 3% DSS solution in the drinking water for 6 consecutive days. 3 Stard7$^{Lox/lox}$ (WT) and 3 Stard7$^{\Delta IEC}$ (KO) mice were left untreated while 9 Stard7$^{Lox/lox}$ mice and 7 Stard7$^{\Delta IEC}$ mice were subjected to DSS. After sacrifice, IECs from the colon were collected and the resulting extracts were subjected to western blot analyses using the indicated antibodies. The arrow depicts the specific band for Stard7. (B) Stard7 deficiency does not potentiate the decrease of the colon length upon DSS administration. Pictures of the colon of the indicated mice genotypes subjected or not to DSS are illustrated. On the right, a quantification of the intestinal length of the indicated mice genotypes subjected to DSS is shown (means ± S.D., $T$ test with Welch correction, ns = not significant, $n = 9$ and 7 for WT and KO mice, respectively). (C) Stard7 deficiency in IECs potentiates the production of TNF but not IL-1β upon DSS administration. Mice of the indicated genotypes were subjected to DSS as explained in (A). The resulting mRNA extracts were subjected to quantitative Real-Time PCR experiments to quantify the indicated candidates. mRNA levels in one randomly selected Stard7$^{Lox/lox}$ (WT) mice were set to 1 and levels in other mice were relative to that after normalization with Gapdh ($n = 9$ and 7 for WT and KO mice, respectively, means ± S.D., Mann–Whitney test. Stard7: ****$P < 0.0001$, Tnf: $P = 0.0262$; IL-1β: $P = 0.5054$, ns = no significance). Source data are available online for this figure.

## Normal epithelium

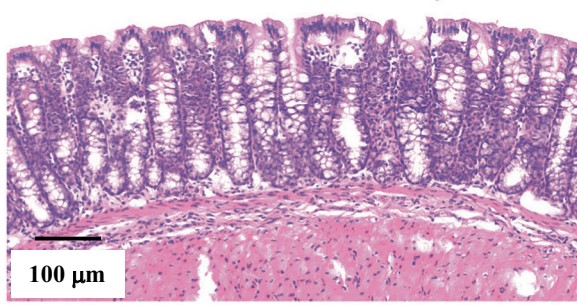

## Low grade

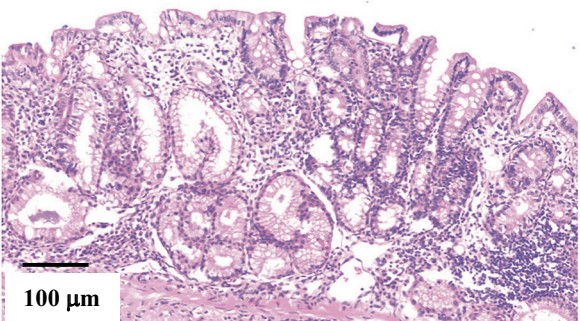

## High grade

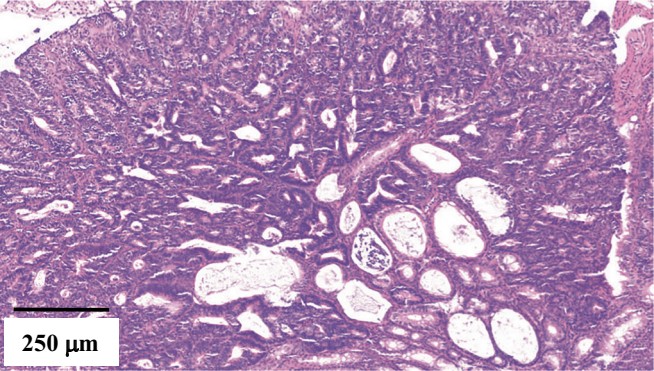

**Figure EV6. Stard7 deficiency impairs inflammation-drive tumor development in the intestine.**

Representative pictures of colon from AOM/DSS-treated mice are illustrated. These images were used as classification standards for tumor grading (see Fig. 2B).

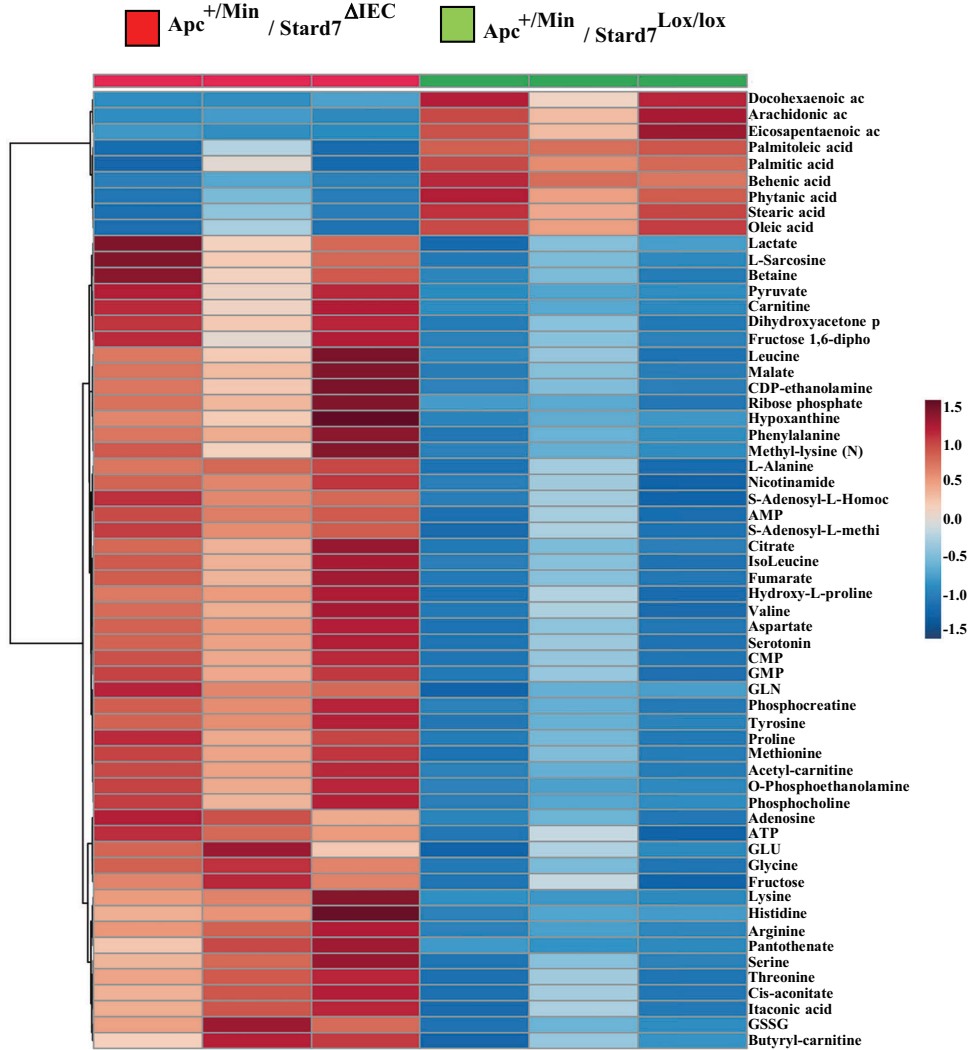

**Figure EV7.   Metabolic reprogramming upon epithelial Stard7 deficiency in Apc$^{+/Min}$ mice.**

The metabolic signature of extracts from the colon of the indicated mouse genotypes (100 days old) was established (*n* = 3 per genotype, all males). This signature reveals decreased levels of fatty acids as well as an accumulation of both TCA intermediates and multiple amino acids in IECs from Apc$^{+/Min}$/Stard7$^{\Delta IEC}$ mice.

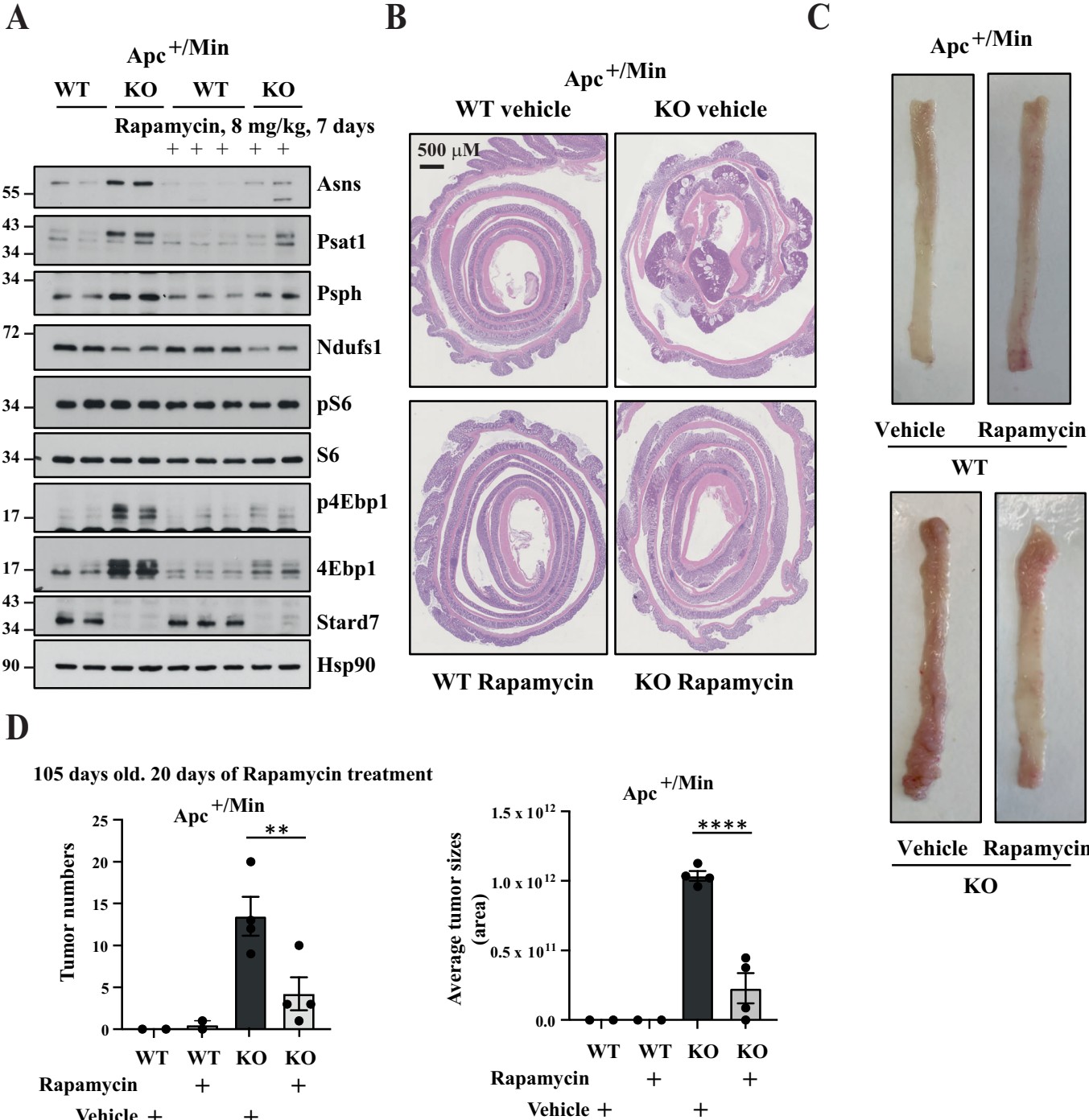

**Figure EV8. Wnt-driven intestinal tumors lacking epithelial Stard7 are sensitive to mTORC1 inhibition.**

(A) Enhanced expression of enzymes involved in serine biosynthesis seen upon Stard7 deficiency in IECs showing or not constitutive Wnt signalling is driven by mTORC1. 105 days old Apc$^{+/Min}$ mice lacking or not Stard7 in IECs (1 WT male, 3 WT females; 4 KO males, 4 KO females) were treated or not with Rapamycin (8 mg/kg) for 7 days and extracts from IECs were subjected to western blot analyses. p4Ebp1 phosphorylation was assessed on Threonine 70. (B–D) mTORC1 pharmacological inhibition in Apc$^{+/Min}$/Stard7$^{\Delta IEC}$ mice efficiently triggers tumor regression in the distal colon. Mice of the indicated genotype (1 WT male, 3 WT females; 4 KO males, 4 KO females) were treated or not with Rapamycin for 20 days and consequences on both tumor sizes and numbers were assessed by IHC analyses (B). Representative colons of mice of the indicated genotype and treated or not with Rapamycin are illustrated (C). The histograms show quantitative data on both tumor numbers and sizes in the indicated experimental conditions (D) (means ± S.D., $n = 4$ mice (Rapamycin-treated mice: 4 males; Vehicle-treated mice: 1 male and 3 females), Dunnett's multiple comparisons test. Histogram on the left: Untreated versus Rapamycin-treated KO mice: **$P = 0.0084$; Histogram on the right: Untreated versus Rapamycin-treated KO mice: ****$P < 0.0001$). Source data are available online for this figure.

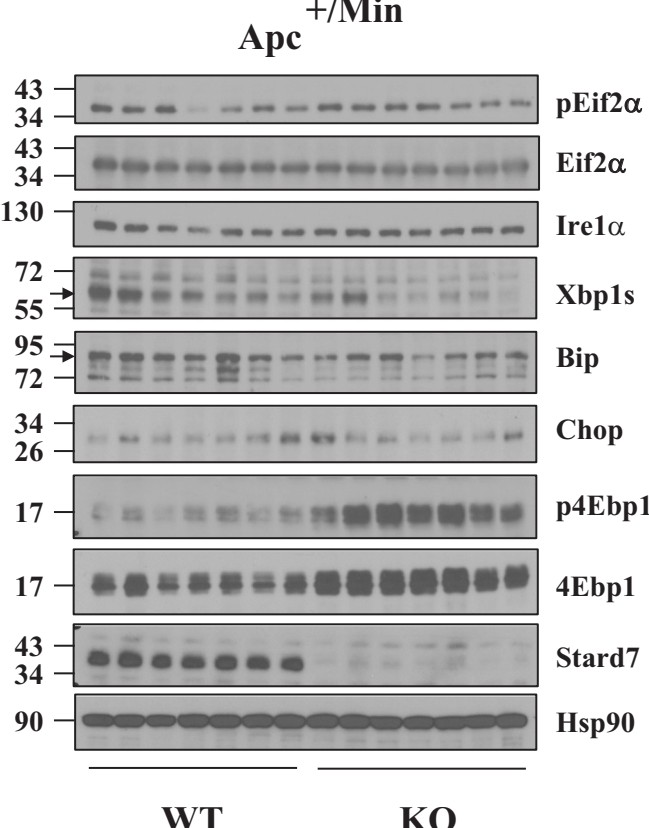

**Figure EV9.   Epithelial Stard7 deficiency in Apc⁺/Min mice does not influence protein levels of UPR effectors.**

Extracts from 105 days old Apc⁺/Min mice lacking or not Stard7 in IECs (3 WT males, 4 WT females; 4 KO males, 3 KO females) were subjected to WB analyses using the indicated antibodies. The arrows depict the specific band for Xbp1s and Bip. Source data are available online for this figure.

