## [Peer Review File · EMBO Molecular Medicine]

The lipid transfer protein STARD7 controls intestinal tumor development in a context-dependent manner

Kateryna Shostak, Yu Chen, Chloé Maurizy, Gilles Rademaker, Xinyi Xu, Arnaud Blomme, Pierre Close, Olivier Renson, Matthias Van Hul, Patrice Cani, Sebastian Klein, Alexandra Florin, Reinhard Buettner, Didier Cataldo, Philippe Delvenne, Ivan Nemazanyy, Caroline Wathieu, Alexandre Hego, Sandra Ormenese, Olivier Peulen, Marc Thiry, Roopesch Krishnankutty, Jair Marques, Alexander von Kriegsheim, and Alain Chariot

Corresponding author: Alain Chariot (alain.chariot@uliege.be)

Review Timeline:

Submission Date:	1st Apr 25
Editorial Decision:	25th Apr 25
Revision Received:	24th Dec 25
Editorial Decision:	16th Jan 26
Revision Received:	26th Jan 26
Editorial Decision:	17th Feb 26
Revision Received:	23rd Feb 26
Accepted:	9th Mar 26

Editor: Lise Roth

Transaction Report:

25th Apr 2025

Dear Prof. Chariot,

Thank you for the submission of your manuscript to EMBO Molecular Medicine. We have now received feedback from the three reviewers who agreed to evaluate your manuscript. As you will see from the reports below, the referees acknowledge the interest of the study and are overall supporting publication of your work pending appropriate revisions.

Addressing the reviewers' concerns in full will be necessary for further considering the manuscript in our journal, and acceptance of the manuscript will entail a second round of review. EMBO Molecular Medicine encourages a single round of revision only and therefore, acceptance or rejection of the manuscript will depend on the completeness of your responses included in the next, final version of the manuscript. For this reason, and to save you frustration at the end, I would strongly discourage you from returning an incomplete revision.

We are expecting your revised manuscript within three to four months, if you anticipate any delay, please contact us.

We require:

4) A .docx formatted letter INCLUDING the reviewers' reports and your detailed point-by-point responses to their comments. As part of the EMBO Press transparent editorial process, the point-by-point response is part of the Review Process File (RPF), which will be published alongside your paper.

5) A complete author checklist, which you can download from our author guidelines (<https://www.embopress.org/page/journal/17574684/authorguide#submissionofrevisions>). Please insert information in the checklist that is also reflected in the manuscript. The completed author checklist will also be part of the RPF.

6) All Materials and Methods need to be described in the main text using our 'Structured Methods' format. According to this format, the Methods section includes a Reagents and Tools Table (listing key reagents, experimental models, software and relevant equipment and including their sources and relevant identifiers) followed by a Methods and Protocols section describing the methods, ideally using a step-by-step protocol format. The aim is to facilitate adoption of the methodologies across labs. Please download and fill our Reagents and Tools Table template (.docx), which you can find in our author guidelines: <https://www.embopress.org/page/journal/14693178/authorguide#structuredmethods>.

7) Please note that all corresponding authors are required to supply an ORCID ID for their name upon submission of a revised manuscript.

8) It is mandatory to include a 'Data Availability' section after the Materials and Methods. Before submitting your revision, primary datasets produced in this study need to be deposited in an appropriate public database, and the accession numbers and database listed under 'Data Availability'. Please remember to provide a reviewer password if the datasets are not yet public (see <https://www.embopress.org/page/journal/17574684/authorguide#dataavailability>).

9) For data quantification: please specify the name of the statistical test used to generate error bars and P values, the number (n) of independent experiments (specify technical or biological replicates) underlying each data point and the test used to calculate p-values in each figure legend. The figure legends should contain a basic description of n, P and the test applied. Graphs must include a description of the bars and the error bars (s.d., s.e.m.). Please provide exact p values.

10) Our journal encourages inclusion of *data citations in the reference list* to directly cite datasets that were re-used and obtained from public databases. Data citations in the article text are distinct from normal bibliographical citations and should directly link to the database records from which the data can be accessed. In the main text, data citations are formatted as follows: "Data ref: Smith et al, 2001" or "Data ref: NCBI Sequence Read Archive PRJNA342805, 2017". In the Reference list, data citations must be labeled with "[DATASET]". A data reference must provide the database name, accession number/identifiers and a resolvable link to the landing page from which the data can be accessed at the end of the reference. Further instructions are available at .

11) We replaced Supplementary Information with Expanded View (EV) Figures and Tables that are collapsible/expandable online. EV Figures should be cited as 'Figure EV1, Figure EV2' etc... in the text and their respective legends should be included in the main text after the legends of regular figures.

12) The paper explained: EMBO Molecular Medicine articles are accompanied by a summary of the articles to emphasize the major findings in the paper and their medical implications for the non-specialist reader. Please provide a draft summary of your article highlighting

13) Author contributions: CRediT has replaced the traditional author contributions section because it offers a systematic machine readable author contributions format that allows for more effective research assessment. Please remove the Authors Contributions from the manuscript and use the free text boxes beneath each contributing author's name in our system to add specific details on the author's contribution. More information is available in our guide to authors.

Please also suggest a visual abstract to illustrate your article as a PNG file 550 px wide x 300-600 px high. A cropped portion of this image will serve as thumbnail for the table of content on our webpage.

16) As part of the EMBO Publications transparent editorial process initiative (see our Editorial at <http://embomolmed.embopress.org/content/2/9/329>), EMBO Molecular Medicine will publish online a Review Process File (RPF) to accompany accepted manuscripts.

In the event of acceptance, this file will be published in conjunction with your paper and will include the anonymous referee reports, your point-by-point response and all pertinent correspondence relating to the manuscript. Let us know whether you agree with the publication of the RPF and as here, if you want to remove or not any figures from it prior to publication. Please note that the Authors checklist will be published at the end of the RPF.

I look forward to receiving your revised manuscript.

Yours sincerely,

Lise Roth

**** Reviewer's comments ****

Referee #1 (Remarks for Author):

In this manuscript, the authors try to uncover the function of STARD7, one of PC transporters, in colon cancer development. However, there are many issues or problems. It is listed below.

1. The molecular mechanism of how STARD7 regulates cancer growth via PC, is not presented although the authors test ROS, MTORC signal, gene expression, and DNA mutant. The change in lipid metabolism may have a huge effect on cells, so I suggest the authors determine PC levels in mitochondrial, plasma membrane, and ER and explain why;
2. The figures are not organized well. It is difficult to follow and understand. Why do you list all these genes in WB but not explain the reason well, e.g., Fig 5E, why the level of pmTOR and pAKT did not change but a little for p4Ebp1? Fig 5F, Why you didn't determine pmTOR, pAKT, and pS6? Why STARD7 KO can do that? Some wrong labels and statistical analysis (e.g., one-way ANOVA, 4 mice for each group?);
3. What is the relationship between STARD7 and microbiota? Why?
4. The authors do not understand cancer lipid metabolism well, e.g., PMID: 39674303;

Referee #2 (Comments on Novelty/Model System for Author):

The authors use a wide set of techniques, including two mouse models and top omics approaches, together with conventional biochemical approaches to figure out the role of STARD7 in colon tumorigenesis.

Referee #2 (Remarks for Author):

In this work Shostak and collaborators addressed the importance of the protein STARD7 in the development of intestinal tumors. The authors used a wide and impressive array of techniques in order to figure out how this lipid transfer protein participates in tumorigenesis. In general, the paper is well written and the points to evaluate are explained to follow the rationale. The findings are very interesting and may be helpful to take advantage of the use of drugs like rapamycin or antioxidants in the treatment of intestinal cancers (after previous cancer genomic characterization).

As the authors explain and stress along the manuscript introduction STARD7 is phosphatidyl transporter, and its functions have been linked to mitochondrial functions. As might have been expected the authors describe here that STARD7 genetic elimination in intestinal epithelial cells results in dramatic alterations in the content of different cellular lipids, and in impairment of Complex I activity, which finally led to mitochondrial stress. My main concern is why the authors didn't evaluate, in any of the models they used, the number, size and appearance of the mitochondria. Using immunofluorescence or cytometric approaches can be helpful to address this issue.

Additional observations:

- 1) Page 6. The first part of results regarding the use of tumor cell lines, supplementary information 1, can be moved after the first tumorigenesis section to keep a better order of line of evidence.

- 2) Page 8. IECs section. What are the effects of STARD7 deletion in the appearance of mitochondria? Can the authors evaluate ROS production in IECs instead/ besides, using cell lines?
- 3) Page 8. It could be useful to explain more about Atf4 here, complete name? and association with mTOR to understand the relationship between this protein and the effects described later in this and in other results sections related to transcriptional reprogramming.
- 4) End of Page 11. It is not clear here if wnt signaling results in genomic instability or this is the result of high ROS. Please clarify before describing the use of exome sequencing experiments.
- 5) Please review and homogenize the labelling of STARD7 deficient IECs in the figures: AIEC or KO?, for example in figure 7.

Referee #3 (Remarks for Author):

The manuscript from Shostak and colleagues investigates the role of StarD7, a lipid transfer protein, in the development of two different intestinal cancer models: a genetically determined one, the Apc+/Min, and a chemically inducible one, deriving from DSS-induced colitis. Interestingly, the present work uncovers opposing effects of StarD7 deficiency on tumorigenesis, demonstrating that StarD7 acts as a tumor suppressor in Apc haploinsufficiency but as an oncogene in inflammation-driven cancer. The effects seem to rely on an mTORc-dependent mechanism involving ROS and metabolic reprogramming. Overall, the work is rooted into the existing literature and presents interesting data concerning the effects of StarD7 in cancer development, which is still an understudied topic compared to its potential significance as a prognostic factor for human cancers. However, the paper presents a few critical issues from a methodological standpoint, and in my opinion some further experiments are required in order to strengthen the story.

General concerns about the present work are the following:

1. Other relevant papers should be discussed, namely:

- 10.1016/j.freeradbiomed.2016.08.023, where a link of StarD7 with ER stress and ROS but not necessarily mitochondrial stress is delineated. The differences in findings should be addressed.
- 10.1002/mc.23560, in which the authors study the TCF4-dependent regulation of StarD7 in colon cancer, which partially overlaps with the results reported in Fig. S1. The paper seems in accordance with the present work, highlighting overexpression of StarD7 in colon cancer, and it should be cited.
- 10.1111/febs.16979, which suggests that reduced OXPHOS upon StarD7 deficiency is due to metabolic reprogramming. The paper further highlights that mitophagic flux is implicated in rebalancing the mitochondrial defects, which fits with the observation that intestinal homeostasis in the absence of cancer is not altered.

2. At least in some panels (for example Fig. 1H) there is evidence of culturing on an immortalized IECs line: the details should be reported in the methods. In another panel (Fig. 7C), the use of organoids is reported: are the Apc+/Min derived from tumoral tissue or from healthy looking parts of the gut? The specifics should be included in the methodology section, and at least their morphology should be integrated in a supplementary panel.

3. Quantitative PCR data are reported as a fold over non-StarD7 mutant, as seen from example in Fig. 5D: "Levels of these candidates in Apc+/Min/Stard7Lox/lox mice were set to 1 and levels in Apc+/Min/Stard7ΔIEC mice were relative to that after normalization with Gapdh mRNA levels (n > 4 mice for each genotype, means + S.D.)". It is quite evident to the eye that in many panels the mean of the "WT" condition is not set to 1, see here TNF and IL-6 for extreme cases. The authors should correct either the graphs or their explanation in the legend, concerning the following panels: Fig. 3F, Fig. 5D, Fig. 6B, Fig. S2D, Fig. S3C.

4. No quantification and no number of replicates is reported for any of the presented WB. While in some cases an adequate number of replicates is shown and striking differences are evident, some blots are of unclear interpretation. Please provide the number of replicates and quantifications. Furthermore, some of the panels are clearly assembled from different gels (such as S1F, S1H, S6A): the housekeeping should be provided for each gel at least as uncropped, if not in the assembled figures. The authors are encouraged to provide all relevant raw data in the supporting material.

Specific points:

1. In Fig. S1 the following statement is made: "Therefore, STARD7 expression is not regulated by Wnt signaling in colon cancer-derived cell lines as well as in mouse IECs but promotes the expression of Wnt effectors". The conclusion regarding IECs should be reformulated, as the results from the β -catenin mice do not prove that StarD7 promotes Wnt in non-tumoral IECs.

2. In Fig. 1 and Fig. S2 the authors argue that intestinal homeostasis is not altered in the small intestine of Stard7ΔIEC mice: however, only cell junctions are examined for the colon, and in subsequent panels the authors shift to using human colon cancer cell lines for their assessment of ROS. This is inconsequential, especially given that the rest of the paper discusses the differences in StarD7 requirement not only between healthy and tumor, but even across different tumor models. The authors should show ROS levels in the pertinent model, IECs or organoids, to be able to claim that mitochondrial defects induce high ROS.

Of note, a measure of Complex I activity and reduced ETC proteins do not justify the conclusion of mitochondrial dysfunction, even given previous reports of increased glycolysis at the expense of OXPHOS linked to StarD7 deficiency (10.1111/febs.16979). The authors should perform mitochondrial imaging to check their mass and morphology, and test for more specific mitochondrial stress markers, such as Atf5 or MAM proteins for ER connectivity. Measuring markers of mitophagy will determine if its increase is responsible for maintaining homeostasis in the absence of cancer. As an optional point, it would be interesting to test if indeed a switch between OXPHOS and glycolysis is occurring.

Fig. 1H does not show the blot for p4Ebp1, it should be included.

Fig. S2F is missing the molecular weight markers.

3. In Fig. 3F the authors observe lower induction of inflammatory cytokines in Stard7 Δ IEC mice compared to controls. Could the authors speculate regarding the mechanism driving this difference? Could it be linked to the downregulation of Tigar and upregulation of p $\text{H}2\text{AX}$ reported in Fig. 3C? Regardless, the significance of these alterations should be clarified (and the uncropped blots for Tigar, p $\text{H}2\text{AX}$ and CC3 should be included in the supporting file).

Furthermore, the uncropped blots for Fig. 3E report a lane for Lysozyme, which is not presented in the panel (and shouldn't exist, as the sample is colon). Please check that the blots presented are correct.

4. "Stard7 is promoting the expression of Wnt effectors in IECs but it is nevertheless unclear whether Stard7 is required in Wnt-driven tumor initiation". Fig. S2D clearly shows that upon Stard7 deficiency, in normal epithelium, no alteration of Wnt effectors can be observed, as noted in the corresponding paragraph: "mRNA levels of Wnt target genes (Lgr5, c-Myc, Sox9 and Cd44) remained unchanged in IECs lacking Stard7 (Supplementary Fig. 2D)". As per point #1, the statement should be corrected.

5. In Fig. 5, the authors report that the increase in tumor burden in the Apc+/Min model lacking Stard7 is due to increased mutational rate. Is there a previous report suggesting higher instability in Apc+/Min cancers compared to inflammation-driven ones? The parallel between Fig. 3F and Fig. 5D suggests that the differential upregulation of inflammatory molecules is a key difference between the two cancer types in the absence of Stard7, as per point #3.

In Fig. 5E but not in 5F, the WB relative to duodenal extracts is presented. Given that the whole characterization of the Apc+/Min model was focused on the colon and on the fact that potential differences between the two organs are not discussed, the relative data should be removed.

Fig 5F: The link between Gpx4, Coenzyme Q and Stard7 is promising, but it is unclear why ferroptosis would limit tumor development in the Apc model and not in the DSS one. The authors should discuss this and, if possible, give a direct measurement of ferroptosis: an extensive array of methods to quantify ferroptosis is available in the literature, including but not limited to 10.1038/cdd.2015.158.

6. In Fig. 6, the authors show reduced OXPHOS as measured by Complex I activity; in Fig. S5, they identify increased intermediates of the TCA cycle, suggesting that the metabolic demands of the tumor in Apc+/Min/Stard7 Δ IEC conditions are fueled mostly by glycolysis, which feeds pyruvate into the TCA. It seems that many of the effects seen in the Apc+/Min background (the lipid signature and reduced activity of Complex I, for example) are similar to the ones seen in non-tumoral tissue and just depend on the Stard7 status. Could the authors clarify which alterations are tumor-specific, and among those, which are specific to the DSS model but not the Apc+/Min one?

7. Fig. 7B highlights the effects of NAC on the molecular signature of the tumor. Is the effect applicable to Stard7 lox/lox mice as well? It is unclear from the blots presented, and if so, it would invalidate the point that ROS are a main driver of tumor formation in the specific context of Stard7 deficiency. Does the treatment affect tumor number/area? A characterization as seen in Fig. S6B-D for Rapamycin should be included.

8. In Fig. 7C, 4Ebp1 phosphorylation seems to be decreased in the WT Apc condition and not at all triggered in Apc+/Min background, contrary to what is stated in the text. Similarly, Wnt effectors appear to decrease in the WT background upon ROS stimulation. Is it linked to severe increase in apoptosis, as pJnk suggests? In that case, the highest H 2O_2 dose should not be taken into account.

9. Fig. 8 shows convincing data proving that antibiotics reduce the tumor load in Apc+/Min Stard7 Δ IEC mice, but it is difficult to evaluate if the effect is specific for the Stard7-deficient status, given that Stard7 lox/lox mice develop almost no tumors (similarly to what is seen in Fig. S6D, but not in Fig. 4D). It is unclear what this experiment adds compared to the paper Li et al, 2012 that the authors cite. Are the reported alterations in microbiota composition specific to Apc loss or are they conserved in mice with DSS induced carcinogenesis/no tumor background?

10. In the discussion, the authors speculate that "Mitochondria lacking Stard7 may face OXPHOS deficiency by potentiating FAO as an alternative source of energy". This is a good suggestion, and I don't agree that it is unreasonable due to mitochondrial stress: as I discussed above, lower OXPHOS rate does not equal mitochondrial stress or inability to do FAO. The hypothesis should be tested by inhibiting FAO with drugs such as Etomoxir in IECs or organoids and checking for its contribution to total ATP production.

11. DNA and RNA sequencing data are reported to be aligned on the human reference genome. It is probably a mistake and should be corrected in the text.

Overall, most of the main messages in the present work are solid and well-constructed. I believe that the experimental requests will be reasonably feasible and that most of the points I raised could be tackled by integrating the discussion or reformulating the phrasing of the text. If these are satisfactorily addressed, I recommend the publication in EMBO Molecular Medicine.

Referee #1 (Remarks for Author):

In this manuscript, the authors try to uncover the function of STARD7, one of PC transporters, in colon cancer development. However, there are many issues or problems. It is listed below.

1. The molecular mechanism of how STARD7 regulates cancer growth via PC, is not presented although the authors test ROS, MTORC signal, gene expression, and DNA mutant. The change in lipid metabolism may have a huge effect on cells, so I suggest the authors determine PC levels in mitochondrial, plasma membrane, and ER and explain why;

Our answer:

We thank Reviewer 1 for his/her interesting experimental suggestion. To indeed more precisely define the consequences of Stard7 deficiency on the lipid composition of distinct subcellular compartments, we repeated these lipidomic analyses using ER, mitochondrial, cytoplasmic and lysosomal/plasma membrane extracts from the intestine of both WT and KO mice. In agreement with a role of Stard7 in PC transport from the ER to mitochondria, multiple PC derivatives significantly accumulated in the ER of IECs lacking Stard7 (Please see our new Extended View Figure 3B). This conclusion did not apply to all PC derivatives as levels of 3 of them (i.e. PC 35:1, PC O-42:3 and PC 40:0) were higher in the mitochondrial extracts from the intestine of Stard7 KO mice (Extended View Figure 3B). Levels of multiple PC derivatives also significantly increased in lysosomal but not in cytoplasmic extracts upon Stard7 deficiency (Extended View Figure 3B). Strikingly, the distribution of other candidates such as Ceramides and Plasmalogen lipids also changed in both ER and mitochondria extracts from the intestine of KO versus WT mice (Extended View Figure 3B). Therefore, Stard7 deficiency has profound consequences on the landscape of lipid composition in multiple cellular organelles. It is also important to note that cells lacking Stard7 also show more mitochondrial-associated membranes (MAMs) (see our new Expanded View Figure 3C), as previously demonstrated by our team in breast cancer cells (Dondajewska et al., *Advanced Science* (2025), May 30: e03022, PMID 40443279). This most likely reflects a compensatory mechanism due to defects in PC transport.

We also updated our discussion by adding the following paragraph:

“Our lipidomic analyses indicate that Stard7 deficiency leads to a profound reprogramming of the lipid landscape in cellular organelles such as ER, mitochondria and lysosomes. This observation suggests that the loss of Stard7 has a strong impact on PC levels in both ER and mitochondria with some consequences on the composition of other lipids found in multiple membranes. Our results also suggest that Stard7 may regulate tumor development through additional ROS-independent pathways and involving lipids as signaling molecules. While this hypothesis is interesting to experimentally address, it is important to note that the changes in the lipid landscape seen in cells lacking Stard7 were very similar in both wild type and Apc-mutated intestinal epithelial cells. Yet, Stard7 has opposite roles in tumor development in both experimental models, which strengthens the notion that the Apc status has to be taken into account for any experiment aiming at defining the role of lipids as

2. The figures are not organized well. It is difficult to follow and understand. Why do you list all these genes in WB but not explain the reason well, e.g., Fig 5E, why the level of pmTOR and pAKT did not change but a little for p4Ebp1? Fig 5F, Why you didn't determine pmTOR, pAKT, and pS6? Why STARD7 KO can do that? Some wrong labels and statistical analysis (e.g., one-way ANOVA, 4 mice for each group?);

Our answer:

The issue regarding the organization of our figures is not shared by other reviewers. We nevertheless re-organized some of them by first describing our data in healthy mice (Figs. 1 and Extended View Figures 1 to 3) before moving into our mouse models of cancer. We did not assess pmTOR, pAKT and pS6 in Fig. 5F as these results are already illustrated in Fig. 5E. We nevertheless illustrated p4Ebp1 and total 4Ebp1 as our « internal positive control », given the fact that elevated p4Ebp1 and total 4Ebp1 levels were systematically seen in several experimental mouse models lacking *Stard7* as well as in established colon cancer cell lines.

Regarding the issue raised on statistical analyses, we used several statistical tests depending on the number of parameters we analysed. Many experiments were carried out with two distinct genotypes and the T-test with Welch' correction was often used. When other statistical tests were used, we mentioned this information in figure legends. For Expanded View Figure 8D (we are assuming that it is the figure Reviewer 1 is talking about), we used the Dunnett's multiple comparison test). Our statistical analyses were done using the Prism 10 software.

Regarding wrong labels, it is unclear what Reviewer 1 precisely means. We nevertheless went through all figures and provided a consistent way to describe our genotypes : « WT » for *Stard7*^{Lox/lox} and « KO » for *Stard7*^{ΔIEC}.

3. What is the relationship between STARD7 and microbiota? Why?

Our answer:

Our initial goal was to address the role of *Stard7* in intestinal cancers using a couple of mouse models of cancer in which *Stard7* is genetically inactivated in intestinal epithelial cells (the AOM/DSS model and the *Apc*^{+/^{Min}} model). We were actually not expecting to see that *Stard7* deficiency would enhance colon tumorigenesis upon constitutive Wnt signaling, at least through ROS-dependent mTORC1 activation. We and others have demonstrated that microbiota contributes to Wnt-driven intestinal tumor development (see for example our Cancer Research paper by Goktuna et al. 2016, PMID: 26980769). Therefore, it made sense to us to investigate to which extent microbiota was involved in this phenotype and this hypothesis turned out to be true. Therefore, our study defines mechanisms through which the loss of a lipid transfer protein potentiates ROS-dependent mTORC1 activation and enhances the number of mutations in the intestine to fuel colon tumorigenesis. This phenotype is also associated to a specific microbiota signature also found in patients suffering from colon cancer. We believe that our findings are important for the research community interested in generating new mouse models of cancer that nicely mimic the human disease : this

$Apc^{+/Min}/Stard7^{\Delta IEC}$ model meets this goal as it can be an elegant model to further explore the role of specific bacterial strains and their metabolites in colon tumorigenesis.

4. The authors do not understand cancer lipid metabolism well, e.g., PMID: 39674303;

Our answer:

Scientists have the freedom to explore new fields of research during their career. As signaling experts in NF- κ B-dependent cascades, we then moved to mouse models of intestinal cancers 10 years ago in order to integrate our findings on oncogenic signaling pathways in vivo. It is unclear why Reviewer 1 is stating that we do not understand lipid metabolism well as we lack any specific information that would support this claim. In any case, it is also very interesting to note that despite very similar (if not identical...) transcriptomic and lipidomic signatures seen upon *Stard7* deficiency in both experimental models of intestinal cancers, we show that *Stard7* deficiency has opposite consequences on tumor development. Therefore, we think that our study should be of interest to many scientists studying lipid metabolism in cancer.

Referee #2 (Comments on Novelty/Model System for Author):

The authors use a wide set of techniques, including two mouse models and top omics approaches, together with conventional bicochemical approaches to figure out the role of STARD7 in colon tumorigenesis.

Referee #2 (Remarks for Author):

In this work, Shostak and collaborators addressed the importance of the protein STARD7 in the development of intestinal tumors. The authors used a wide and impressive array of techniques in order to figure out how this lipid transfer protein participates in tumorigenesis. In general, the paper is well written and the points to evaluate are explained to follow the rationale. The findings are very interesting and may be helpful to take advantage of the use of drugs like rapamycin or antioxidants in the treatment of intestinal cancers (after previous cancer genomic characterization). As the authors explain and stress along the manuscript introduction STARD7 is phosphatidyl transporter, and its functions have been linked to mitochondrial functions. As might have been expected the authors describe here that STARD7 genetic elimination in intestinal epithelial cells results in dramatic alterations in the content of different cellular lipids, and in impairment of Complex I activity, which finally led to mitochondrial stress. My main concern is why the authors didn't evaluate, in any of the models they used, the number, size and appearance of the mitochondria. Using immunofluorescence or cytometric approaches can be helpful to address this issue.

Our answer:

We thank Reviewer 2 for his/her positive comments. We are now providing a revised version in which the requested experiments, namely the quantification of the number, size and appearance of mitochondria were carried out (see our new Expanded View Figure 3A).

Additional observations:

c1) Page 6. The first part of results regarding the use of tumor cell lines, supplementary information 1, can be moved after the first tumorigenesis section to keep a better order of line of evidence.

Our answer:

We moved former Expanded View Figure 1 (link between STARD7 and Wnt signaling as well as STARD7 expression in human cases of colon cancer) to Expanded View Figure 4 while data in the former Expanded View Figure 2 (role of Stard7 in intestinal homeostasis) are now illustrated in both Expanded View Figures 1 and 2. With this, we first describe the role of Stard7 in intestinal homeostasis before moving into data in which the role of STARD7 in intestinal cancer is assessed, as requested by Reviewer 2.

2) Page 8. IECs section. What are the effects of STARD7 deletion in the appearance of

mitochondria? Can the authors evaluate ROS production in IECs instead/ besides, using cell lines?

Our answer:

The requested experiments are illustrated in Extended View Figure 3A. We first carried out transmission electronic microscopy experiments to assess mitochondria morphology in intestinal tissues from both *Stard7^{Lox/lox}* and *Stard7^{ΔIEC}* mice. We noticed that the number of mitochondria per μm^2 of cytoplasm significantly decreased in cells lacking *Stard7* (Extended View Figure 3A). In contrast, the mean mitochondrial area increased upon *Stard7* deficiency (Extended View Figure 3A). As a result, the total mitochondrial area in the cytoplasm did not significantly change in cells lacking *Stard7* (Extended View Figure 3A). Importantly, we also showed that mitochondria-associated membranes (MAMs) were enhanced in cells lacking *STARD7* (see our new Extended View Figure 3C), as previously demonstrated by our team in breast cancer cells (Dondajewska and colleagues, *Advanced Science* (2025), PMID: 40443279). Collectively, these results demonstrate that IECs lacking *STARD7* show enlarged mitochondria with more MAMs. These observations may reflect mitochondria swelling but this hypothesis would deserve more experimental evidence.

Regarding ROS production in other experimental systems than established colon cancer cell lines, we agree with Reviewer 2 that this issue is very important to address. We have been trying very hard to quantify ROS production through several experimental approaches. Regarding in vivo ROS measurement, the probe described in the *Gastroenterology* paper (2015) published by S. Foersch and colleagues (PMID: 25797700), was discontinued in May 2019. We were invited to use other probes such as ROS Brite 800 but this probe was discontinued as well. The ROS Brite 700 probe was not suitable for ROS detection in the intestine. Therefore, we have no choice but to conclude that there is unfortunately no reliable way to measure ROS production in the mouse intestine today. We then moved to ex-vivo organoids lacking or not *Stard7*. We followed-up the protocol established by A. Stedman and colleagues (see PMID 34605823). We had to culture these ex-vivo organoids without NAC (or any other ROS inhibitors). In those circumstances, we noticed that FACS analyses carried out gave highly variable results within the same genotype. Indeed, changes in ROS levels highly depend on the size and shape of ex-vivo organoids in each well, i.e. on factors that are very hard to totally control in these 3D cultures. As a result, we could not reach any reliable conclusions using these cultures. Because 4-HNE is an aldehyde product of lipid peroxidation seen as a sensitive marker of oxidative damage, we then decided to carry out 4-HNE stainings in intestinal tissues from both *Apc^{+Min}/Stard7^{Lox/lox}* and *Apc^{+Min}/Stard7^{ΔIEC}* mice (see our new Fig. 5G). We now show that the number of 4-HNE⁺ cells significantly increases in sections from *Apc^{+Min}/Stard7^{ΔIEC}* mice. This result, combined with the fact that NAC interferes with mTORC1 activation and with the downstream transcriptional signature (see Fig. 7B) and with elevated ROS levels seen in established colon cancer cell lines lacking *Stard7*, demonstrates that *Stard7* deficiency enhances ROS levels.

3) Page 8. It could be useful to explain more about Atf4 here, complete name? and association with mTOR to understand the relationship between this protein and the

effects described later in this and in other results sections related to transcriptional reprogramming.

Our answer:

We defined Atf4 (Activating Transcription factor 4) in the revised version. Please also note that we mentioned that Atf4 is an mTORC1 effector. We also described the published study by Torrence and colleagues (eLife, 2021, PMID : PMC7997658) in which a specific transcriptional signature downstream of the mTORC1-ATF4 pathway was identified.

4) End of Page 11. It is not clear here if wnt signaling results in genomic instability or this is the result of high ROS. Please clarify before describing the use of exome sequencing experiments.

Our answer:

We thank Reviewer 2 for pointing out this issue. We added more explanations in our revised version by adding the published study by R. Fodde and colleagues (Nature Cell Biology (2001), PMID: 11283620) in which loss-of-function mutations of the tumor suppressor gene *Apc* were found to cause chromosomal instability. We also better phrased our hypothesis, namely the fact that the loss of function of the *Apc* tumor suppressor gene, which triggers chromosomal instability, combined with the enhanced production of ROS due *Stard7* deficiency, would enhance the rate of mutations in IECs.

5) Please review and homogenize the labelling of STARD7 deficient IECs in the figures: AIEC or KO?, for example in figure 7.

Our answer:

For clarity, we decided to use the WT versus KO labelling in « healthy » mice (i.e. in wild type *Apc*-expressing mice). We also decided to use the same labelling for WT and KO mice in the *Apc*^{+/^{Min} background. This nomenclature may bring some confusion to future readers as we may not easily know whether we are dealing with wild type *Apc*-expressing mice or with *Apc*^{+/^{Min} mice. Therefore, we decided to systematically mention « *Apc*^{+/^{Min} mice » when relevant for clarity purposes.}}}

Referee #3 (Remarks for Author):

The manuscript from Shostak and colleagues investigates the role of StarD7, a lipid transfer protein, in the development of two different intestinal cancer models: a genetically determined one, the $Apc^{+/Min}$, and a chemically inducible one, deriving from DSS-induced colitis. Interestingly, the present work uncovers opposing effects of StarD7 deficiency on tumorigenesis, demonstrating that StarD7 acts as a tumor suppressor in *Apc* haploinsufficiency but as an oncogene in inflammation-driven cancer. The effects seem to rely on an mTorc-dependent mechanism involving ROS and metabolic reprogramming.

Overall, the work is rooted into the existing literature and presents interesting data concerning the effects of StarD7 in cancer development, which is still an understudied topic compared to its potential significance as a prognostic factor for human cancers. However, the paper presents a few critical issues from a methodological standpoint, and in my opinion some further experiments are required in order to strengthen the story. General concerns about the present work are the following:

1. Other relevant papers should be discussed, namely:
 - [10.1016/j.freeradbiomed.2016.08.023](https://doi.org/10.1016/j.freeradbiomed.2016.08.023), where a link of StarD7 with ER stress and ROS but not necessarily mitochondrial stress is delineated. The differences in findings should be addressed.

Our answer:

We thank Reviewer 3 for pointing this issue out. The link between the loss of STARD7 and ER stress is indeed interesting to assess and to discuss. Previous studies showed that STARD7 deficiency triggers some changes in the mitochondrial architecture with consequences on ROS production, as described in the reference mentioned by Reviewer 3. Regarding the link with ER stress, it is fair to say that levels of UPR effectors known to be induced and/or activated upon ER stress (*peIF2 α* , *CHOP*, *Ire1 α* and *PERK*) are not dramatically enhanced upon *Stard7* deficiency (please see Figure 3 of this study). Yet, we found the link with ER stress interesting to assess and we recently published that the loss of STARD7 in some but not all breast cancer-derived cell lines (i.e. in triple-negative breast cancer-derived MDA-MB231 cells but not in ER α^+ -derived breast cancer MCF7 cells) enhanced protein levels of *PERK*, *IRE1 α* , *BIP*, *XBP1* and *ATF4* (see Figure 2E of our paper by Dondajewska et al. *Advanced Science* (2025), May 30: e03022, PMID: 40443279). We also showed that mitochondria-ER contacts (MAMs) were enhanced in breast cancer cells lacking STARD7 (see Figure 2B of our paper), which confirms that STARD7 deficiency has some consequences on mitochondrial architecture but also on contacts with other organelles.

To explore whether the enhanced ROS production seen in colon cancer cells showing constitutive Wnt signaling and lacking STARD7 (see Figure 1F) could result from ER stress, we established the expression profile of several UPR effectors known to be induced and/or activated upon ER stress in IECs of 100 days old $Apc^{+/Min}/Stard7^{Lox/lox}$ (“WT”) and $Apc^{+/Min}/Stard7^{\Delta IEC}$ (“KO”) mice and did not notice any change in their expression profile in contrast to both unphosphorylated and phosphorylated 4Ebp1 levels (see our results illustrated in

Expanded View Figure 9). We also updated our discussion on this issue in our revised manuscript.

- **10.1002/mc.23560**, in which the authors study the TCF4-dependent regulation of StarD7 in colon cancer, which partially overlaps with the results reported in Fig. S1. The paper seems in accordance with the present work, highlighting overexpression of StarD7 in colon cancer, and it should be cited.

Our answer:

We added this reference in our revised manuscript and also added one paragraph in the discussion to integrate findings between both studies : « *The fact that STARD7 acts as a tumor suppressor gene in Wnt-driven tumor development was unexpected given the fact that we and others showed that STARD7 is overexpressed in clinical cases of colon cancer known to show constitutive Wnt signaling in most cases (Zhao et al, 2023). While STARD7 overexpression in colon cancer is now well established, the link with Wnt signaling is not clearly established. Both STARD7 and TCF4 expression were reported to be positively correlated (Zhao et al, 2023). However, our data obtained in multiple experimental models do not support the notion that Wnt signaling promotes STARD7 expression. We rather show that STARD7 acts upstream of Wnt signaling by controlling the expression of several Wnt effectors, even if the underlying mechanism remains unclear.* »

- **10.1111/febs.16979**, which suggests that reduced OXPHOS upon StarD7 deficiency is due to metabolic reprogramming. The paper further highlights that mitophagic flux is implicated in rebalancing the mitochondrial defects, which fits with the observation that intestinal homeostasis in the absence of cancer is not altered.

Our answer:

We indeed confirm that metabolic reprogramming is occurring upon StarD7 deficiency. We also addressed the mitochondria morphology and noticed that mitochondria lacking StarD7 were enlarged (see Expanded View Figure 3A). We also saw a reduced number of mitochondria per μm^2 of cytoplasm (Expanded View Figure 3A), which is in agreement with the reference Reviewer 3 is talking about. Regarding the mitophagic flux, this point will be extensively discussed in response to one of the following issues raised by Reviewer 3 (see here after).

2. At least in some panels (for example Fig. 1H) there is evidence of culturing on an immortalized IECs line: the details should be reported in the methods. In another panel (Fig. 7C), the use of organoids is reported: are the $\text{Apc}^{+/Min}$ derived from tumoral tissue or from healthy looking parts of the gut? The specifics should be included in the methodology section, and at least their morphology should be integrated in a supplementary panel.

Our answer:

Regarding the culture of mIECs, we added the following sentence in the methods section (« Cell cultures and generation of ex-vivo organoids »): « *mIECs were maintained in*

Dulbecco's Modified Eagle Medium (Capricorn scientific) supplemented with 10% FBS (Gibco), Penicillin-Streptomycin Mixture (Lonza) and L-Glutamine (Capricorn scientific). »

Regarding the generation of Apc^{+Min}-derived ex-vivo organoids, we isolated cells from the duodenum which had both untransformed cells and tumors at that time. Ex-vivo organoids were generated and maintained in DMEM/F12 supplemented with EGF (20 ng/ml), Noggin (100 ng/ml) but without R-Spondin. As a result, only transformed epithelial cells (i.e. cells showing constitutive Wnt signaling) can grow and ultimately generated ex-vivo organoids showing a typical morphology (see Figure 1A of our paper by Duong and colleagues, Cancer Research, 2018, 78, 4533-4548, PMID: 29915160 for example). These experimental details have been added in the methods section of our revised version in the methods section (« Cell cultures and generation of ex-vivo organoids »). We also added representative pictures of these ex-vivo organoids next to the western blot data.

3. Quantitative PCR data are reported as a fold over non-StarD7 mutant, as seen from example in Fig. 5D: "Levels of these candidates in Apc^{+Min}/Stard7^{Lox/lox} mice were set to 1 and levels in Apc^{+Min}/Stard7^{ΔIEC} mice were relative to that after normalization with Gapdh mRNA levels (n > 4 mice for each genotype, means + S.D.)". It is quite evident to the eye that in many panels the mean of the "WT" condition is not set to 1, see here TNF and IL-6 for extreme cases. The authors should correct either the graphs or their explanation in the legend, concerning the following panels: Fig. 3F, Fig. 5D, Fig. 6B, Fig. S2D, Fig. S3C.

Our answer:

We thank Reviewer 3 for pointing out this issue and we apologize for this mistake. Results in « control » conditions were indeed set to 1 in experiments in which extracts from established colon cancer cell lines were used (see Expanded View Figure 4C). However, for all other Real-Time PCR analyses carried out with mouse extracts, levels from one randomly selected WT mouse were set to 1 and levels in all other experimental conditions were relative to that after normalization with Gapdh mRNA levels. This corrected statement for relevant figures has been added in the revised version.

4. No quantification -and no number of replicates is reported for any of the presented WB. While in some cases an adequate number of replicates is shown and striking differences are evident, some blots are of unclear interpretation. Please provide the number of replicates and quantifications. Furthermore, some of the panels are clearly assembled from different gels (such as S1F, S1H, S6A): the housekeeping should be provided for each gel at least as uncropped, if not in the assembled figures. The authors are encouraged to provide all relevant raw data in the supporting material.

Our answer:

The number of replicates (as well as the number of mice when relevant) used for western blot analyses are now mentioned in figure legends. Note that for Expanded View Figure 4E and 4F in which STARD7 protein levels were assessed in 6 distinct cell lines, these western blot analyses were carried out from one experiment only. We did not feel the need to repeat these experiments given the fact that STARD7 protein levels remained the same in all experimental

conditions in all tested cell lines. The requested housekeeping gels are now illustrated in the Uncropped gels file (we would prefer not to show multiple gels for the same housekeeping genes in the main figures due to space limitations). Source files/Raw data are also provided in the supporting material. The quantification of western blot analyses is provided as a supplementary file (« Densitometry analyses »).

Specific points:

1. In Fig. S1 the following statement is made: "Therefore, STARD7 expression is not regulated by Wnt signaling in colon cancer-derived cell lines as well as in mouse IECs but promotes the expression of Wnt effectors". The conclusion regarding IECs should be reformulated, as the results from the β -catenin mice do not prove that StarD7 promotes Wnt in non-tumoral IECs.

Our answer:

We modified our statement by writing-up the following sentence : « *Therefore, STARD7 expression is not regulated by Wnt signaling in colon cancer-derived cell lines as well as in mouse IECs but promotes the expression of Wnt effectors in established colon cancer cell lines.* »

2. In Fig. 1 and Fig. S2 the authors argue that intestinal homeostasis is not altered in the small intestine of Stard7 ^{Δ IEC} mice: however, only cell junctions are examined for the colon, and in subsequent panels the authors shift to using human colon cancer cell lines for their assessment of ROS. This is inconsequential, especially given that the rest of the paper discusses the differences in StarD7 requirement not only between healthy and tumor, but even across different tumor models. The authors should show ROS levels in the pertinent model, IECs or organoids, to be able to claim that mitochondrial defects induce high ROS.

Our answer:

We included three additional results that further support the dispensable role of Stard7 in intestinal homeostasis. We first showed that Stard7 deficiency in IECs did not change the architecture nor the number of goblet cells in the colon (Extended View Figure 2A). Moreover, we also showed that the body weight of 21 months-old Stard7^{Lox/lox} and Stard7 ^{Δ IEC} mice was very similar, which excludes the possibility that Stard7 deficiency in IECs leads to spontaneous diseases such as intestinal inflammation which is often associated with a body weight loss (Extended View Figure 2C). Likewise, mRNA levels of several pro-inflammatory cytokines such as Tnf, IL-6 and IL-1 β were also similar in both genotypes (Extended View Figure 2D). Collectively, our data suggest that Stard7 deficiency in IECs does not cause any change in the intestinal architecture, does not influence mRNA levels of all tested specific markers of epithelial subtypes and does not lead to any spontaneous inflammatory disorders, even in aged mice. Based on these findings, we can conclude that Stard7 is dispensable in intestinal homeostasis.

Regarding ROS production in other experimental systems than established colon cancer cell lines, we agree with Reviewer 3 that this issue is very important to address. We have been trying very hard to quantify ROS production through several experimental approaches. Regarding in vivo ROS measurement, the probe described in the Gastroenterology paper (2015) published by S. Foersch and colleagues (PMID 25797700), was discontinued in May 2019. We were invited to use other probes such as ROS Brite 800 but this probe was discontinued as well. The ROS Brite 700 probe was not suitable for ROS detection in the intestine. Therefore, we have no choice but to conclude that there is unfortunately no reliable way to measure ROS production in the mouse intestine today. We then moved to ex-vivo organoids lacking or not Stard7. We followed-up the protocol established by A. Stedman and colleagues (see the PubMed 34605823). We had to culture these ex-vivo organoids without NAC (or any other ROS inhibitors). In those circumstances, we noticed that FACS analyses carried out gave highly variable results within the same genotype. Indeed, changes in ROS levels highly depend on the size and shape of ex-vivo organoids in each well, i.e. on factors that are very hard to totally control in these 3D cultures. As a result, we could not reach any reliable conclusions using these cultures. Because 4-HNE is an aldehyde product of lipid peroxidation seen as a sensitive marker of oxidative damage, we then decided to carry out 4-HNE stainings in intestinal tissues from both $Apc^{+/Min}/Stard7^{Lox/lox}$ and $Apc^{+/Min}/Stard7^{\Delta IEC}$ mice (see our new Fig. 5G). We now show that the number of 4-HNE⁺ cells significantly increases in sections from $Apc^{+/Min}/Stard7^{\Delta IEC}$ mice. This result, combined with the fact that NAC interferes with mTORC1 activation and with the downstream transcriptional signature (see Fig. 7B) and with elevated ROS levels seen in established colon cancer cell lines lacking Stard7, demonstrates that Stard7 deficiency enhances ROS levels.

Of note, a measure of Complex I activity and reduced ETC proteins do not justify the conclusion of mitochondrial dysfunction, even given previous reports of increased glycolysis at the expense of OXPHOS linked to Stard7 deficiency (10.1111/febs.16979). The authors should perform mitochondrial imaging to check their mass and morphology, and test for more specific mitochondrial stress markers, such as Atf5 or MAM proteins for ER connectivity. Measuring markers of mitophagy will determine if its increase is responsible for maintaining homeostasis in the absence of cancer. As an optional point, it would be interesting to test if indeed a switch between OXPHOS and glycolysis is occurring. Fig. 1H does not show the blot for p4Ebp1, it should be included. Fig. S2F is missing the molecular weight markers.

Our answer:

We thank Reviewer 3 for these insightful comments. The requested experiments are illustrated in Extended View Figure 2A. We first carried out transmission electronic microscopy experiments to assess mitochondria morphology in intestinal tissues from both $Stard7^{Lox/lox}$ and $Stard7^{\Delta IEC}$ mice. We noticed that the number of mitochondria per μm^2 of cytoplasm significantly decreased in cells lacking Stard7 (Extended View Figure 3A). In contrast, the mean mitochondrial area increased upon Stard7 deficiency (Extended View Figure 3A). As a result, the total mitochondrial area in the cytoplasm did not significantly change in cells lacking Stard7 (Extended View Figure 3A). Importantly, we also showed that mitochondria-associated membranes (MAMs) were enhanced in cells lacking STARD7 (see our new

Extended View Figure 3C), as previously demonstrated by our team in breast cancer cells (Dondajewska and colleagues, *Advanced Science* (2025), PMID: 40443279). Collectively, these results demonstrate that IECs lacking STARD7 show enlarged mitochondria with more MAMs. These observations may reflect mitochondria swelling but this hypothesis would deserve more experimental evidence. Pathological swelling ultimately leads to mitophagy. We also assessed mitophagy markers as requested by Reviewer 3 and our results are illustrated here below. While we confirmed again that IECs lacking Stard7 showed elevated ASNS levels, we did not find any changes in protein levels of mitophagy effectors such as Vdac1, a key actor acting downstream of Parkin. BNIP3L/Nix levels remained unchanged as well. Of note, Mitofusin2, which is critically involved in mitochondrial fusion also showed similar levels in both genotypes. Although we understand that many additional experiments (most of them are challenging to do in mouse models...), would have to be done in order to bring a final answer on this interesting issue, we cannot conclude, at least based on these western blot analyses, that mitophagy is dramatically occurring in IECs lacking Stard7 in vivo.

Figure legend: Markers of mitophagy are properly expressed in IECs lacking Stard7. Western blot analyses were carried out with protein extracts from the intestine of both *Stard7^{Lox/lox}* and *Stard7^{ΔIEC}* mice.

Regarding the use of an alternative source of energy seen in cells lacking *Stard7*, we also updated our discussion by adding the reference mentioned by Reviewer 3 (Rojas and colleagues, (2024), *FEBS Journal*, PMID 37846201). The following paragraph was added: “*Mitochondria lacking Stard7 may face OXPHOS deficiency by potentiating FAO as an alternative source of energy. In this context, we demonstrated that the maximum oxygen consumption rate severely decreased upon Stard7 deficiency in cells treated with the FAO inhibitor Etomoxir, which experimentally supports this hypothesis. Interestingly, glycolysis is used as another alternative source of energy in myoblasts lacking Stard7, which suggests that the metabolic reprogramming seen upon Stard7 deficiency is cell type-dependent (Rojas et al, 2024).*”

Regarding the need to add the anti-p4Ebp1 western blot in Figure 1H, please find here below the requested result. As you can see, this blot does not add anything more to our message and a lighter exposure does not further help. We indeed noticed that the total 4Ebp1 blot was actually much better to detect the phosphorylated 4Ebp1 band (i.e. the upper band) in this experiments carried out with extracts from mIECs subjected to the indicated treatments. Therefore, we would prefer not to add this blot in our revised version and we thank Reviewer 3 for his/her understanding.

Figure legend: H₂O₂ triggers mTORC1 activation in a ROS-dependent manner in IECs. Mouse immortalized intestinal epithelial cells (mIECs) were pre-incubated or not with NAC for 1 hour and subsequently left untreated or stimulated with H₂O₂ for 5 hours at the indicated concentrations. Extracts were subjected to western blot analyses using the indicated antibodies.

The missing molecular weights in Figure S2F (now Expanded View Figure 2B in our revised version) have been added and we apologize about this omission.

3. In Fig. 3F the authors observe lower induction of inflammatory cytokines in Stard7^{ΔIEC} mice compared to controls. Could the authors speculate regarding the mechanism driving this difference? Could it be linked to the downregulation of Tigar and upregulation of pH₂AX reported in Fig. 3C? Regardless, the significance of these alterations should be clarified (and the uncropped blots for Tigar, pH₂AX and CC3 should be included in the supporting file). Furthermore, the uncropped blots for Fig. 3E report a lane for Lysozyme, which is not presented in the panel (and shouldn't exist, as the sample is colon). Please check that the blots presented are correct.

Our answer:

We thank Reviewer 3 for pointing out this issue. We updated the result section by adding the following paragraph: « *Consistent with a defective cellular response to oxidative stress, protein levels of the anti-oxidant protein Tigar were decreased in the colon of Stard7^{ΔIEC} mice treated with AOM/DSS (Fig. 3C) (Bensaad et al, 2006). Likewise, levels of pH₂AX, a*

DNA damage marker, were enhanced upon Stard7 deficiency (Fig. 3C). Yet, cleaved Caspase 3 levels were similar in the colon of both AOM/DSS-treated genotypes, indicating that a Caspase 3-dependent cell death pathway was not triggered upon Stard7 deficiency in vivo (Fig. 3C)."

We also updated our discussion by adding the following paragraph : « *It is also interesting to note that tumors from AOM/DSS-treated Stard7^{ΔIEC} mice show lower levels of the anti-oxidant protein Tigar, a candidate whose deficiency enhances the production of cytokines by pancreatic cancer cells (Cheung et al, 2024). Therefore, it is unlikely that lower levels of Tigar seen upon Stard7 deficiency in our inflammatory-driven tumor development model contributes to the decreased mRNA levels of pro-inflammatory cytokines, even if both mouse models of pancreatic and colon cancers may have organ-specific differences* ». We do not feel comfortable to add anything more on this specific issue as we would prefer not to be too much speculative. Indeed, much more would have to be done to precisely define the cellular source of pro-inflammatory cytokines we measured. Although macrophages are expected to be the main source of these cytokines, we cannot rule out the possibility that other cell types, including cancer cells, would contribute to their production. An exhaustive characterization of the underlying mechanisms would bring us beyond the scope of the present study.

Regarding the detection of Lysozyme in the colon, previous studies demonstrated that Lysozyme⁺ cells can indeed be detected in this organ. In one PNAS paper published by Hans Clevers and colleagues, Lysozyme⁺ cells were indeed observed in the colon of their genetically-modified mouse model of intestinal cancers (see Figure 6 of the paper by Paul W. Tetteh and colleagues, PNAS (2016), PMID: 27708166). They defined this as « Paneth cell hyperplasia » in the colon. Likewise, Lysozyme⁺/IDO⁺ cells have also been detected in the colon of Apc^{+/-Min} mice (see the paper by Sandra Pflügler and colleagues published in Communication biology (2020), PMID: 32444775). Given the fact that it is unclear whether we are talking about differentiated Paneth cells or about « Paneth-cell like » cells and because it remains unclear whether this observation critically contributes to the observed phenotype, we decided to remove this blot as it may bring too many questions.

The missing uncropped gels were added in the Uncropped gels file while the anti-Lysozyme blot has been removed from this file.

4. "Stard7 is promoting the expression of Wnt effectors in IECs but it is nevertheless unclear whether Stard7 is required in Wnt-driven tumor initiation". Fig. S2D clearly shows that upon StarD7 deficiency, in normal epithelium, no alteration of Wnt effectors can be observed, as noted in the corresponding paragraph: "mRNA levels of Wnt target genes (Lgr5, c-Myc, Sox9 and Cd44) remained unchanged in IECs lacking Stard7 (Supplementary Fig. 2D)". As per point #1, the statement should be corrected.

Our answer:

We corrected this sentence by writing-up the following statement. « *Stard7 is promoting the expression of Wnt effectors in established colon cancer cell lines showing constitutive Wnt signaling but it is nevertheless unclear whether Stard7 is required in Wnt-driven tumor initiation* ».

5. In Fig. 5, the authors report that the increase in tumor burden in the $Apc^{+/Min}$ model lacking StarD7 is due to increased mutational rate. Is there a previous report suggesting higher instability in $Apc^{+/Min}$ cancers compared to inflammation-driven ones? The parallel between Fig. 3F and Fig. 5D suggests that the differential upregulation of inflammatory molecules is a key difference between the two cancer types in the absence of StarD7, as per point #3.

Our answer:

There is indeed a high chromosomal instability due to the loss of *Apc* (see the paper by Fodde and colleagues (Nature Cell Biology (2001), 3, 433-438, PMID: 11283620). However, the inflammation-driven tumor model also increases the mutational rate, at least through the activation of tumor-associated macrophages known to cause mutations in epithelial cells through ROS production. Our data using ex-vivo organoids indicate that the loss of *Apc* helps epithelial cells to resist to H₂O₂-dependent JNK activation. Therefore, we believe that *Apc*-mutated intestinal cells can better deal with the mitochondrial stress that results from *StarD7* deficiency and therefore resist to the accumulation of mutations seen during tumor development. It is likely that they actually benefit from the accumulation of these mutations to support cancer development in $Apc^{+/Min}/StarD7^{\Delta IEC}$ mice. There is no report, to the best of our knowledge, suggesting higher genomic instability in $Apc^{+/Min}$ mice than in the inflammation-driven model: we think that the *Apc* status is the key parameter that underlies the capacity of transformed intestinal epithelial cells to deal with increasing numbers of mutations during tumor development. Regarding levels of pro-inflammatory cytokines, we and others have demonstrated that the loss of *Apc* disrupts the architecture of the single layer of intestinal epithelial cells, which leads to the establishment of a pro-inflammatory signature, at least through TLR activation (see for example our paper by Goktuna and colleagues, Cancer Research (2016), 76, 2587-2599, PMID: 26980769). Therefore, both Wnt-driven tumor development as well as the inflammation-driven cancer model both rely on an inflammatory signature to support tumor development. However, the key difference for both model is the *Apc* status. As this specific mutational status helps intestinal cells to better cope with ROS levels, several pro-survival oncogenic signaling pathways are robustly activated, which leads to the production of chemokines and to the attraction of Tumor-Associated Macrophages (TAMs), which are the main source of pro-inflammatory cytokines. We believe that this is the reason why $Apc^{+/Min}/StarD7^{\Delta IEC}$ mice show elevated levels of these cytokines. In other words, these pro-inflammatory cytokines contribute to tumor development in both experimental models but their distinct expression profile in control versus *StarD7*-inactivated epithelial cells reflects the opposite outcome on tumor development in both models due to distinct *Apc* status. We thank Reviewer 3 for pointing out this issue and we took this opportunity by adding a paragraph in the discussion of our revised manuscript.

In Fig. 5E but not in 5F, the WB relative to duodenal extracts is presented. Given that the whole characterization of the $Apc^{+/Min}$ model was focused on the colon and on the fact that potential differences between the two organs are not discussed, the relative data should be removed.

Our answer:

We felt that it was important to illustrate the fact that *Stard7* deficiency in IECs potentiates mTORC1 activation in the colon and to a less extent in the duodenum. Indeed, this observation, combined with the specific microbiota signature found in the colon of *Apc^{+Min} Stard7^{ΔIEC}* mice, opens the possibility that the potentiated mTORC1 activation seen in these mice could result from cell-autonomous effects in transformed intestinal epithelial cells (i.e. through a ROS-dependent pathway) but also from effects of oncometabolites produced by specific strains found in the distal colon of these mice. Although currently speculative, this later hypothesis deserves further investigation and we updated the discussion on this issue in the revised manuscript. We thank Reviewer 3 for this comment, which gave us the opportunity to better explain/discuss our findings.

Fig 5F: The link between Gpx4, Coenzyme Q and StarD7 is promising, but it is unclear why ferroptosis would limit tumor development in the *Apc* model and not in the DSS one. The authors should discuss this and, if possible, give a direct measurement of ferroptosis: an extensive array of methods to quantify ferroptosis is available in the literature, including but not limited to 10.1038/cdd.2015.158.

Our answer:

Reviewer 3 is raising a key issue that indeed deserves more experimental data and explanations. We now provide a new immunohistological analysis in which levels of 4-HNE, a toxic product of lipid peroxidation defined as a reliable marker of ferroptosis, are enhanced in the colon of *Apc^{+Min}/Stard7^{ΔIEC}* mice (see our new Figure 5G). This new result, combined with our previous western blot showing elevated GPX4 protein levels in the colon of these mice, strongly suggests that mice showing constitutive Wnt signaling have more cells undergoing ferroptosis in the colon when *Stard7* is genetically inactivated in IECs. While this result appears convincing to us, it remains unclear why this phenomenon is only seen in mice showing constitutive Wnt signaling but not in the inflammation-driven model of colon cancer. At this stage, only hypothesis that would deserve some experimental validation can be made but this may bring us beyond the scope of this manuscript. The link between STARD7 deficiency and ferroptosis was made by Thomas Langer and colleagues. They demonstrated that ferroptosis was due to defects in the intracellular transport of coenzyme Q in cells lacking STARD7 (Deshwal et al., Nature Cell Biology, 2023, PMID: 36658222). This study, combined with our results, suggests that STARD7 regulates the intracellular transport of coenzyme Q in a context-dependent manner. This statement is valid only if we assume that the ferroptosis seen upon *Stard7* deficiency is exclusively due to defects in the intracellular transport of coenzyme Q and not to any other mechanisms. It is however totally unclear which parameters would be part of this specific cellular context (amount of coenzyme Q to transport that may differ from one cell type to the other? Distinct levels of enzymes involved in coenzyme Q synthesis? Compensatory mechanisms to prevent ferroptosis in some cell types? etc...). As this is only speculations, we would prefer not to discuss about this issue in the revised manuscript.

6. In Fig. 6, the authors show reduced OXPHOS as measured by Complex I activity; in Fig. S5, they identify increased intermediates of the TCA cycle, suggesting that the metabolic demands of the tumor in *Apc^{+Min}/Stard7^{ΔIEC}* conditions are fueled mostly by glycolysis, which feeds pyruvate into the TCA. It seems that many of the effects seen in

the $Apc^{+/Min}$ background (the lipid signature and reduced activity of Complex I, for example) are similar to the ones seen in non-tumoral tissue and just depend on the $Stard7$ status. Could the authors clarify which alterations are tumor-specific, and among those, which are specific to the DSS model but not the $Apc^{+/Min}$ one?

Our answer:

Reviewer 3 is raising the central issue of our study: despite very similar (if not identical...) signatures (see our transcriptomic and lipidomic analyses for example), the consequences of $Stard7$ deficiency in Wnt-driven or in inflammation-driven tumor development are totally different. This is why we believe that the mutational status (i.e. the Apc status) has to be taken into account when aiming to understand the context-dependent role of $Stard7$ in intestinal tumor development. This fact is actually also relevant for the role of mTORC1 in the same experimental models. mTORC1 (whose activity is inversely correlated to $Stard7$ expression in our experimental model) has been described as a positive regulator of Wnt-driven tumor initiation but as a negative regulator of inflammation-driven tumor development in the intestine (cf the paper published in *Cell Metabolism* by Marta Brandt and colleagues, PMID: 29275959 and cited in our discussion). Our data are very consistent with this study.

7. Fig. 7B highlights the effects of NAC on the molecular signature of the tumor. Is the effect applicable to $Stard7^{lox/lox}$ mice as well? It is unclear from the blots presented, and if so, it would invalidate the point that ROS are a main driver of tumor formation in the specific context of $Stard7$ deficiency. Does the treatment affect tumor number/area? A characterization as seen in Fig. S6B-D for Rapamycin should be included.

Our answer:

We carried out the requested experiment and surprisingly found that NAC did not interfere with the number of tumors in the distal colon of both $Apc^{+/Min}/Stard7^{Lox/lox}$ and $Apc^{+/Min}/Stard7^{\Delta IEC}$ mice (Fig. 7B, right panels). Therefore, NAC may have mTORC1-independent and pro-tumoral effects in addition to anti-tumoral effects through mTORC1 inhibition (Fig. 7B, right panels). How can we explain this result? NAC unexpectedly promotes intestinal tumor progression in $Apc^{+/Min}$ mice (Zou ZV et al., *Antioxydants* (2021), 10, 241, PMID: 33557356)). This new reference has been added in the revised discussion. Although underlying mechanisms remain unclear, it was postulated that low doses of NAC (i.e. doses we also used in our study) could neutralize damaging ROS produced by mitochondria while having no effects on proliferative ROS produced through a RAC1-dependent pathway (Zou *et al*, 2021; Cheung *et al*, 2016). Regarding effects on $Stard7^{Lox/lox}$ mice, our data show that $Stard7$ deficiency leads to similar molecular consequences (ROS production and mTORC1 activation) but with consequences on tumor development which depends on the Apc status. In other words, looking at effects of ROS on the molecular signature of intestinal tumors (and more specifically on the mTORC1/Atf4 signaling cascade) cannot predict the outcome of ROS on tumor development if we do not take the Apc status into account in our experimental models.

8. In Fig. 7C, 4Ebp1 phosphorylation seems to be decreased in the WT Apc condition and not at all triggered in $Apc^{+/Min}$ background, contrary to what is stated in the text.

Similarly, Wnt effectors appear to decrease in the WT background upon ROS stimulation. Is it linked to severe increase in apoptosis, as pJnk suggests? In that case, the highest H₂O₂ dose should not be taken into account.

Our answer:

We updated this Figure 7C by adding a new blot for p4Ebp1 (shorter exposure) to better see the phosphorylated versus the non phosphorylated form of p4Ebp1. In contrast to what is stated in the technical sheet of this antibody, we can indeed detect the unphosphorylated form of 4Ebp1 in addition to the phosphorylated form using this antibody. With this exposure time, we clearly see that H₂O₂ enhances the phosphorylated/unphosphorylated ratio of 4Ebp1 in ex-vivo organoids generated from both wild type Apc-expressing and Apc^{+Min} mice, which means that H₂O₂ triggers mTORC1 activation independently of the Apc status. This conclusion is actually in perfect agreement with our data showing that p4EBP1 levels are increased, even in healthy (i.e. wild type Apc-expressing mice) lacking Stard7 in IECs. However, consequences of H₂O₂ treatment are indeed dramatically distinct on JNK phosphorylation as well as on the profile of Wnt effectors, as pointed out by Reviewer 3. Whether a direct link exists between the lack of JNK phosphorylation and the induction of Wnt effectors such as c-Myc in our experimental model is currently unclear. Let's note that c-Myc has been demonstrated to inhibit JNK-dependent cell apoptosis *in vivo*, at least in *Drosophila* (Huang J, Feng Y, Chen X, Li W, Xue L, Apoptosis (2017), 22, 479-490, PMID : 28150056). We added this reference in the result section and we thank Reviewer 3 for having brought this issue.

To more specifically address consequences downstream of H₂O₂-dependent JNK phosphorylation in Apc-mutated versus WT Apc-expressing ex-vivo organoids, we tested a variety of effectors of cell apoptosis, including c-Jun phosphorylation, a hallmark of JNK activation. This new experiment confirmed that the Wnt effector Sox9 was induced by H₂O₂ in Apc-mutated but not in WT Apc-expressing ex-vivo organoids (see our new Figure illustrated here after). In agreement with JNK activation seen upon H₂O₂ stimulation in WT Apc-expressing but not in Apc-mutated ex-vivo organoids, we saw elevated levels of phosphorylated c-Jun (of note, total c-Jun levels were elevated too). It is also important to note that Hsp90 is known to be cleaved by Caspase 10 in cells subjected to an oxidative stress and undergoing apoptosis (see the study published by R. Beck and colleagues in *Biochemical Pharmacology*, 2009, PMID : 19014912). This is what we saw again in WT Apc-expressing ex-vivo organoids treated with H₂O₂ at high concentrations. While we cannot rule out the possibility that other types of JNK-dependent cell death pathways may also occur in WT Apc-expressing ex-vivo organoids treated with H₂O₂, our results clearly demonstrate that the Apc status critically define the outcome of ex-vivo organoids subjected to an oxidative stress. We also would like to say that an exhaustive characterization of these pro-cell death pathways in ex-vivo organoids is technically very challenging given the fact that high numbers of dying cells are seen, especially at the heart of these 3D structures. As a result, despite multiple attempts, it is very difficult to provide convincing results on other effectors of apoptosis such as cleaved Caspase 3 or Bax levels for example. The quality of many commercially-available antibodies for mouse ex-vivo organoids should also improve. These conclusions were raised based on our 13 years of experience in dissecting signaling pathways in ex-vivo organoids.

Figure legend: H₂O₂-dependent cell death in WT *Apc*-expressing but not in *Apc*-mutated ex-vivo organoids. Extracts from the indicated ex-vivo organoids were subjected to western blot analyses using the indicated antibodies.

9. Fig. 8 shows convincing data proving that antibiotics reduce the tumor load in *Apc*^{+/*Min*} *Stard7*^{ΔIEC} mice, but it is difficult to evaluate if the effect is specific for the *Stard7*-deficient status, given that *Stard7*^{lox/lox} mice develop almost no tumors (similarly to what is seen in Fig. S6D, but not in Fig. 4D). It is unclear what this experiment adds compared to the paper Li et al, 2012 that the authors cite. Are the reported alterations in microbiota composition specific to *Apc* loss or are they conserved in mice with DSS induced carcinogenesis/no tumor background?

Our answer:

We thank Reviewer 3 for raising this interesting issue. It is currently unclear whether this modified microbiota composition seen in *Apc*^{+/*Min*}/*Stard7*^{ΔIEC} mice actively contribute or not to tumor development. We can only speculate at this stage and future studies will be conducted in order to find any oncogenic metabolite produced by bacterial strains enriched in our model. As an attempt to assess whether or not any difference in the microbiota composition that would result from *Stard7* deficiency could also be seen in a no tumor background, we established the microbiota composition in both *Stard7*^{Lox/lox} and *Stard7*^{ΔIEC} mice treated with DSS. As illustrated here below, we did not find any significant changes in the microbiota composition in both DSS-treated genotypes. As our paper already contains many panels, we would prefer not to show this new result in the manuscript.

Figure legend: Stard7 deficiency in intestinal epithelial cells does not significantly impact the microbiota composition in DSS-treated mice (see methods in the manuscript for experimental details). Cecal microbiota composition was determined using 16S rRNA amplicon sequencing (V4 region) (n = 7 mice per genotype) and the top phyla are shown in bar plots per genotype. Mice are individually represented in a principal coordinate analysis (PCA) with a Weight Unifrac distance metric. The most abundant taxa are shown at the level of Phylum.

10. In the discussion, the authors speculate that "Mitochondria lacking Stard7 may face OXPHOS deficiency by potentiating FAO as an alternative source of energy". This is a good suggestion, and I don't agree that it is unreasonable due to mitochondrial stress: as I discussed above, lower OXPHOS rate does not equal mitochondrial stress or inability to do FAO. The hypothesis should be tested by inhibiting FAO with drugs such as Etomoxir in IECs or organoids and checking for its contribution to total ATP production.

Our answer:

We carried out the experiments requested by Reviewer 3. We now show that ATP production was impaired in mIECs lacking Stard7 (see our new Figure 1E). Moreover, mIECs lacking Stard7 critically rely on fatty acid oxidation (FAO) as the maximal oxygen consumption rate (OCR) dropped in Stard7-depleted but not in control cells when treated with Etomoxir, a FAO inhibitor (see our new Figure 1G).

11. DNA and RNA sequencing data are reported to be aligned on the human reference genome. It is probably a mistake and should be corrected in the text. Overall, most of the main messages in the present work are solid and well-constructed. I believe that the experimental requests will be reasonably feasible and that most of the points I raised could be tackled by integrating the discussion or reformulating the phrasing of the text. If these are satisfactorily addressed, I recommend the publication in EMBO Molecular Medicine.

Our answer:

We apologize for these mistakes. We indeed aligned both DNA and RNA sequencing data to the mouse reference GRCm38 and GRCm39 release 107, respectively. These mistakes have been corrected in the revised version. We would like to sincerely thank Reviewer 3 for his/her numerous insightful comments on our study. We now provide a revised version in which all comments were experimentally addressed and corrections were made.

16th Jan 2026

Dear Prof. Chariot,

Thank you for submitting your revised manuscript to EMBO Molecular Medicine. The manuscript has been reviewed by the three initial referees. As you will see below, while referee #2 is satisfied with the revisions, referees #1 and #3 still raise some concerns on the revised manuscript. We further discussed the reports within our editorial team, and agreed not to ask for additional mechanistic experiments, as suggested by referee #1. However, we would like to invite you to further revise the manuscript to address referee #3' comments.

As EMBO Press usually only allows one round of revisions, please be aware that this will be your last opportunity to address these remaining issues. The revised manuscript will be reviewed again, and we cannot guarantee a positive outcome at this stage.

Additionally, please address the following editorial matters:

1/ please correct the titles in the figure legends to "Figure EV1" etc.

2/ please ensure that all funders listed in the Acknowledgments are also entered into the funders list in our system.

3/ author contributions: please provide CRediT (Contributor Role Taxonomy) terms in the submission system. These replace a narrative author contribution section in the manuscript

4/ please provide the Paper Explained: EMBO Molecular Medicine articles are accompanied by a summary of the articles to emphasize the major findings in the paper and their medical implications for the non-specialist reader. Please provide a draft summary of your article highlighting

5/ rename the suppl. tables Table EV1 and Table EV2 and upload them as separate files. Please correct the entries for Table EV1 and Table EV2 in the reagent table.

6/ please remove "data not shown". As per our policy, all results discussed in the manuscript must be shown in the main or supplementary figures.

7/ Synopsis:

- please resize your visual abstract to a file 550 px wide x 300-600 px high.

- please remove the synopsis text from the manuscript file and upload it as a separate file.

9/ there are callouts left in the text for a Supplementary Fig. 1A and a Supplementary Fig. 4, please correct.

8/ please address the queries from our data editors in the figure legends:

- Please note that the exact p values are not provided in the legends of figures 1D, E, F; 2A, B; 3F, 4B, C, D, E, F, G; 5B, C, D, G; 6B, F; 7B, 8D-F; EV2 D, EV3 A, C; EV4 A, C; EV5 C, EV8 D

- Please note that the box plots need to be defined in terms of minima, maxima, centre, bounds of box and whiskers, and percentile in the legends of figures 1C, 7A, 8D, E, F, G

- Please note that information related to n is missing in the legends of figures 1C, D, F; 7A, 8D-G; EV5 B

- Please note that the error bars are not defined in the legends of figures 2A, B; 4B, D, E, F, G; 5B, C; 6F, 7B, EV1 A, B; EV4 A, B; EV5 B, EV8 D

- Please note that for heatmap present in figures 1H, 3A, 6A, D a numbered scale bar is not provided. This needs to be rectified.

Should you find that the requested revisions are not feasible within the constraints outlined here and prefer, therefore, to submit your paper elsewhere, we would welcome a message to this effect.

I look forward to seeing a revised form of your manuscript as soon as possible. Use this link to login to the manuscript system and submit your revision: <https://embomolmed.msubmit.net/cgi-bin/main.plex>

Yours sincerely,

Lise Roth

***** Reviewer's comments *****

Referee #1 (Comments on Novelty/Model System for Author):

The revised manuscript needs to be enhanced regarding how STARD7 KO affects lipid change in the tumor, which therefore leads to other determined changes and the tumor growth.

Referee #2 (Comments on Novelty/Model System for Author):

The authors made new experiments and showed new data to address my previous observations and concerns. They also clearly explain the difficulties to resolve some technical aspects to address the issues. They built a final work that conveys different top experimental techniques to provide valuable findings in the context of lipid metabolism and tumorigenesis.

Referee #2 (Remarks for Author):

The new version of the manuscript is improved, correctly address previous concerns and delivers the main idea and findings of the study.

Referee #3 (Comments on Novelty/Model System for Author):

From a methodological point of view, I have a serious issue with the lack of quantification and statistical analysis on Western blot experiments. I recommend that the journal ensures that these data meet the highest standards of rigor and transparency before publication. Given the inherent complexity of Western blot experiments, careful quantification and clear presentation are essential, particularly as the main conclusions rely heavily on these results.

Referee #3 (Remarks for Author):

I'd like to preface my comments on this revised version by sincerely thanking the authors for the time and effort they dedicated to addressing my comments and those of my colleagues. I acknowledge the experimental effort to address most of my requests. In particular, several new citations and discussion points were included in the manuscript or extensively discussed in the response to my comments, rendering the authors' conclusions more compelling.

From an experimental perspective, the revised version is much improved. However, some methodological concerns I previously raised remain unresolved, preventing the paper from being acceptable. Below, I provide my remaining comments:

- The authors claim that several UPR factors are not dramatically enhanced upon Stard7 deficiency, but the UPR term by GSEA is significantly induced in KO mice, as the text states: "...both Unfolded Protein Response and mTORC-dependent signaling pathways were potentiated upon Stard7 deficiency ... (Fig. 3A)". This should be commented on.

- My previous concerns regarding the appropriate representation and quantification of Western blot data have not been adequately addressed. While I appreciate the inclusion of uncropped raw images as supplementary material, in line with journal policy, the lack of systematic densitometric analysis remains unresolved.

In a manuscript where 12 out of 15 figures rely on Western blotting, it is essential to provide quantitative analyses, at minimum provided in EV figures if not in main. Several blots are not readily interpretable by visual inspection alone, particularly where antibodies produce multiple nonspecific bands (for example, Xbp1s and Bip in EV Figure 9, each showing 3-4 bands).

Arrowheads pointing to the band(s) considered as specific signal should be included.

The "densitometry analyses" provided as Excel files in the raw data do not sufficiently address this issue. Quantification is missing for a substantial fraction of the Western blot panels despite my previous request, and, where present, only a subset of the reported markers has been measured, without accompanying statistical analysis.

Given the existence of clear reporting guidelines (including those outlined in the EMBO Molecular Medicine author instructions), the current presentation does not meet the expected standards for data interpretation and rigor.

- I understand the frustration of the authors over the lack of standardized ROS measurement tools in vivo. The choice of 4-HNE staining as a proxy for lipid peroxidation is methodologically sound and connects well with the Gpx4 data that was already present in the first submission. I would recommend however showing two comparable areas of the colon in Figure 5G, as the WT image represents a normal epithelium while the KO one appears to show a tumor: do the authors wish to conclude that the percentage of 4-HNE cells is different because KO mice have more tumors/tumor area? If so, it can be explicitly stated. Incidentally, 4-HNE should be written in the figure panel, and the graph should state "% of 4-HNE-positive cells" on the Y axis, to improve clarity and readability of the figure.

Gpx4 is shown overexpressed by Western blot in Apc+/Min KO mice in Figure 5F, but downregulated in Apc+/Min KO mice in the reactive oxygen species pathway enrichment plot presented in Figure 6A. This discrepancy should be addressed in the text.

Overall, the authors have done a remarkable work strengthening the experimental evidence for their claims. The message of the paper has not changed compared to the first submission, but multiple points have been better characterized, and the overall story is convincing. The remaining comments I have on the manuscript are more clarifications about data interpretation and

should not impact on the general message.

However, the methodological concerns regarding Western blot presentation that I raised in the first round of revision have not been addressed. I do not feel comfortable recommending the publication of this paper in EMBO Molecular Medicine until the authors provide the quantifications and appropriate statistical analysis for all proteins presented in their figures. These quantitative analyses are necessary to fully support the conclusions drawn from the Western blots experiments and should be considered as an essential part of data submission.

Referee #3 (Remarks for Author):

I'd like to preface my comments on this revised version by sincerely thanking the authors for the time and effort they dedicated to addressing my comments and those of my colleagues. I acknowledge the experimental effort to address most of my requests. In particular, several new citations and discussion points were included in the manuscript or extensively discussed in the response to my comments, rendering the authors' conclusions more compelling.

From an experimental perspective, the revised version is much improved. However, some methodological concerns I previously raised remain unresolved, preventing the paper from being acceptable. Below, I provide my remaining comments:

Our answer:

We would like to thank again Reviewer 3 for his/her insightful and constructive comments. Those helped us to improve our study and we will now address his/her remaining concerns.

The authors claim that several UPR factors are not dramatically enhanced upon Stard7 deficiency, but the UPR term by GSEA is significantly induced in KO mice, as the text states: "...both Unfolded Protein Response and mTORC-dependent signaling pathways were potentiated upon Stard7 deficiency ... (Fig. 3A)". This should be commented on.

Our answer:

We thank Reviewer 3 for bringing this issue out, which indeed, needs more clarification. The UPR signature found in GSEAs carried out with extracts from KO mice refers to elevated levels of Atf4 (an UPR effector) and downstream targets of Atf4 such as Psat1 and Asns for example. In fact, there is an overlapping signature with the enrichment plot for hallmarks of mTORC1 signaling as both Asns and Psat1 are also part of this latter signature. This is the reason why we are talking about the mTORC1-ATF4 pathway whose activation is potentiated in intestinal epithelial cells lacking Stard7. Given the fact that Atf4 is defined as an UPR effector, GSEAs systematically highlight an UPR signature when Atf4 levels targets are upregulated, even in circumstances in which other UPR branches do not change. We added the following sentences in the revised version :

Results section on page 13: « *as evidenced by elevated Atf4 mRNA levels for example* ».

Discussion section on page 25: “*The UPR signature found in our GSEAs carried out with extracts lacking Stard7 actually resulted from elevated levels of Atf4 and downstream targets rather than from the activation of all UPR branches.*”

My previous concerns regarding the appropriate representation and quantification of Western blot data have not been adequately addressed. While I appreciate the inclusion of uncropped raw images as supplementary material, in line with journal policy, the lack of systematic densitometric analysis remains unresolved. In a manuscript where 12 out of 15 figures rely on Western blotting, it is essential to provide quantitative analyses, at minimum provided in EV figures if not in main. Several blots are not

readily interpretable by visual inspection alone, particularly where antibodies produce multiple nonspecific bands (for example, Xbp1s and Bip in EV Figure 9, each showing 3-4 bands). Arrowheads pointing to the band(s) considered as specific signal should be included. The "densitometry analyses" provided as Excel files in the raw data do not sufficiently address this issue. Quantification is missing for a substantial fraction of the Western blot panels despite my previous request, and, where present, only a subset of the reported markers has been measured, without accompanying statistical analysis. Given the existence of clear reporting guidelines (including those outlined in the EMBO Molecular Medicine author instructions), the current presentation does not meet the expected standards for data interpretation and rigor.

Our answer:

The requested information has now been added in the revised version. The requested arrows have been added on some western blots when needed (see the anti-Bip blot illustrated in EV Figure 9 for example). Note that some western blots show multiple bands (see the anti-4Ebp1 blots for example) but this reflects the numerous phosphorylated forms of this protein. For Figure 7, both Jnk1/2 isoforms are detected. It is also fair to say that many data obtained with western blots analyses were confirming key results obtained through both transcriptomic and proteomic analyses (see for example Figures 1I, 3D, 6C, 6E). Densitometry analyses are illustrated in Fig. EV10. As our main figures already have a lot of panels, it would render them unreadable if we add these densitometry analyses next to corresponding western blots in those main figures. Therefore, we had no choice but to put these quantifications in EV figures. This Figure EV10 now has up to 23 slides. We thank Reviewer 3 for his/her understanding. The link with Figure EV10 has been mentioned in the methods section.

I understand the frustration of the authors over the lack of standardized ROS measurement tools in vivo. The choice of 4-HNE staining as a proxy for lipid peroxidation is methodologically sound and connects well with the Gpx4 data that was already present in the first submission. I would recommend however showing two comparable areas of the colon in Figure 5G, as the WT image represents a normal epithelium while the KO one appears to show a tumor: do the authors wish to conclude that the percentage of 4-HNE cells is different because KO mice have more tumors/tumor area? If so, it can be explicitly stated. Incidentally, 4-HNE should be written in the figure panel, and the graph should state "% of 4-HNE-positive cells" on the Y axis, to improve clarity and readability of the figure.

Our answer:

We agree with Reviewer 3 that the percentage of 4-HNE-positive cells seen in KO mice results from the enhanced tumor development, at least based on the illustrated figure 5G. We added the following sentence in our revised version on page 16: « *We found that 4-HNE positive cells were more detected in tumors rather than in adjacent normal tissues in these mice (Fig. 5G). As a result, the higher percentage of 4-HNE positive cells found in tissues from $Apc^{+Min}/Stard7^{\Delta EEC}$ mice resulted from enhanced tumor development* ».

Gpx4 is shown overexpressed by Western blot in Apc^{+Min} KO mice in Figure 5F, but

downregulated in $Apc^{+/Min}$ KO mice in the reactive oxygen species pathway enrichment plot presented in Figure 6A. This discrepancy should be addressed in the text.

Our answer:

We thank Reviewer 3 for pointing out this issue. This gave us the opportunity to clarify this issue in this rebuttal letter as well as in the revised version. Data illustrated on Figure 6A relates to GPX4 mRNA levels while Figure 5F concentrates on GPX4 protein levels. We have seen throughout the years that positive correlation between both mRNA and protein levels of any candidate (and especially in tumor tissues...) cannot be systematically done, given the numerous post-translational modifications that can occur with important consequences on protein levels. We can only be speculative here but it is possible that *Stard7* deficiency in $Apc^{+/Min}$ mice enhances *Gpx4* stability by preventing its degradation for example. As we would go beyond the scope of this manuscript, we would prefer not to go further than mentioning this hypothesis in the revised version. The following sentence was added on page 17 : « *Of note, Gpx4 mRNA levels were downregulated in tissues from $Apc^{+/Min}/Stard7^{ΔIEC}$, which is in contrast to elevated Gpx4 protein levels found in these tissues, which possibly reflects the involvement of post-translational modifications of Gpx4 occurring upon Stard7 deficiency (Fig. 5F)* ».

Overall, the authors have done a remarkable work strengthening the experimental evidence for their claims. The message of the paper has not changed compared to the first submission, but multiple points have been better characterized, and the overall story is convincing. The remaining comments I have on the manuscript are more clarifications about data interpretation and should not impact on the general message. However, the methodological concerns regarding Western blot presentation that I raised in the first round of revision have not been addressed. I do not feel comfortable recommending the publication of this paper in EMBO Molecular Medicine until the authors provide the quantifications and appropriate statistical analysis for all proteins presented in their figures. These quantitative analyses are necessary to fully support the conclusions drawn from the Western blots experiments and should be considered as an essential part of data submission.

Our answer:

The requested information has now been added in the revised version. We thank Reviewer 3 for his/her constructive work.

17th Feb 2026

Dear Prof. Chariot,

Thank you for submitting your revised study. We have now received the report from referee #3, who is satisfied with the revisions. I will therefore be able to accept your manuscript once the following editorial matters are addressed:

1/ Manuscript text:

- Please remove the yellow highlights and only keep in track changes mode any new modification.
- Material and Methods should be renamed Methods:
 - o Cells: please indicate whether the cells were authenticated.
 - o Mice: for all, please provide age and gender at time of experiments.
 - o Antibodies: please provide dilutions.
 - o Statistics: provide a statement on blinding, randomization, sample size, and inclusion/exclusion criteria.
- Data Availability: please provide URLs for the deposited datasets, including lipidomics and metabolomics.

2/ Figures:

- For figure panels with n=2, please only indicate individual data points and remove error bars and statistics.
- Thank you for providing Figure EV10 in response to the referee's request. Unfortunately, figure EV10 cannot be published as it is, due to formatting issues. I would suggest to make it an Appendix file, with table of content and page numbers. I would also propose to remove the Western blots and only keep the quantifications. If you choose to keep the blots, please indicate reuse in the figure legends.
- Figure reuse is allowed, but should be indicated in the figure legends (Figure 4G and Figure EV8B).

3/ Author checklist:

- please fill the right column indicating where the information is provided.
- please check the section Experimental animals/animal observed in or captured from the field, as I don't think it applies to your study.
- please fill the entire section "Experimental study design and statistics".

4/ Source Data: please provide the Source Data for stainings in Figures 2, 4 and 5.

5/ I introduced minor edits in The paper explained, please let me know if you agree:

Medical issue

Colon cancer is the third most common cancer and the second leading cause of cancer-related deaths worldwide. There is an urgent need for new targets to inform the design of new therapeutic approaches to be used in combination with existing drugs. Defining new targets also relies on generating and characterising mouse models of colon cancer that closely mimic human colon malignancies. In this context, more than 80% of clinical cases of colon cancer display constitutive activation of the Wnt signalling pathway. However, Apc+/Min mice, in which this oncogenic pathway is constitutively activated, exhibit comparatively few tumours in the distal colon.

Results

Apc+/Min mice lacking the lipid transfer protein Stard7 in intestinal epithelial cells (Apc+/Min/Stard7-IEC mice) exhibit accelerated colon tumour formation and thus effectively mimic human colon malignancies. The microbiota signature found in the colon of these mice is also present in patients with colon cancer. While Stard7 acts as a tumour suppressor gene in Wnt-driven tumour development, it functions as an oncogenic candidate in inflammation-driven cancer models. Therefore, the mutational status of intestinal tumours must be considered when investigating the role of any candidate gene in cancer development.

Clinical impact

Apc+/Min/Stard7-IEC mice are a suitable model for exploring the contribution of intestinal microbiota to tumour development in the distal colon.

6/ I introduced minor modifications in your synopsis, please let me know if you agree or amend as you see fit:

"The effects of STARD7, a lipid transfer protein and negative regulator of mTORC1, on intestinal tumor development depend on the Apc mutational status. Apc+/Min/Stard7-IEC mice are a useful model for exploring the mechanisms underlying colonic tumour formation.

- The loss of the Stard7 lipid transfer protein in intestinal epithelial cells leads to increased ROS production and strengthens the mTORC1/ATF4 signaling cascade.
- A deficiency in Stard7 in intestinal epithelial cells blocks inflammation-driven tumor development, but enhances colonic tumorigenesis in the presence of constitutive Wnt signaling.
- APC+/Min/Stard7-IEC mice provide a good model for human intestinal cancers."

Thank you for resizing your visual abstract. Please check the top of the image, as it seems cropped a bit too tight. Dimensions

should remain 550px x 300-600px.

7/ As part of the EMBO Publications transparent editorial process initiative (see our Editorial at <http://embomolmed.embopress.org/content/2/9/329>), EMBO Molecular Medicine will publish online a Review Process File (RPF) to accompany accepted manuscripts.

This file will be published in conjunction with your paper and will include the anonymous referee reports, your point-by-point response and all pertinent correspondence relating to the manuscript. Let us know whether you agree with the publication of the RPF and as here, if you want to remove or not any figures from it prior to publication.

I look forward to receiving your revised manuscript.

Yours sincerely,

Lise Roth

***** Reviewer's comments *****

Referee #3 (Remarks for Author):

I would like to thank the authors again for their thorough efforts in addressing my comments on the manuscript. The revised version satisfactorily addresses the points I previously raised and is improved in both rigor and clarity.

Overall, the study presents a compelling and well-supported body of work on the role of Stard7 in mouse models of colon tumorigenesis: I therefore consider the manuscript in its current form suitable for publication in EMBO Molecular Medicine.

The authors addressed the remaining editorial issues.

9th Mar 2026

Dear Prof. Chariot,
Dear Alain,

Thank you for providing the revised manuscript file and updating the links to the lipidomics data. Please note that the datasets must be publicly accessible before online publication of the manuscript.

I am pleased to inform you that your manuscript is accepted for publication and is now being sent to our publisher to be included in the next available issue of EMBO Molecular Medicine.

You may qualify for financial assistance for your publication charges - either via a Springer Nature fully open access agreement or an EMBO initiative. Check your eligibility: <https://link.springer.com/journal/44321/how-to-publish-with-us>

With kind regards,

Lise

>>> Please note that it is EMBO Molecular Medicine policy for the transcript of the editorial process (containing referee reports and your response letter) to be published as an online supplement to each paper. If you do NOT want this, you will need to inform the Editorial Office via email immediately. More information is available here: <https://link.springer.com/partners/embo-press/editorial-policies#Peer%20review>